

# Estimation of rate coefficients and branching ratios for reactions of organic peroxy radicals for use in automated mechanism construction

Michael E. Jenkin[1,2], Richard Valorso[3], Bernard Aumont[3], Andrew R. Rickard[4,5]

[1] Atmospheric Chemistry Services, Okehampton, Devon, EX20 4QB, UK
[2] School of Chemistry, University of Bristol, Cantock's Close, Bristol, BS8 1TS, UK
[3] LISA, UMR CNRS 7583, Université Paris Est Créteil et Université Paris Diderot, Institut Pierre Simon Laplace, 94010 Créteil, France
[4] Wolfson Atmospheric Chemistry Laboratories, Department of Chemistry, University of York, York, YO10 5DD, UK
[5] National Centre for Atmospheric Science, University of York, York, YO10 5DD, UK

*Correspondence to*: Michael E. Jenkin (atmos.chem@btinternet.com)

**Abstract.**

Organic peroxy radicals ($RO_2$), formed from the degradation of hydrocarbons and other volatile organic compounds (VOCs), play a key role in tropospheric oxidation mechanisms. Several competing reactions may be available for a given $RO_2$ radical, the relative rates of which depend on both the structure of $RO_2$ and the ambient conditions. Published kinetics and branching ratio data are reviewed for the bimolecular reactions of $RO_2$ with NO, $NO_2$, $NO_3$, OH and $HO_2$; and for their self-reactions and cross-reactions with other $RO_2$ radicals. This information is used to define generic rate coefficients and structure-activity relationship (SAR) methods that can be applied to the bimolecular reactions of a series of important classes of hydrocarbon and oxygenated $RO_2$ radical. Information for selected unimolecular isomerization reactions (i.e. H-atom shift and ring-closure reactions) is also summarised and discussed. The methods presented here are intended to guide the representation of $RO_2$ radical chemistry in the next generation of explicit detailed chemical mechanisms.

## 1 Introduction

Organic peroxy radicals ($RO_2$) are important intermediates in the tropospheric degradation of hydrocarbons and other volatile organic compounds (VOCs). It is well established that their chemistry plays a key role in the mechanisms that generate ozone ($O_3$), secondary organic aerosol (SOA) and other secondary pollutants (e.g. Lightfoot et al., 1992; Jenkin and Clemitshaw, 2000; Tyndall et al., 2001; Archibald et al., 2009; Orlando and Tyndall, 2012; Ehn et al., 2017), and rigorous representation of their chemistry is therefore essential for chemical mechanisms used in chemistry-transport models. As discussed in the preceding papers in this series (Jenkin et al., 2018a; 2018b), they are formed rapidly and exclusively from the reactions of $O_2$ with the majority of carbon-centred organic radicals (R) (reaction R1), these in turn being produced from



the reactions that initiate VOC degradation (e.g. reaction with OH radicals), or from other routes, such as decomposition of larger oxy radicals (M denotes a third body, most commonly $N_2$ or $O_2$ under atmospheric conditions):

$$R + O_2 (+M) \rightarrow RO_2 (+M) \tag{R1}$$

Under tropospheric conditions, a given $RO_2$ radical may have several competing reactions available, the relative rates of which are dependent both on the prevailing ambient conditions and on the structure of $RO_2$. These include a series of bimolecular reactions (i.e. with NO, $NO_2$, $NO_3$, OH and $HO_2$; and the self-reaction and cross-reactions with the multitude of other $RO_2$ radicals present in the atmosphere), which are generally available for all $RO_2$ radicals; and specific unimolecular isomerization reactions (i.e. H-atom shift or ring-closure reactions), that are potentially available for some classes of $RO_2$. The propagating channel of the reaction of $RO_2$ with NO (reaction R2a) plays a key role in tropospheric $O_3$ formation, through oxidising NO to $NO_2$, and also usually represents the major reaction for $RO_2$ radicals under comparatively polluted conditions:

$$RO_2 + NO \rightarrow RO + NO_2 \tag{R2a}$$

The efficiency of this reaction is influenced by the relative importance of the other reactions available for a given $RO_2$ radical. The contribution of the terminating channel of the reaction of $RO_2$ with NO (forming an organic nitrate product, $RONO_2$) depends on the structure and size of $RO_2$; and the reaction of $NO_2$ with selected $RO_2$ radicals forms stable peroxynitrate products, $ROONO_2$. The formation, transport and degradation of these oxidised organic nitrogen reservoirs from the $RO_2$ + NO and $RO_2$ + $NO_2$ reactions has potential impacts in a number of ways, ranging from the inhibition of $O_3$ formation on local/regional scales to influencing the global budget and distribution of $NO_x$ and $O_3$ (e.g. Perring et al., 2013). The reactions of $RO_2$ radicals with $NO_3$ primarily play a role during the night-time in moderately polluted air, providing a radical propagation route that potentially supplements night-time chain oxidation processes (e.g. Carslaw et al., 1997; Bey et al., 2001a; 2001b; Geyer et al., 2003; Walker et al., 2015).

The reactions with OH, $HO_2$ and the pool of $RO_2$ radicals gain in importance as the availability of $NO_x$ becomes more limited, and therefore also inhibit $O_3$ formation by competing with reaction (R2a). In many cases, the reactions are significantly terminating and collectively make a major contribution to controlling atmospheric free radical concentrations under $NO_x$-limited conditions, although the branching ratios for the propagating and terminating reaction channels depend on the structure of $RO_2$. For some classes of $RO_2$, unimolecular isomerization reactions can compete with (or dominate over) the bimolecular reactions. These reactions therefore potentially play an important role in $HO_x$ radical recycling under $NO_x$-limited conditions, and in rapid chain oxidation mechanisms generating highly oxidised multifunctional molecules, HOMs (e.g. Peeters et al., 2009; 2014; Crounse et al., 2013; Ehn et al., 2014; 2017; Jokinen et al., 2014; Rissanen et al., 2015). The relative contributions of the various reactions available for $RO_2$ thus influence the distribution and functional group content of the oxidized products formed, and their physicochemical properties (e.g. volatility and solubility), and therefore the SOA formation propensity of the chemistry.



In this paper, published data on the kinetics and branching ratios for the above bimolecular reactions of hydrocarbon and oxygenated $RO_2$ radicals, and for selected unimolecular isomerization reactions, are reviewed and discussed. The information is used to define and document a set of rules and structure-activity relationship (SAR) methods (a chemical protocol) to guide the representation of the $RO_2$ reactions in future detailed chemical mechanisms (Vereecken et al., 2018).

In particular, the methods presented below are being used to design the next generation of explicit mechanisms based on the Generator for Explicit Chemistry and Kinetics of Organics in the Atmosphere, GECKO-A (Aumont et al., 2005), and the Master Chemical Mechanism, MCM (Saunders et al., 2003). Application of the methods is illustrated with examples in the supporting information provided in the Supplement.

## 2 Bimolecular reactions of $RO_2$ radicals

### 2.1 The reaction of $RO_2$ with NO

Rate coefficients for the reactions of NO with a variety of specific hydrocarbon and oxygenated $RO_2$ radicals have been reported, as summarized in Table 1. For the vast majority of the $RO_2$ radicals formed in detailed mechanisms, however, kinetic data are unavailable, and it is therefore necessary to assign generic rate coefficients based on the reported data.

For acyl peroxy radicals (i.e. of structure $RC(O)O_2$), a generic rate coefficient ($k_{APNO}$) is applied:

$$k_{APNO} = 7.5 \times 10^{-12} \exp(290/T) \text{ cm}^3 \text{ molecule}^{-1} \text{ s}^{-1} \tag{1}$$

This is based on the IUPAC Task Group[1] recommendation for the reaction of NO with $CH_3C(O)O_2$. As shown in Table 1, this is also close to the rate coefficients recommended for the less studied acyl peroxy radicals, $C_2H_5C(O)O_2$ and $CH_2=CH(CH_3)C(O)O_2$. The 298 K value reported for $CH(OOH)CH_2CH_2CH_2CH(OOH)C(O)O_2$ (Berndt et al., 2015) is also broadly consistent with $k_{APNO}$, although further studies of highly-oxygenated acyl peroxy radicals would help to establish the

20 effects of additional substituent groups.

For other classes of hydrocarbon and oxygenated peroxy radical, a generic rate coefficient ($k_{RO2NO}$) is applied:

$$k_{RO2NO} = 2.7 \times 10^{-12} \exp(360/T) \text{ cm}^3 \text{ molecule}^{-1} \text{ s}^{-1} \tag{2}$$

The value of $k_{RO2NO}$ at 298 K ($9.0 \times 10^{-12}$ cm$^3$ molecule$^{-1}$ s$^{-1}$) is based on a rounded average of the 298 K rate coefficients listed for the $\geq C_2$ alkyl, cycloalkyl, hydroxyalkyl, hydroxyalkenyl, oxoalkyl, hydroxy-oxyalkyl and hydroxy-dioxa-bicyclo $RO_2$

radicals in Table 1, which show no significant trends related to the identity and structure of R. The temperature dependence is similarly based on the rounded average of the available values within this group, which are limited to those for $C_2H_5O_2$, $n$-$C_3H_7O_2$ and $i$-$C_3H_7O_2$. In practice, the preferred values for all the $\geq C_2$ (non-acyl) $RO_2$ radicals in Table 1 are also equivalent to $k_{RO2NO}$ within the reported uncertainties, such that the generic rate coefficient can reasonably be applied for simplicity in all cases except

---

[1] The "IUPAC Task Group on Atmospheric Chemical Kinetic Data Evaluation" is abbreviated to "IUPAC Task Group" for simplicity. The evaluation is available at http://iupac.pole-ether.fr/ (access date January 2019 throughout).





$CH_3O_2$. Although derived from a more extensive dataset, the expression for $k_{RO2NO}$ in Eq. (2) is identical to that recommended previously by Atkinson (1997).

The following channels are considered for the reactions of $RO_2$ with NO:

$$RO_2 + NO \rightarrow RO + NO_2 \tag{R2a}$$

$$RO_2 + NO\ (+M) \rightarrow RONO_2\ (+M) \tag{R2b}$$

It is well established that the branching ratio for alkyl peroxy radicals depends on temperature, pressure, and the size and degree of substitution of the peroxy radical (e.g. Carter and Atkinson, 1989; Arey et al., 2001; Yeh and Ziemann, 2014a). The branching ratio has also been reported to be influenced by the presence of oxygenated substituents, with most systematic information reported for β- and δ- hydroxy groups (e.g. O'Brien et al., 1998; Matsunga and Ziemann, 2009, 2010; Yeh and

Ziemann, 2014b; Teng et al., 2015).

The fraction of the reaction forming a nitrate product ($RONO_2$) via the terminating channel, $R_{2b} = k_{2b}/(k_{2a}+k_{2b})$, is calculated following the method originally reported for secondary alkyl peroxy radicals by Carter and Atkinson (1989), and subsequently updated by Arey et al. (2001). Based on this method, and adopting the terminology used by Orlando and Tyndall (2012), the reference branching ratio for secondary alkyl peroxy radicals, $R^\circ = (k_{2b}/k_{2a})^\circ$ is calculated as follows,

$$R^\circ = [A/(1 + (A/B))]\ F^z \tag{3}$$

with $A = 2 \times 10^{-22} \exp(n_{CON})\ [M]\ (T/300)$, $B = 0.43\ (T/300)^{-8}$, $F = 0.41$, and $z = (1 + (\log_{10}(A/B))^2)^{-1}$. $n_{CON}$ is the number of carbon, oxygen and nitrogen atoms in the organic group of the peroxy radical (equivalent to the carbon number in alkyl peroxy radicals), $T$ is the temperature (in K) and $[M]$ is the gas density (in molecule cm$^{-3}$).

The fractions of the reaction proceeding via the terminating channel, $R_{2b}$, and the propagating channel, $R_{2a}$ (= 1-$R_{2b}$), for a

specific peroxy radical are then given by:

$$R_{2b} = f_a f_b\ (R^\circ/(1+R^\circ)) \tag{4}$$

The effect of the degree of substitution (i.e. whether the radical is primary, secondary or tertiary) is described by $f_a$, with a unity value applied to secondary peroxy radicals, by definition. A further scaling factor, $f_b$, is used to describe systematic variations in the yields of $RONO_2$ resulting from the presence of oxygenated substituents (e.g. the effect of hydroxyl

substituents, as indicated above), or for specific peroxy radical classes, with a value of $f_b$ being required to account for the effect of each relevant substituent. The applied values of $f_a$ and $f_b$ are summarized in Tables 2 and 3, and example calculations are provided in Sect. S1.

It is also recognised that channel (R2a) is significantly exothermic, such that prompt decomposition or isomerization of a fraction of the initially-formed chemically activated oxy radicals has been reported to occur in some cases; with the

remainder being collisionally deactivated to form thermalized RO (e.g. Orlando et al., 2003; Calvert et al., 2015). This is particularly important for β-hydroxy-oxy radicals (e.g. Orlando et al., 1998; Vereecken et al., 1999; Vereecken and Peeters,



1999; Caralp et al., 2003) and some other oxygenated oxy radicals (e.g. Christensen et al., 2000; Orlando et al., 2000; Wallington et al., 2001). The contributions and treatment of these reactions is summarized in Sect. S2.

**2.2 The reaction of RO$_2$ with NO$_2$**

The reactions of RO$_2$ with NO$_2$ have generally been reported to proceed via a reversible association reaction in each case to

form a peroxy nitrate (ROONO$_2$):

$$RO_2 + NO_2\ (+M) \rightleftharpoons ROONO_2\ (+M) \tag{R3a}$$

Rate coefficients for the forward and reverse reactions for a number of RO$_2$ radicals are summarized in Table 4. Those for CH$_3$O$_2$ and C$_2$H$_5$O$_2$, and for the two simplest acyl peroxy radicals, CH$_3$C(O)O$_2$ and C$_2$H$_5$C(O)O$_2$, are based on (or informed by) the IUPAC Task Group recommendations, and describe the pressure and temperature dependences of the reactions. In all other cases,

the reactions are assumed to be at the high pressure limit under atmospheric conditions, and generic parameters are applied. The parameters $k_{f\,PN}$ and $k_{b\,PN}$ (given in Table 4) can reasonably be applied to reactions involving non-acyl peroxy radicals, being based on the high pressure limiting rate coefficients ($k_\infty$) for the forward and reverse reactions of C$_2$H$_5$O$_2$ and those reported for a number of higher alkyl peroxy radicals at close to atmospheric pressure (see Table 4 comments). This assumption is also broadly consistent with the limited information available for the forward or reverse reactions of other non-acyl oxygenated peroxy

radicals (e.g. Orlando and Tyndall, 2012). In practice, however, these reactions are often omitted from atmospheric chemical mechanisms, owing to the instability of the ROONO$_2$ products under lower tropospheric conditions (lifetime ≈ 0.2 s at 298 K). As a result, only the formation and decomposition of methyl peroxy nitrate, CH$_3$OONO$_2$, from the most abundant non-acyl peroxy radical, CH$_3$O$_2$, have previously been represented in the MCM (Saunders et al., 2003). This approach remains advocated here for application to lower tropospheric conditions.

The reactions are generally represented for acyl peroxy radicals, for which the product peroxyacyl nitrates, RC(O)OONO$_2$, are particularly stable (lifetime ≈ 40 − 50 minutes at 298 K). The generic parameters, $k_{f\,PAN}$ and $k_{b\,PAN}$, are applied in the majority of cases (see Table 4). As shown in Fig. 1, larger acyl peroxy radicals have been reported to be slightly more stable than those derived from CH$_3$C(O)O$_2$ and C$_2$H$_5$C(O)O$_2$ (Roberts and Bertman, 1992; Kabir et al., 2014), and the assigned value of $k_{b\,PAN}$ is consistent with the data for the larger species.

Reported data for CH$_3$OC(O)O$_2$, C$_6$H$_5$OC(O)O$_2$ and C$_2$H$_5$OC(O)O$_2$ (Kirchner et al., 1999; Bossolasco et al., 2011) indicate a reduced thermal stability of peroxyacyl nitrates derived from formate esters, and an increased decomposition rate ($2 \times k_{b\,PAN}$) is therefore applied to ROC(O)OONO$_2$ species in general.

In a limited number of cases, the reaction of RO$_2$ with NO$_2$ has been reported to oxidize NO$_2$ to NO$_3$ in an irreversible reaction:

$$RO_2 + NO_2 \rightarrow RO + NO_3 \tag{R3b}$$



These cases include $HC(O)C(O)O_2$ (Orlando and Tyndall, 2001), $CH_3C(O)C(O)O_2$ (Jagiella and Zabel, 2008) and the phenylperoxy radical, $C_6H_5O_2$ (Jagiella and Zabel, 2007). Reaction (R3b) is therefore applied generally to $HC(O)C(O)O_2$, $RC(O)C(O)O_2$, $C_6H_5O_2$ and substituted phenylperoxy radicals, using the generic rate coefficient $k_{fPAN}$.

## 2.3 The reaction of $RO_2$ with $NO_3$

On the basis of reported information for $CH_3O_2$ and $C_2H_5O_2$ (e.g. Biggs et al., 1995; Kukui et al., 1997), the reactions of $RO_2$ with $NO_3$ are assumed to proceed via a single channel in each case, as follows:

$$RO_2 + NO_3 \rightarrow RO + NO_2 + O_2 \tag{R4}$$

Reported rate coefficients are summarised in Table 5. The reaction of $C_2H_5O_2$ with $NO_3$ is the most studied, with consistent 298 K rate coefficients reported in a number of studies (Biggs et al., 1995; Ray et al., 1996; Vaughan et al., 2006; Laversin et

al., 2016) and with the temperature dependence systematically investigated (Laversin et al., 2016). The corresponding parameters in Table 5 therefore form the basis of a generic rate coefficient for the reactions of non-acyl peroxy radicals with $NO_3$:

$$k_{RO2NO3} = 8.9 \times 10^{-12} \exp(-390/T) \text{ cm}^3 \text{ molecule}^{-1} \text{ s}^{-1} \tag{5}$$

Within the reported uncertainties, the value of the rate coefficient at 298 K is consistent with that for $c\text{-}C_6H_{11}O_2$ and with the

approximate value for $c\text{-}C_5H_9O_2$ reported by Vaughan et al. (2006); and the temperature dependence expression for $k_{RO2NO3}$ is consistent with those reported for the oxygenated primary peroxy radicals, $(CH_3)_2C(OH)CH_2O_2$, $CH_3OCH_2O_2$ and $CH_3C(O)CH_2O_2$, by Kalalian et al. (2018). $k_{RO2NO3}$ is therefore currently considered appropriate for application to all $\geq C_2$ non-acyl peroxy radicals. For $CH_3O_2$, the reaction has been well studied at 298 K, and the value in Table 5 is applied in conjunction with the $k_{RO2NO3}$ pre-exponential factor, leading to:

$$k(CH_3O_2 + NO_3) = 8.9 \times 10^{-12} \exp(-600/T) \text{ cm}^3 \text{ molecule}^{-1} \text{ s}^{-1} \tag{6}$$

The generic rate coefficient for acyl peroxy radicals is based on data for $CH_3C(O)O_2$, which has been shown to react slightly more rapidly with $NO_3$ (Canosa-Mas et al., 1996; Doussin et al., 2003). The value at 298 K in Table 5 (based on that reported by Doussin et al., 2003) is once again applied in conjunction with the $k_{RO2NO3}$ pre-exponential factor, leading to:

$$k_{APNO3} = 8.9 \times 10^{-12} \exp(-305/T) \text{ cm}^3 \text{ molecule}^{-1} \text{ s}^{-1} \tag{7}$$

The resultant weak temperature dependence yields a value of $k_{APNO3}$ in the range 403-443 K that is fully consistent with that reported by Canosa-Mas et al. (1996).

## 2.4 The reaction of $RO_2$ with OH

Kinetics determinations have been reported for the reactions of OH with $C_1$-$C_4$ alkyl peroxy radicals. As shown in Table 6, these reactions are reported to occur rapidly at room temperature, with the rate coefficients for all the reactions being





essentially equivalent at 298 K, within the reported uncertainties. Based on the study of Yan et al. (2016), a weak temperature dependence is recommended for the reaction of $CH_3O_2$ with OH, and the resultant expression,

$$k_{RO2OH} = 3.7 \times 10^{-11} \exp(350/T) \text{ cm}^3 \text{ molecule}^{-1} \text{ s}^{-1} \qquad (8)$$

is also adopted in the present work as a generic rate coefficient for the reactions of $RO_2$ with OH.

The following product channels are considered, but with their branching ratios being strongly dependent on the size of R:

$$RO_2 + OH \rightarrow RO + HO_2 \qquad (R5a)$$

$$RO_2 + OH \rightarrow ROH + O_2 \qquad (R5b)$$

$$RO_2 + OH (+M) \rightarrow ROOOH (+M) \qquad (R5c)$$

In their theoretical studies of the reaction of $CH_3O_2$ with OH, Bian et al. (2015), Müller et al. (2016) and Assaf et al. (2018)
calculated channel (R5a) to be the most favourable, with experimental confirmation of a dominant contribution from this channel reported for $CH_3O_2$ by Assaf et al. (2017a; 2018). A number of alternative channels have been considered in modelling assessments (e.g. Archibald et al., 2009), including formation of $CH_2O_2$ and $H_2O$ or $CH_3OH$ and $O_2$. However, no evidence for formation of $CH_2O_2$ and $H_2O$ has been observed at room temperature, indicating that this product channel is at most minor (< 5%) (Yan et al., 2016; Assaf et al., 2017a; Caravan et al., 2018). The formation of $CH_3OH$ and $O_2$ via channel
(R5b) has been shown to make a minor contribution (6 – 9 %) in the experimental study of Caravan et al. (2018), consistent with the theoretical estimate of ∼ 7 % by Müller et al. (2016). As a result, values of $k_{5a}/k_5 = 0.93$ and $k_{5b}/k_5 = 0.07$ are assigned to the reaction of $CH_3O_2$ with OH in the present work.

The experimental and theoretical study of Assaf et al. (2018) for a series of $C_1$-$C_4$ alkyl peroxy radicals has demonstrated that the reaction can more generally be regarded as proceeding by either channel (R5a) or (R5c). Formation of the
thermalized hydrotrioxide, ROOOH, via channel (R5c) was found to be increasingly important for the larger $RO_2$. Based approximately on their theoretical calculations for 298 K and 1 atmosphere pressure, $k_{5c}/k_5$ is thus currently assigned a value of 0.0 for $CH_3O_2$, 0.8 for $RO_2$ for which $n_{CON} = 2$ (e.g. $C_2H_5O_2$ and $HOCH_2O_2$) and 1.0 for all other $RO_2$ radicals. In the $n_{CON} = 2$ case, the balance of the reaction is assigned to channel (R5a), i.e. with $k_{5b}/k_5 = 0$. As discussed by Assaf et al. (2018), detailed experimental and theoretical studies of the atmospheric fate of ROOOH are therefore clearly required for the
effect of the $RO_2$ + OH reaction to be fully assessed and represented. A provisional treatment is provided in Sect. S3, based mainly on rate coefficients reported in the theoretical studies of Müller et al. (2016), Assaf et al. (2018) and Anglada and Solé (2018).



## 2.5 The reaction of RO$_2$ with HO$_2$

Rate coefficients for the reactions of HO$_2$ with a variety of specific hydrocarbon and oxygenated RO$_2$ radicals have been reported, as summarized in Table 7. For the vast majority of the RO$_2$ radicals formed in detailed mechanisms, however, kinetic data are unavailable, and it is therefore necessary to assign generic rate coefficients based on the reported data.

As discussed previously (Jenkin et al., 1997; Saunders et al., 2003; Boyd et al., 2003a; Orlando and Tyndall, 2012), the 298 K rate coefficients tend to increase with the size of the organic group. Fig. 2 shows the data plotted as a function of $n_{CON}$. The data for alkyl peroxy radicals and β-hydroxyalkyl peroxy radicals (the most systematically studied groups) show comparable values across the $n_{CON}$ range. Based on optimization to these data, the following expression is derived for application to non-acyl peroxy radicals:

$$k_{RO2HO2} = 2.8 \times 10^{-13} \exp(1300/T) \; [1\text{-}\exp(-0.23n_{CON})] \; cm^3 \; molecule^{-1} \; s^{-1} \tag{9}$$

The temperature dependence is typical of that reported for > C$_2$ alkyl and β-hydroxy RO$_2$ radicals, and remains unchanged from that applied previously by Saunders et al. (2003).

Based on the limited data for acyl peroxy radicals (see Fig. 2 and Table 7), the 298 K rate coefficients are assigned values that are a factor of two greater than those defined by Eq. (9). The temperature dependences reported for acyl peroxy radicals

appear to be weaker than those for similar sized radicals in other classes, and the temperature coefficient is based on that recommended for CH$_3$C(O)O$_2$. The following expression is therefore assigned to acyl peroxy radicals:

$$k_{APHO2} = 6.3 \times 10^{-12} \exp(580/T) \; [1\text{-}\exp(-0.23n_{CON})] \; cm^3 \; molecule^{-1} \; s^{-1} \tag{10}$$

On the basis of reported information, the following channels are considered for the reactions of RO$_2$ with HO$_2$:

$$RO_2 + HO_2 \quad \rightarrow ROOH + O_2 \tag{R6a}$$

$$\rightarrow ROH + O_3 \tag{R6b}$$

$$\rightarrow R_{-H}O + H_2O + O_2 \tag{R6c}$$

$$\rightarrow RO + OH + O_2 \tag{R6d}$$

$$\rightarrow R_{-H} + OH + HO_2 \tag{R6e}$$

Formation of a hydroperoxide product (ROOH) and O$_2$ via terminating channel (R6a) is reported to be dominant for

reactions of alkyl peroxy radicals, and this is also taken to be the default where no information is available. However, the reactions of HO$_2$ with oxygenated peroxy radicals have received considerable attention, and evidence has been reported for several additional channels leading to both radical termination, (R6b) and (R6c), and radical propagation, (R6d) and (R6e). Table 8 summarizes the 298 K branching ratios that are applied to several classes of oxygenated peroxy radical, based on reported information.





The temperature-dependences of the reaction channels have generally not been studied, and the branching ratios in Table 8 are thus applied independent of temperature in most cases. The only exception is the reaction of $HO_2$ with (non-phenyl) acyl peroxy radicals. This class of reaction (in particular the reaction of $HO_2$ with $CH_3C(O)O_2$) has received the most attention, and is also a class for which radical propagation is reported to be particularly important at temperatures near 298 K. As

shown in Table 8, channels (R6a), (R6b) and (R6d) are reported to contribute. The temperature dependence of $k_{6a}/k_{6b}$ is defined using the experimental characterization of the $CH_3C(O)O_2 + HO_2$ reaction reported by Horie and Moortgat (1992), with $k_{6d}/k_{6b}$ provisionally based on the results of the theoretical calculations of Hasson et al. (2005) for the same reaction between 250 K and 300 K at atmospheric pressure:

$$k_{6a}/k_{6b} \, (RC(O)O_2) = 3.4 \times 10^2 \exp(-1430/T) \tag{11}$$

$$k_{6d}/k_{6b} \, (RC(O)O_2) = 2.34 \times 10^4 \exp(-2600/T) \tag{12}$$

The corresponding temperature dependences of the channel branching ratios ($k_{6a}/k_6$, $k_{6b}/k_6$ and $k_{6d}/k_6$), derived from $CH_3C(O)O_2$ data, are applied to all (non-phenyl) acyl peroxy radicals. The variation of the branching ratios is illustrated in Fig. S2, for the 250-300 K temperature range. The temperature dependence of OH formation, described by $k_{6d}/k_6$, was used to correct the temperature coefficient applied to the overall reaction kinetics, because the effect of the resultant reagent

radical regeneration was not taken into account in reported studies of the temperature dependence of the $CH_3C(O)O_2 + HO_2$ reaction (see Sect. S4 for further details).

**2.6 The permutation reactions of $RO_2$**

The "permutation" reactions of a given $RO_2$ radical are its self-reaction (R7), and its cross-reactions (R8) with other peroxy radicals, $R'O_2$, for which a number of product channels may occur:

$RO_2 + RO_2$  $\rightarrow RO + RO + O_2$                        (R7a)

       $\rightarrow R_{-H}O + ROH + O_2$                     (R7b)

       $\rightarrow ROOR + O_2$                        (R7c)

$RO_2 + R'O_2$ $\rightarrow RO + R'O + O_2$                      (R8a)

       $\rightarrow R_{-H}O + R'OH + O_2$                    (R8b)

$\rightarrow R'_{-H}O + ROH + O_2$                    (R8c)

       $\rightarrow ROOR' + O_2$                       (R8d)

In view of the large number of $RO_2$ radicals generated in a detailed chemical mechanism, however, it is unrealistic to represent these reactions explicitly, and the use of simplified parameterizations is essential (Madronich and Calvert, 1990).





As described in detail previously (Jenkin et al., 1997), a very simplified approach has traditionally been adopted in the MCM, in which each peroxy radical is assumed to react with all other peroxy radicals (i.e. the peroxy radical "pool") at a single, collective rate. This is achieved by defining a parameter "$\sum[RO_2]$" which is the sum of the concentrations of all peroxy radicals, excluding $HO_2$. The collective rate of all the permutation reactions of a particular peroxy radical is then

represented by a single pseudo-unimolecular reaction, which has an assigned rate coefficient equal to $k_9 \times \sum[RO_2]$,

$$RO_2 \rightarrow products \qquad\qquad (R9)$$

with the value of $k_9$ depending on the structure of the reacting $RO_2$ radical. A similar, but more detailed, approach has been applied in GECKO-A, in which the peroxy radical population is divided into a number of reactivity classes (Aumont et al., 2005). This requires the inclusion of a pseudo-unimolecular reaction (analogous to reaction (R9)) for reaction of a given peroxy

radical with each peroxy radical class, but has the advantage that differential reactivity with each of those classes can be represented, as appropriate. The following paragraphs describe how rate parameters are assigned to the single parameterized permutation reactions (reaction (R9)) for each peroxy radical in the more simplified MCM approach. Extension of the method to reactions with a number of reactivity classes (as traditionally applied with GECKO-A) is described in Sect. S5.

Rate coefficients for the self-reactions and cross-reactions of a variety of specific hydrocarbon and oxygenated $RO_2$ radicals have

15 been reported (as summarized in Tables 9-11), and these form the basis of assigning rate parameters to the parameterized permutation reaction (reaction (R9)) for each peroxy radical. The data show that the self-reaction reactivity, relative to that of alkyl peroxy radicals, is activated by the presence of numerous functional groups (including allyl-, benzyl-, hydroxy-, alkoxy-, oxo- and acyl-), and that the rate coefficients follow the general trend of decreasing reactivity, primary > secondary > tertiary, for peroxy radicals containing otherwise similar functionalities. It also appears that reactivity tends to increase with the size of the

20 organic group towards a "plateau" value, as most clearly demonstrated by the systematic study of secondary alkyl peroxy radicals reported by Boyd et al. (1999). Based on optimization to the complete secondary alkyl peroxy radical dataset, an expression almost identical to that recommended by Boyd et al. (1999) is thus derived as a reference rate coefficient for secondary peroxy radicals at 298 K, as illustrated in Fig. 3 (units of $k$ are $cm^3$ molecule$^{-1}$ s$^{-1}$) :

$$\log_{10}(k^{\circ}_{RO2RO2(sec)}) = -12.9-(3.2 \times \exp[-0.64(n_{CON}-2.3)]) \qquad\qquad (13)$$

The data for primary alkyl peroxy radicals are more limited. Those for $C_2H_5O_2$, $n$-$C_3H_7O_2$, $i$-$C_4H_9O_2$ and $neo$-$C_5H_{11}O_2$ suggest a similar trend for primary alkyl peroxy radicals, and an analogous expression to Eq. (13) is therefore derived as a reference rate coefficient at 298 K (see Fig. 3):

$$\log_{10}(k^{\circ}_{RO2RO2(prim)}) = -11.7-(3.2 \times \exp[-0.55(n_{CON}-0.52)]) \qquad\qquad (14)$$

It is noted, however, that rate coefficients for the self-reactions of $n$-$C_4H_9O_2$ and $n$-$C_5H_{11}O_2$ are reported to be comparable to that

of $n$-$C_3H_7O_2$, and a factor of two to three lower than those for $i$-$C_4H_9O_2$ and $neo$-$C_5H_{11}O_2$ (see Table 9), suggesting that there may be sensitivity to whether the alkyl group is linear or branched. In the absence of additional data (and noting that the kinetics of




*neo*-$C_5H_{11}O_2$ were the most directly determined of the set of $C_4$ and $C_5$ primary alkyl peroxy radicals), the above (stronger) size dependence is provisionally applied here.

Data for tertiary alkyl peroxy radicals are currently limited to *t*-$C_4H_9O_2$, and the corresponding rate coefficient is currently applied as the reference rate coefficient at 298 K, independent of radical size (see Fig. 3):

$$k°_{RO2RO2(tert)} = 2.1 \times 10^{-17} \text{ cm}^3 \text{ molecule}^{-1} \text{ s}^{-1} \tag{15}$$

Fig. 3 also shows data for allyl and β-hydroxyalkyl $RO_2$, demonstrating that the presence of both these functionalities has an activating effect on self-reaction reactivity. The allyl peroxy radical category includes two δ-hydroxyallyl peroxy radicals, and the assumption is made here that the δ-hydroxy group is too remote to have an influence. Table 12 summarizes a series of activation factors (defined in terms of the parameters α and β) for allyl-, benzyl-, hydroxy-, alkoxy- and oxo- groups, optimized on the basis of the data in Tables 9 and 10. These are used in conjunction with the reference rate coefficients in Eq. (13)-(15), to calculate the self-reaction rate coefficient for a given peroxy radical at 298 K, $k_{RO2RO2}$, as follows:

$$k_{RO2RO2} = k°_{RO2RO2} \times \alpha/(k°_{RO2RO2})^\beta = \alpha \times (k°_{RO2RO2})^{1-\beta} \tag{16}$$

Here, $k°_{RO2RO2}$ represents the appropriate reference rate coefficient (i.e. for primary, secondary or tertiary $RO_2$, as appropriate) as defined by Eq. (13)-(15); and the term $\alpha/(k°_{RO2RO2})^\beta$ describes the level of activation from the given substituent. The inclusion of $k°_{RO2RO2}$ within this activation term is required because the relative enhancement of reactivity resulting from a given substituent appears to decrease as the reactivity increases, as illustrated for the β-hydroxyalkyl group data in Fig. 3. Based on this method, the estimated rate coefficients correlate well with those observed for the series of peroxy radicals for which data are currently available (summarized in Tables 9 and 10), as shown in Fig. 4. It is emphasized, however, that the parameters for several of the substituent groups are based on data for very limited sets of peroxy radicals, and additional data would be valuable to test and constrain the method.

Information on the effects of multiple substituents is limited to the data for the secondary and tertiary β-hydroxyallyl peroxy radicals, $HOCH_2CH(O_2)CH=CH_2$ and $HOCH_2C(CH_3)(O_2)C(CH_3)=CH_2$, given in Table 10. The reported rate coefficients are consistent with the activating impacts of the β-hydroxy and allyl substituents being approximately cumulative, suggesting that an activation factor should be applied for each relevant organic substituent. However, this would lead to unreasonably large estimated values of $k_{RO2RO2}$ for secondary and tertiary peroxy radicals containing two or three of the most activating substituents, such that the impact needs to be limited. In multifunctional peroxy radicals, therefore, an activating factor is only applied for the most activating oxygenated substituent in a given peroxy radical, with an additional factor also applied only for the specific cases of an allyl or a benzyl substituent, again limited to one (i.e. the most activating) factor if the peroxy radical contains more than one allyl or benzyl group. In these specific cases, therefore, $\alpha = (\alpha_1 \times \alpha_2)$ and $\beta = (\beta_1 + \beta_2)$, where $\alpha_1$ and $\beta_1$ refer to the oxygenated substituent, and $\alpha_2$ and $\beta_2$ refer to either the allyl substituent or the benzyl substituent. Further information is required to allow the impacts of multiple substituents to be defined more rigorously.





The rate coefficients for cross-reactions of peroxy radicals (reaction (R8)) have often been inferred from those for the self-reactions of the participating peroxy radicals, using a geometric mean rule as first suggested by Madronich and Calvert (1990). Fig. 5 shows that such a correlation provides a reasonable guide in many cases (although a clear deviation from the rule occurs for the particular case of reactions involving acyl peroxy radicals). In the very simplified MCM approach, the rate coefficient for the

single parameterized permutation reaction of a given peroxy radical (reaction (R9)) is based on that estimated for the cross-reaction of the peroxy radical with $CH_3O_2$. This is regarded as a logical choice, because $CH_3O_2$ is the most abundant organic peroxy radical in the atmosphere (and therefore most commonly the major reaction partner), and also possesses a self-reaction rate coefficient that is in the middle of the range of reported values (see Tables 9 and 10). Taking account of the correlations in Fig. 5, the rate coefficients (in $cm^3$ molecule$^{-1}$ s$^{-1}$) for the parameterized permutation reactions at 298 K are defined as follows:

For acyl RO$_2$: $\qquad\qquad\qquad\qquad k_{AP(298K)} = 1.1 \times 10^{-11}$ $\qquad\qquad\qquad\qquad\qquad\qquad\qquad$ (17)

For other RO$_2$ (except CH$_3$O$_2$): $\qquad k_{RO2(298K)} = f_{RO2} \times 2 \times (k_{RO2RO2} \times k_{298}(CH_3O_2+CH_3O_2))^{0.5}$ $\qquad\qquad$ (18)

Here, $k_{298}(CH_3O_2+CH_3O_2)$ is the rate coefficient for the self-reaction of $CH_3O_2$ at 298 K (= $3.5 \times 10^{-13}$ cm$^3$ molecule$^{-1}$ s$^{-1}$) and $k_{RO2RO2}$ is the 298 K self-reaction rate coefficient, estimated as described above. $f_{RO2}$ is a scaling factor that is introduced to describe systematic deviations from the geometric mean rule, if required. Based on the correlations in Fig. 5, a unity value of

$f_{RO2}$ is acceptable for primary and secondary peroxy radicals (i.e. no deviation from the geometric mean rule), whereas a value of $f_{RO2} = 2$ is applied to tertiary peroxy radicals. This elevated scaling factor is based on observation of Jenkin et al. (1998) for complex tertiary RO$_2$ cross reactions.

Based on the reported temperature dependences of peroxy radical self- and cross-reactions (see Tables 9 and 10, and Table 11 comments), $k_{AP}$ and $k_{RO2}$ are assigned respective pre-exponential factors of $2.0 \times 10^{-12}$ and $1.0 \times 10^{-13}$ cm$^3$ molecule$^{-1}$ s$^{-1}$. For acyl

peroxy radicals, this is consistent with the temperature dependence reported for the reaction of $CH_3C(O)O_2$ with $CH_3O_2$, and results in the following temperature dependent expression in all cases:

$k_{AP} = 2.0 \times 10^{-12} \exp(508/T)$ cm$^3$ molecule$^{-1}$ s$^{-1}$ $\qquad\qquad\qquad\qquad\qquad\qquad\qquad\qquad\qquad\qquad$ (19)

For $k_{RO2}$, the pre-exponential factor is a rounded value, based on the geometric mean of those for the self-reactions of non-acyl peroxy radicals given in Tables 9 and 10. This results in the following temperature dependence expression for non-acyl peroxy

radicals (except CH$_3$O$_2$),

$k_{RO2} = 1.0 \times 10^{-13} \exp(-(E_{RO2}/R)/T)$ cm$^3$ molecule$^{-1}$ s$^{-1}$ $\qquad\qquad\qquad\qquad\qquad\qquad\qquad\qquad\qquad$ (20)

with $E_{RO2}/R$ having a case dependent value of -298 x $\ln(k_{RO2(298\ K)}/10^{-13})$, where $k_{RO2(298\ K)}$ is defined by Eq. (18). Examples of specific rate coefficients estimated using this method are given in Sect. S5, for the peroxy radicals formed from the sequential addition of OH and O$_2$ to isoprene. As indicated above, the collective rate of all the permutation reactions of a

particular peroxy radical is then represented by a pseudo-unimolecular reaction (reaction (R9)), which has an assigned rate coefficient equal to $k_{AP} \times \Sigma[RO_2]$ for acyl peroxy radicals, and $k_{RO2} \times \Sigma[RO_2]$ for all other peroxy radicals (except CH$_3$O$_2$).



For the specific case of CH$_3$O$_2$, the applied rate coefficient ($k_{CH3O2}$) is twice the self-reaction rate coefficient given in Table 9,

$k_{CH3O2} = 2.06 \times 10^{-13} \exp(365/T)$ cm$^3$ molecule$^{-1}$ s$^{-1}$          (21)

with the pseudo-unimolecular reaction rate coefficient equal to $k_{CH3O2} \times \sum[RO_2]$. This representation is therefore consistent with CH$_3$O$_2$ being lost via its self-reaction with the recommended rate coefficient when it is the dominant radical.

Each reaction potentially has up to four product channels, the branching ratios of which depend on the structure of the radical, as shown in Table 13:

RO$_2$    → RO                                                     (R9a)

        → R$_{-H}$O                                            (R9b)

        → ROH                                               (R9c)

[→ RO$_{(peroxide)}$                                      (R9d)]

Channels (R9a)-(R9c) have been considered previously in the MCM (Jenkin et al., 1997; Saunders et al., 2003). They are the pseudo-unimolecular representation of the self-reaction channels (R7a) and (R7b) and the cross-reaction channels (R8a)-(R8c), which are reported to account for most of the reaction, particularly for smaller peroxy radicals (e.g. Lightfoot et al., 1992; Orlando and Tyndall, 2012). As shown in Table 13, channels (R9a)-(R9c) continue to represent the complete reaction

in the current methodology.

Although not currently included, channel (R9d) is listed to acknowledge the potential formation of peroxide products (i.e. reactions (R7c) and (R8d)). Although these channels have generally been reported to be minor for small peroxy radicals (e.g. Lightfoot et al., 1992; Orlando and Tyndall, 2012), recent studies suggest that they may be more significant for larger peroxy radicals containing oxygenated substituents, and they have been reported to play a role in the formation of low volatility

products in a number of studies (Ziemann, 2002; Ehn et al., 2014; Jokinen et al., 2014; Mentel et al., 2015; Rissanen et al., 2015; Berndt et al., 2015; 2018a; 2018b; Zhang et al., 2015; McFiggans et al., 2019). These reactions may therefore play a potentially important role in particle formation and growth in the atmosphere. The product denoted "RO$_{(peroxide)}$" in the pseudo-unimolecular approach represents the monomeric contribution the given peroxy radical makes to the total formation of (dimeric) peroxide products, but is not an independent species for which subsequent gas phase chemistry can be

rigorously defined. In principle, channel (R9d) can be included for the permutation reactions of a subset of larger peroxy radicals, with the RO$_{(peroxide)}$ product assumed to transfer completely to the condensed phase (i.e. not participating in gas phase reactions). However, there is currently insufficient information on the contributions of channels (R7c) or (R8d) to the overall self- and cross-reactions to allow the branching ratio of channel (R9d) to be defined reliably. Further systematic studies of these channel contributions are therefore required as a function of peroxy radical size and functional group

content.



## 3 Unimolecular reactions of RO$_2$ radicals

Unimolecular isomerization reactions are potentially available for some classes of RO$_2$. These generally fall into the category of either ring-closure reactions (where the peroxy radical adds intra-molecularly to an unsaturated linkage to form a peroxide-bridged radical product); or reactions involving the migration of a hydrogen atom to the peroxy radical group (e.g.

forming a hydroperoxy-substituted organic radical product when abstraction from a C-H bond occurs). For some RO$_2$ structures, these reactions have been shown to compete with (or dominate over) the bimolecular reactions under some atmospheric conditions, as discussed further below in Sects. 3.1 and 3.2. Evidence for the operation of peroxy radical isomerization reactions has been reported in numerous theoretical and laboratory studies (e.g. Vereecken and Peeters, 2004; Peeters et al., 2009; 2014; Crounse et al., 2013; Ehn et al., 2014; 2017; Jokinen et al., 2014; Rissanen et al., 2015; Jørgensen

et al., 2016), and new information is constantly emerging on this important aspect of peroxy radical chemistry. The present section provides a summary of selected classes of isomerization reaction that are currently being considered and represented in ongoing mechanism development work. However, it does not attempt to provide a full treatment of unimolecular reactions of RO$_2$ radicals, which will be considered further in future work as new information becomes available.

### 3.1 Ring-closure reactions of RO$_2$

Table 14 shows the rate coefficients assigned to selected template ring-closure reactions. The first entry relates to the β-hydroxy cyclohexadienylperoxy radicals formed from the addition of O$_2$ to OH-aromatic hydrocarbon adducts. As discussed in the companion paper on the OH-initiated oxidation of aromatic VOCs (Jenkin et al., 2018b), these peroxy radicals are represented to undergo rapid and exclusive ring closure to produce a hydroxy-dioxa-bicyclo or "peroxide-bicyclic" radical. This reaction has been calculated to dominate over alternative bimolecular reactions of the peroxy radicals under atmospheric

conditions (see Table 14), although evidence for competitive loss via bimolecular reactions has been characterized in experimental studies using high concentrations of NO and/or RO$_2$ (e.g. Birdsall et al., 2010; Birdsall and Elrod, 2011).

The remaining reactions in Table 14 are based on information presented by Vereecken and Peeters (2004) for specific peroxy radicals formed from the sequential addition of OH and O$_2$ to isoprene, α-pinene and β-pinene. That information has been used to assign or infer representative rate coefficients to the series of related template peroxy radical structures presented in Table 14. In

these cases, the reactions are expected to occur at rates that can compete to varying extents with loss via bimolecular reactions (or other unimolecular reactions discussed below) under atmospheric conditions.

### 3.2 Hydrogen atom migration reactions of RO$_2$

Table 15 shows selected hydrogen atom migration reactions that are currently considered. The rate coefficient assigned generally to the 1,4 formyl H-shift reaction of α-formyl peroxy radicals is based on that determined for the methacrolein-

derived peroxy radical, HOCH$_2$C(CH$_3$)(O$_2$)C(=O)H, in the experimental study of Crounse et al. (2012). It is noted that this is





slightly higher than, but comparable with, the range of values reported for α-formyl peroxy radicals in the preliminary calculations of Peeters and Nguyen (2012).

The rate coefficients assigned to the 1,4 hydroxyl H-shift reactions of (thermalized) α-hydroxy peroxy radicals are based on those estimated for secondary, tertiary and cyclic peroxy radicals in the theoretical study of Hermans et al. (2005). As

discussed in the companion paper on the OH-initiated oxidation of aliphatic VOCs (Jenkin et al., 2018a), thermalized α-hydroxy peroxy radicals are represented to be increasingly formed from the reactions of $O_2$ with larger α-hydroxy organic radicals (i.e. those with $n_{CON} > 5$). At the assigned rates, the 1,4 hydroxyl H-shift reaction is likely to be the major fate of the majority of thermalized α-hydroxy peroxy radicals under atmospheric conditions, and therefore indistinguishable from that of the chemically activated α-hydroxy peroxy radical adducts that are formed predominantly from the reactions of $O_2$ with

small α-hydroxy organic radicals (see Sect. 6.2 of Jenkin et al., 2018a). However, the rates of the 1,4 hydroxyl H-shift reactions are formalized in the present work, to allow for the representation of competing rapid isomerization reactions for specific structurally-complex peroxy radicals (e.g. the 1,6 enol H-shift reaction discussed below), or with bimolecular reactions under appropriate conditions. It is noted that evidence for competitive loss via bimolecular reactions has been characterized in experimental studies using high concentrations of NO (e.g. Orlando et al., 2000; Jenkin et al., 2005; Aschmann

et al., 2010), leading to the formation of organic acids.

The remaining reactions in Table 15 are inferred from information reported for specific unsaturated peroxy radicals formed during the OH-initiated oxidation of isoprene, taking particular account of the work of Peeters et al. (2009; 2014) on the Leuven isoprene mechanism (LIM1), which has been largely verified by experimental study (e.g. Wennberg et al., 2018; and references therein). The rate coefficients for the 1,5 hydroxyl H-shift reactions are those reported by Peeters et al. (2014) for the

corresponding unsaturated secondary and tertiary β-hydroxy peroxy radicals formed from the sequential addition of OH and $O_2$ to isoprene, with these also being generally consistent with those reported by da Silva et al. (2010). The rate coefficient assigned to the 1,6 hydroxyalkyl H-shift reaction is the geometric mean of rate coefficients applied to (Z)-$CH_2(OH)C(CH_3)=CHCH_2O_2$ (CISOPAO2) and (Z)-$CH_2(OH)CH=C(CH_3)CH_2O_2$ (CISOPCO2) in MCM v3.3.1. As discussed by Jenkin et al. (2015), those rate coefficients are derived from the LIM1 calculations of Peeters et al. (2014), but with some

scaling to recreate the observations of Crounse et al. (2011; 2014). The generic rate coefficient is applied generally to unsaturated δ-hydroxy peroxy radicals containing the sub-structure shown, but with the exceptions of CISOPAO2 and CISOPCO2 themselves, for which the species-specific rate coefficients are applied (see Sect. S6 and Table S5). Similarly, the rate coefficient for the rapid 1,6 enol H-shift reaction is the geometric mean of those calculated for (Z)-$HOCH=C(CH_3)CH(O_2)CH_2OH$ and (Z)-$HOCH=CHC(CH_3)(O_2)CH_2OH$ by Peeters and Nguyen (2012). Once again, the 1,6

enol H-shift reaction is likely to be the major fate of the majority of peroxy radicals containing the relevant sub-structure (see Table 15) under atmospheric conditions, but the rate is formalized in the present work, to allow for the representation of competing rapid isomerization reactions for specific structurally-complex peroxy radicals, e.g. the 1,4 hydroxyl H-shift reaction discussed above, or other reactions that may be considered and represented in future work.





As indicated above, the present paper does not attempt to provide a full treatment of unimolecular reactions of $RO_2$ radicals, which ideally requires systematic information on the rates of a series of 1,$n$ H-shift reactions from C-H and O-H bonds in different environments. In this respect, it is noted that the systematic influence of a series of neighbouring functional groups and transition state sizes have been considered in theoretical studies of some model systems (e.g. Crounse et al., 2013; Jørgensen et al., 2016), and further studies of this type would help to define structure-activity methods for a wider range of $RO_2$ radicals and their potential isomerization reactions. A further consideration, highlighted in those studies, is that the rates of the reverse isomerization reactions are sometimes sufficiently rapid that the product radical may not be fully trapped by onward reaction (e.g. addition of $O_2$) under atmospheric conditions. It is noted that the explicit representation of a very large number of rapid reversible reactions in detailed mechanisms can have implications for computational efficiency.

## 4 Conclusions

Published kinetics and branching ratio data have been reviewed for the bimolecular reactions of organic peroxy radicals ($RO_2$), with information for selected unimolecular isomerization reactions also summarized and discussed. The information has been used to define generic rate coefficients and structure-activity relationship (SAR) methods for the reactions of a series of important classes of hydrocarbon and oxygenated $RO_2$ radical, for application in the next generation of explicit detailed chemical mechanisms, based on GECKO-A and the MCM.

The availability of kinetic and mechanistic data for peroxy radical reactions has increased substantially since the appraisals of Saunders et al. (2003) and Aumont et al. (2005), on which the previous treatments of peroxy radical chemistry in the MCM and GECKO-A were mainly based. These advances have allowed improved and updated methods to be defined and summarized in the present work for an extended set of peroxy radical reactions. Nevertheless, there are still a number of specific areas (commented on in Sects. 2 and 3) where information is lacking and further studies would be beneficial. These include the following:

- Kinetics studies of the reactions with NO have only been reported for a limited number of acyl peroxy radicals. Further studies, particularly for larger and highly-oxygenated acyl peroxy radicals, would help to establish whether size and/or the presence of additional substituent groups has an effect on reactivity.

- For the reactions with $NO_3$, studies for $\geq C_2$ (non-acyl) $RO_2$ are dominated by primary peroxy radicals. Further studies are therefore required for secondary and tertiary radicals, and product information is generally required for a variety of peroxy radical classes to test assumption that the reaction proceeds via a single channel forming RO, $NO_2$ and $O_2$.

- The reactions of $\geq C_2$ hydrocarbon $RO_2$ with OH are believed to produce a thermalized hydrotrioxide, ROOOH, as the major product. Detailed experimental and theoretical studies are therefore required to establish the atmospheric fate of these ROOOH species. Studies of the reactions of oxygenated $RO_2$ with OH are also required.





- The reactions of $HO_2$ with several oxygenated $RO_2$ classes have been shown to proceed via multiple channels, although the temperature-dependences of the product channels have generally not been studied. Additional studies of their temperature dependences would therefore be valuable, in addition to information for larger sets of oxygenated $RO_2$ within some classes. Kinetics studies have only been reported for a limited number of acyl peroxy radicals. Further studies, particularly for larger and highly-oxygenated acyl peroxy radicals, would help to establish whether size and/or the presence of additional substituent groups has an effect on reactivity.

- For the self- and cross-reactions of peroxy radicals, further information is required to allow the impacts of multiple substituents on the kinetics to be defined more rigorously. Further systematic studies of the formation of $ROOR + O_2$ (from the self-reaction of $RO_2$) and $ROOR' + O_2$ (from the cross-reaction of $RO_2$ with $R'O_2$) are also required as a function of peroxy radical size and functional group content.

- For unimolecular isomerization reactions, further systematic studies are required of the rates of $1,n$ H-shift reactions from C-H and O-H bonds in different chemical environments, and of the effect of ring size and substituents on ring-closure reactions.

## Acknowledgements

This work was performed as part of the MAGNIFY project, with funding from the UK Natural Environment Research Council (NERC) via grant NE/M013448/1, and the French National Research Agency (ANR) under project ANR-14-CE01-0010. It was also partially funded by the European Commission through EUROCHAMP-2020 (grant number 730997).

*Author contributions*. All authors defined the scope of the work. MEJ developed and revised the estimation methods and drafted the manuscript, which were reviewed and refined by all authors.

*Competing interests*. The authors declare that they have no conflict of interest.

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

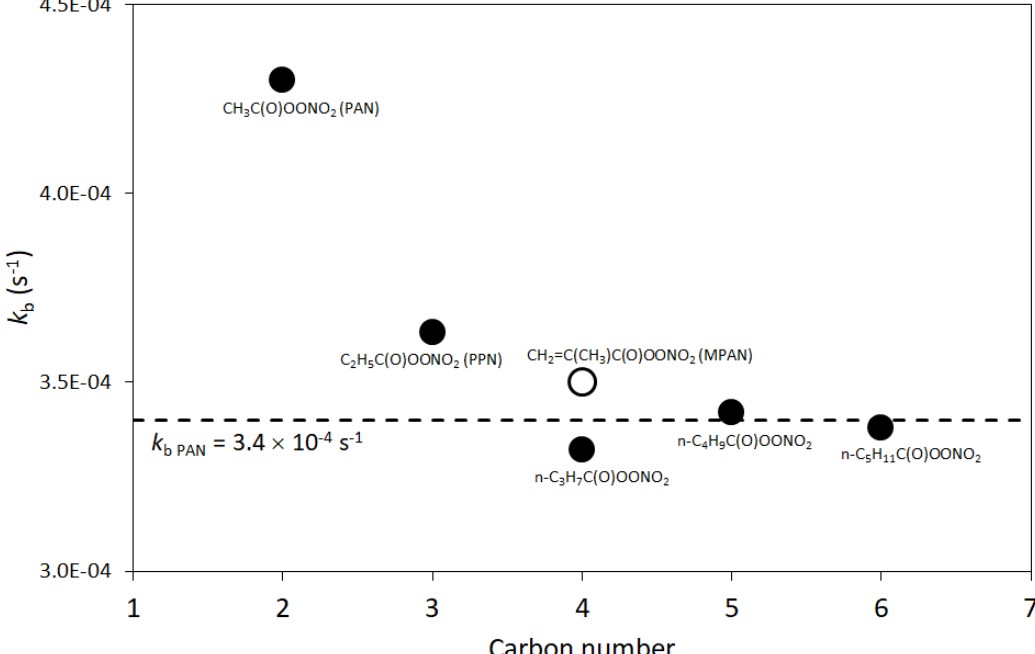

**Figure 1: Reported thermal decomposition rates of selected peroxyacyl nitrates at 298 K and 760 Torr. Values for PAN, PPN and MPAN are the IUPAC Task Group recommendations (http://iupac.pole-ether.fr/). The other values are taken from the systematic study of Kabir et al. (2014), which also reports consistent values for PAN and PPN. The broken line is the generic rate coefficient, $k_{b\ PAN}$, for the decomposition of $RC(O)OONO_2$ structures (see Sect. 2.2 and Table 4).**




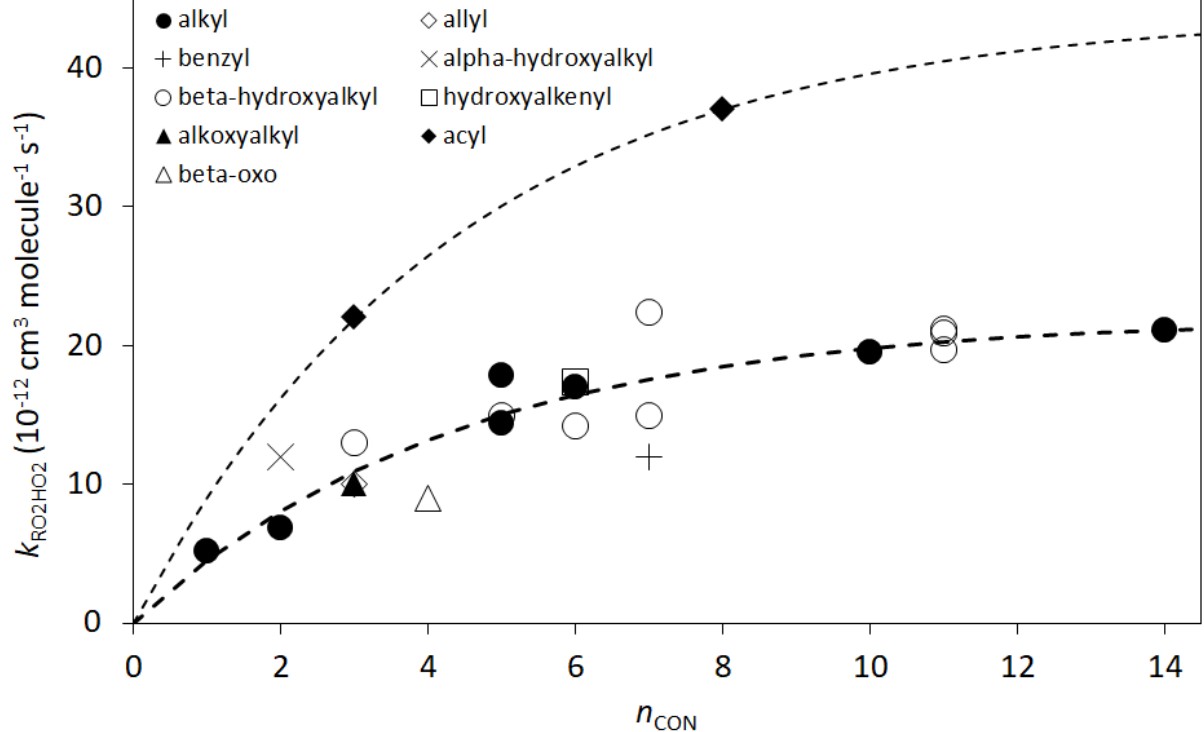

**Figure 2: Rate coefficients for the reactions of various classes of RO$_2$ radicals with HO$_2$ as a function of $n_{CON}$ at 298 K. The heavy broken line is the best fit to the data for alkyl and β-hydroxyalkyl RO$_2$ on the basis of the assumed function $k = A(1−\exp(B.n_{CON}))$. The light broken line is the same function with the value of $A$ increased by a factor of two (see Sect. 2.5).**



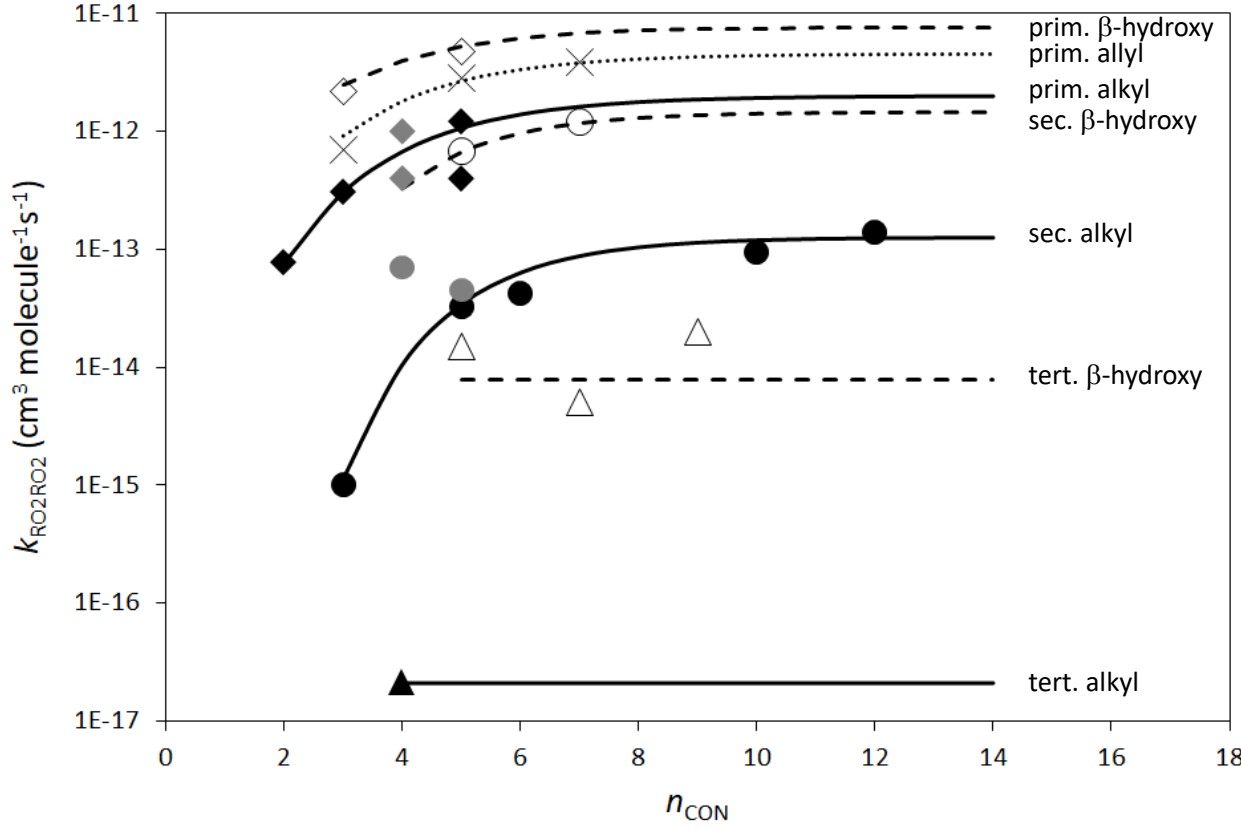

**Figure 3: Rate coefficients for the self-reactions of alkyl (filled points), β-hydroxyalkyl (open points) and allyl (×) RO$_2$ at 298 K as a function of $n_{CON}$. Grey filled points indicate where the reported rate coefficient has not been corrected for secondary chemistry. Where available, data are shown for primary, secondary and tertiary radicals containing the given functionalities. Primary, secondary and tertiary alkyl and β-hydroxyalkyl radicals are shown as diamonds, circles and triangles, respectively. The "allyl" peroxy radical group contains only primary radicals and includes "δ-hydroxyallyl" peroxy radicals. The lines represent the calculated rate coefficients fitted to the data using the methods described in Sect. 2.6.**



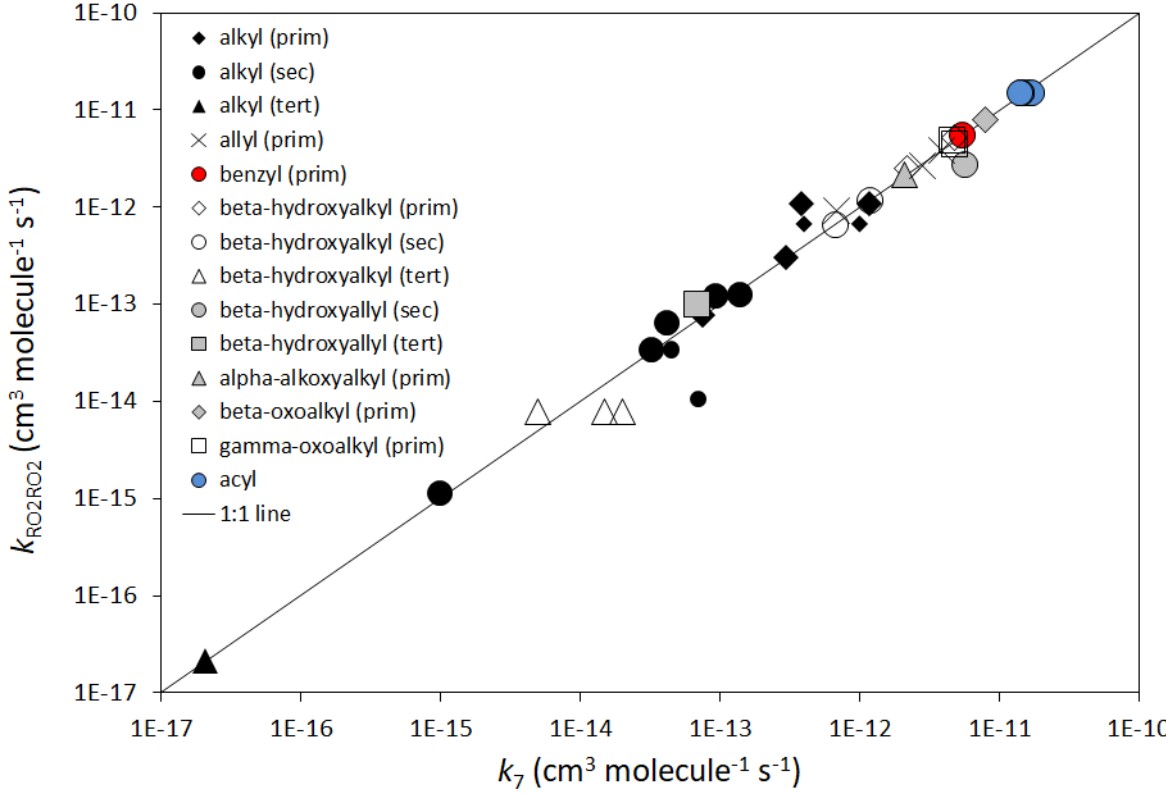

**Figure 4: Scatter plot of estimated rate coefficients ($k_{RO2RO2}$) for peroxy radical self-reactions with those reported ($k_7$), as listed in Tables 9 and 10. Those shown with reduced size symbols are where the reported value of $k_7$ was not corrected for secondary chemistry (see Table 9 comments)**




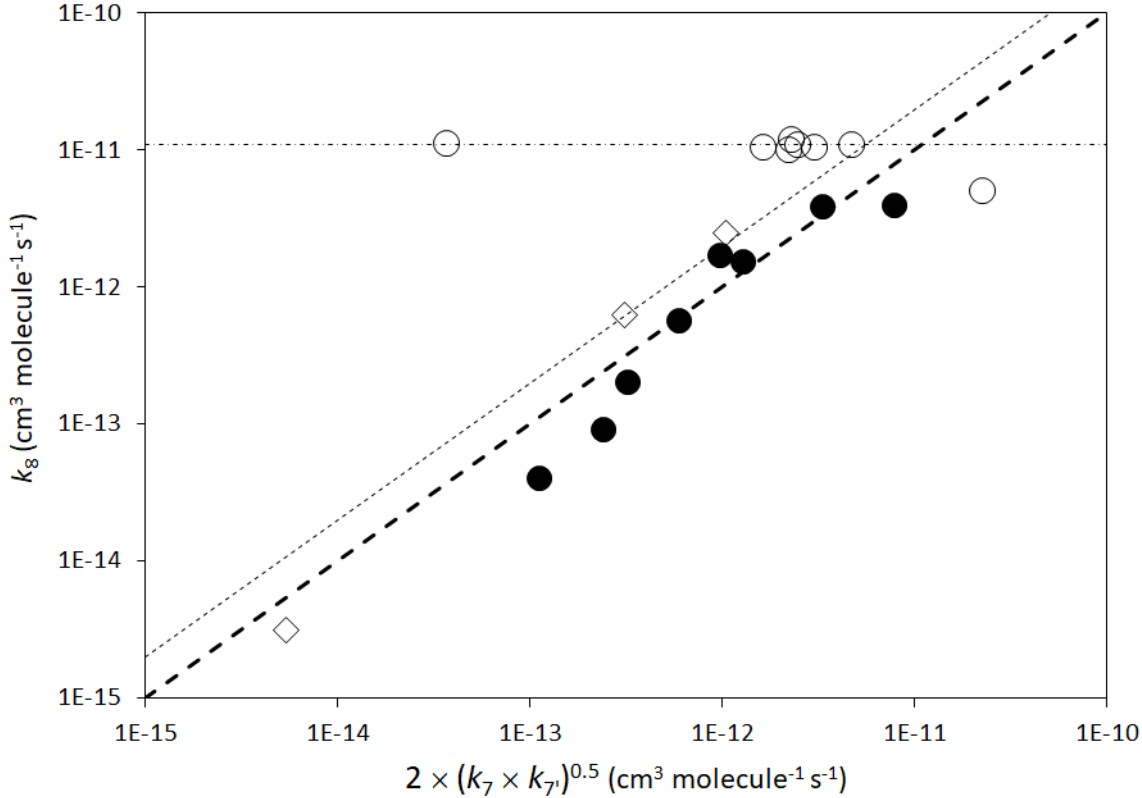

**Figure 5: Scatter plot of rate coefficients for peroxy radical cross-reactions ($k_8$) with the geometric mean of the self-reaction rate coefficients (denoted $k_7$ and $k_{7'}$) for the participating peroxy radicals, $RO_2$ and $R'O_2$. Open circles are reactions involving an acyl peroxy radical and a non-acyl peroxy radical; closed circles are reactions involving combinations of primary and secondary peroxy radicals; open diamonds are reactions involving a tertiary peroxy radical and a primary or secondary peroxy radical. The heavy broken line is a 1:1 relationship; the light broken line is a 2:1 relationship; the dot-dash line is $k_8 = 1.1 \times 10^{-11}$ cm$^3$ molecule$^{-1}$ s$^{-1}$.**





**Table 1. Kinetic data for the reactions of hydrocarbon and oxygenated peroxy radicals with NO. Where available, the temperature dependence is given by $k = A.\exp(-E/RT)$.**

| Peroxy radical | $A$ | $E/R$ | $k_{298\,K}$ | Comment |
|---|---|---|---|---|
| | ($10^{-12}$ cm$^3$ molecule$^{-1}$ s$^{-1}$) | (K) | ($10^{-12}$ cm$^3$ molecule$^{-1}$ s$^{-1}$) | |
| *Alkyl and cycloalkyl* | | | | |
| $CH_3O_2$ | 2.30 | -360 | 7.7 | (a) |
| $C_2H_5O_2$ | 2.55 | -380 | 9.1 | (a) |
| $n$-$C_3H_7O_2$ | 2.90 | -350 | 9.4 | (a),(b) |
| $i$-$C_3H_7O_2$ | 2.70 | -360 | 9.0 | (a),(b) |
| $t$-$C_4H_9O_2$ | | | 8.3 | (a) |
| 2-$C_5H_{11}O_2$ | | | 8.0 | (b) |
| $c$-$C_5H_9O_2$ | | | 10.9 | (b) |
| *Allyl (alk-2-enyl)* | | | | |
| $CH_2$=$CHCH_2O_2$ | | | 10.5 | (b) |
| *β-Hydroxyalkyl* | | | | |
| $HOCH_2CH_2O_2$ | | | 8.7 | (c) |
| propene-derived | | | 9.5 | (c),(d) |
| but-1-ene-derived | | | 9.6 | (c),(e) |
| $CH_3CH(OH)CH(O_2)CH_3$ | | | 9.4 | (c) |
| methylpropene-derived | | | 9.6 | (c),(f) |
| *Hydroxyalkenyl* | | | | |
| buta-1,3-diene derived | | | 8.8 | (c),(g) |
| isoprene-derived | | | 8.8 | (a),(c),(h) |
| *Oxoalkyl* | | | | |
| $CH_3C(O)CH_2O_2$ | | | 8.0 | (a),(i) |
| *Hydroxy-oxyalkyl* | | | | |
| methacrolein-derived | | | 9.3 | (j),(k) |
| methylvinyl ketone-derived | | | 8.4 | (j),(l) |
| *Hydroxy-dioxa-bicyclo* | | | | |
| 1,3,5-trimethylbenzene-derived | | | 7.7 | (m) |
| *Acyl* | | | | |
| $CH_3C(O)O_2$ | 7.5 | -290 | 20 | (a) |
| $C_2H_5C(O)O_2$ | 6.7 | -340 | 21 | (a) |
| $CH_2$=$CH(CH_3)C(O)O_2$ | 8.7 | -290 | 23 | (n) |
| $CH(OOH)CH_2CH_2CH_2CH(OOH)C(O)O_2$ | | | 34 | (o) |

**Comments**

[a] IUPAC Task Group recommendation (http://iupac.pole-ether.fr/); [b] Based on Eberhard and Howard (1996; 1997), Eberhard et al. (1996); [c] Based on Miller et al. (2004); [d] Mixture of $CH_2(OH)CH(O_2)CH_3$ and $CH_2(O_2)CH(OH)CH_3$; [e] Mixture of $CH_2(OH)CH(O_2)C_2H_5$ and $CH_2(O_2)CH(OH)C_2H_5$; [f] Mixture of $CH_2(OH)C(O_2)(CH_3)_2$ and $CH_2(O_2)C(OH)(CH_3)_2$; [g] Mixture of $CH_2(OH)CH(O_2)CH$=$CH_2$, $CH_2(OH)CH$=$CHCH_2O_2$ and $CH_2(O_2)CH(OH)CH$=$CH_2$; [h] Mixture of $CH_2(OH)C(O_2)(CH_3)CH$=$CH_2$, $CH_2(OH)C(CH_3)$=$CHCH_2O_2$, $CH_2(O_2)C(CH_3)(OH)CH$=$CH_2$, $CH_2(OH)CH$=$C(CH_3)CH_2O_2$ and $CH_2(O_2)CH(OH)C(CH_3)$=$CH_2$; [i] Based on Sehested et al. (1998); [j] Based on Hsin and Elrod (2007); [k] Mixture of $CH_2(OH)C(O_2)(CH_3)CHO$ and $CH_2(O_2)C(OH)(CH_3)CHO$; [l] Mixture of $CH_2(OH)CH(O_2)C(=O)CH_3$ and $CH_2(O_2)CH(OH)C(=O)CH_3$; [m] Elrod (2011). Mixture of two complex radicals of molecular formula $HOC_9H_{12}[OO]O_2$, although with one isomer likely dominant; [n] de Gouw and Howard (1997); [o] Berndt et al. (2015). Inferred to be the complex oxo-di-hydroperoxy acyl peroxy radical shown, on the basis of its molecular mass and a proposed mechanism.



**Table 2. Values of the scaling factor, $f_a$, applied to the branching ratio calculation for the reaction of $RO_2$ with NO.**

| Class | substitution | $f_a$ | Comment |
|---|---|---|---|
| default | primary | 0.65 | |
| | secondary | 1.0 | (a) |
| | tertiary | 1.0 | |
| [structure: HO, O, O, OO] | secondary | 1.0 | (b) |
| | tertiary | 0.13 | |
| [structure: HO, R, O, O, OO] | secondary | 0.43 | (b) |
| | tertiary | 0.06 | |

**Comments**

[a] Applied in all cases, except for those covered by comment (b). $f_a = 1$ for secondary peroxy radicals, by definition. The equivalent value for tertiary peroxy radicals, and the lower value for primary peroxy radicals, are based on a consensus of information from Cassanelli et al. (2007), Orlando and Tyndall (2012) and Teng et al. (2015) and on previous consideration of the OH + isoprene system (Jenkin et al., 2015);

[b] Inhibition of nitrate formation has been reported for complex hydroxy-dioxa-bicyclo peroxy radicals derived from aromatics, relative to comparably sized alkyl peroxy radicals, by Rickard et al. (2010) and Elrod (2011), with a particular impact from the presence of alkyl substituents reported by Elrod (2011). The reduced values of $f_a$ for tertiary peroxy radicals, and the general reduction in $f_a$ for peroxy radicals with a neighbouring alkyl substituent (as shown), is inferred from the trend in nitrate yields reported for benzene, toluene, $p$-xylene and 1,3,5-trimethylbenzene by Elrod (2011).





**Table 3. Values of the scaling factor, $f_b$, applied to the branching ratio calculation for the reaction of RO$_2$ with NO [a].**

| Class | $f_b$ | Comment |
|---|---|---|
| $C_nH_{2n+1}OO$ (alkyl peroxy) | 1.0 | (b) |
| OO-C-C(OH)< , OO-C-C(OR)< , OO-C(OH)< , OO-C(OR)< , OO-C-C(ONO$_2$)< ,OO-C-C(OOH)< | 0.65 | (c) |
| δ-hydroxy peroxy | 0.8 | (d) |
| OO-C-C(=O)-, OO-C-C(=O)-O- | 0.3 | (e) |
| OO-C(=O)-, OO-C-O-C(=O)- | 0.0 | (e),(f) |
|  | 0.33 | (g) |
|  | 0.0 | (h) |

**Comments**

[a] A value of $f_b$ needs to be applied to account for the effect of each relevant substituent (see Appendix 1 for further information); [b] $f_b$ = 1 for alkyl peroxy radicals, by definition, and also used as a default in all cases other than those covered by comments (c)-(g); [c] Based on a compromise of information from O'Brien et al. (1998), Matsunaga and Ziemann (2009; 2010), Yeh and Ziemann (2014b) and Teng et al. (2015) for β-hydroxy substituents, but also taking account of information reported for a number of other oxygenated systems (e.g. Tuazon et al., 1998a; Crounse et al., 2012; Lee et al., 2014) and previous consideration of the OH + isoprene system (Jenkin et al., 2015). OO-C-C(OOH)< assumed to be in this category by analogy; [d] Based on the relative impacts of β-OH and δ-OH substituents reported by Yeh and Ziemann (2014a) and previous consideration of the OH + isoprene system (Jenkin et al., 2015); [e] $f_b$ value for OO-C-C(=O)- informed by reported studies of ketone oxidation (Lightfoot et al., 1992; Praske et al., 2015); $f_b$ values for OO-C-C(=O)-O- and OO-C-O-C(=O)- informed by reported studies of ester and dibasic ester oxidation (Tuazon et al., 1998b; 1999; Cavalli et al., 2001; Picquet-Varrault et al., 2001; 2002; Pimentel et al., 2010); [f] $f_b$ = 0 for OO-C(=O)- is based on the general lack of observation of acyl nitrate products in systems where acyl peroxy radicals are formed; [g] Value set to recreate the hydroxy-dioxa-bicyclo nitrate yield reported for benzene by Elrod (2011); In conjunction with the values of $f_a$ in Table 2, this also allows a consistent representation of the yields in the toluene, $p$-xylene and 1,3,5-trimethylbenzene systems (Elrod, 2011; Rickard et al., 2010); [h] $f_b$ = 0 for phenyl peroxy radicals is based on the general lack of observation of phenyl nitrate products during the oxidation of aromatic hydrocarbons.





**Table 4.** Rate coefficients for the reactions of hydrocarbon and oxygenated $RO_2$ radicals with $NO_2$ and for the reverse decomposition of the $RO_2NO_2$ products. Generic rate coefficients for specified $RO_2$ classes are shown in bold font.

| Peroxy radical | $k_0$ | $k_\infty$ | $F_c$ | $k_{298\,K,\,760\,Torr}$ | Comment |
|---|---|---|---|---|---|
| **Forward reaction, $k_f$ (cm$^3$ molecule$^{-1}$ s$^{-1}$)** | | | | | |
| $CH_3O_2$ | $1.2 \times 10^{-30}$ $(T/300)^{-6.9}$ [M] | $1.8 \times 10^{-11}$ | 0.36 | $4.2 \times 10^{-12}$ | (a),(b) |
| $C_2H_5O_2$ | $1.3 \times 10^{-29}$ $(T/300)^{-6.2}$ [M] | $8.8 \times 10^{-12}$ | 0.31 | $5.1 \times 10^{-12}$ | (a),(c) |
| $n$- and $sec$- $C_4H_9O_2$ | - | $9.6 \times 10^{-12}$ | - | $9.6 \times 10^{-12}$ | (d) |
| **$RO_2$** | - | **$9.0 \times 10^{-12}$ (= $k_{f\,PN}$)** | - | **$9.0 \times 10^{-12}$** | (e) |
| $CH_3C(O)O_2$ | $3.28 \times 10^{-28}$ $(T/300)^{-6.87}$ [M] | $1.125 \times 10^{-11}$ $(T/300)^{-1.105}$ | 0.3 | $8.9 \times 10^{-12}$ | (a),(c) |
| $C_2H_5C(O)O_2$ | $1.05 \times 10^{-27}$ $(T/300)^{-6.87}$ [M] | $1.125 \times 10^{-11}$ $(T/300)^{-1.105}$ | 0.36 | $8.9 \times 10^{-12}$ | (a),(f) |
| **$RC(O)O_2$, $ROC(O)O_2$** | - | **$1.125 \times 10^{-11}$ $(T/300)^{-1.105}$ (= $k_{f\,PAN}$)** | - | **$1.1 \times 10^{-11}$** | (e),(g) |
| **Reverse reaction, $k_b$ (s$^{-1}$)** | | | | | |
| $CH_3O_2$ | $9.0 \times 10^{-5}$ exp(-9690/T) [M] | $1.1 \times 10^{16}$ exp(-10560/$T$) | 0.36 | 1.5 | (a),(c) |
| $C_2H_5O_2$ | $4.8 \times 10^{-4}$ exp(-9285/T) [M] | $8.8 \times 10^{15}$ exp(-10440/$T$) | 0.31 | 3.4 | (a),(c) |
| $n$- and $sec$- $C_4H_9O_2$ | - | $8.3 \times 10^{15}$ exp(-10368/$T$) | - | 6.4 | (h) |
| $C_6H_{13}O_2$ isomers | - | $7.5 \times 10^{15}$ exp(-10368/$T$) | - | 5.8 | (h) |
| $C_8H_{17}O_2$ isomers | - | $4.8 \times 10^{15}$ exp(-10368/$T$) | - | 3.7 | (h) |
| **$RO_2$** | - | **$7.6 \times 10^{15}$ exp(-10400/$T$) (= $k_{b\,PN}$)** | - | **5.3** | (e),(i) |
| $CH_3C(O)O_2$ | $1.1 \times 10^{-5}$ exp(-10100/T) [M] | $1.9 \times 10^{17}$ exp(-14100/$T$) | 0.3 | $4.3 \times 10^{-4}$ | (a),(c) |
| $C_2H_5C(O)O_2$ | $1.7 \times 10^{-3}$ exp(-11280/T) [M] | $8.3 \times 10^{16}$ exp(-13940/$T$) | 0.36 | $3.6 \times 10^{-4}$ | (a),(c) |
| $CH_2=C(CH_3)C(O)O_2$ | - | $1.6 \times 10^{16}$ exp(-13500/$T$) | - | $3.5 \times 10^{-4}$ | (c),(j) |
| **$RC(O)O_2$** | - | **$5.2 \times 10^{16}$ exp(-13850/$T$) (= $k_{b\,PAN}$)** | - | **$3.4 \times 10^{-4}$** | (e),(k) |
| **$ROC(O)O_2$** | - | **$2 \times k_{b\,PAN}$** | - | **$6.8 \times 10^{-4}$** | (e),(l) |

**Comments:** [a] Rate coefficient for a pressure-dependent reaction is calculated using the expression: $k = F.k_0.k_\infty/(k_0 + k_\infty)$, where $\log_{10}F = \log_{10}(F_c)/(1+[\log_{10}(k_0/k_\infty)/N]^2)$ and $N = [0.75 - 1.27 \log_{10}(F_c)]$ (see http://iupac.pole-ether.fr/); [b] Based on the evaluation of Golden (2005); [c] Recommended by the IUPAC Task Group (http://iupac.pole-ether.fr/); [d] Reported by McKee et al. (2016) for a mixture of $n$-$C_4H_9O_2$ and $sec$-$C_4H_9O_2$ formed from reaction of Cl with butane; [e] assumed generic rate coefficient; [f] $k_\infty$ assumed equivalent to that for $CH_3C(O)O_2$ + $NO_2$ reaction. $k_0$ scaled relative to that for $CH_3C(O)O_2$ to preserve the $C_2H_5C(O)O_2$ + $NO_2 \rightleftharpoons C_2H_5C(O)OONO_2$ equilibrium constant, $k_f/k_b$, over the pressure range 100-760 Torr. $F_c$ is equivalent to that recommended for $k_b$; [g] Forward reaction rate coefficient, $k_{f\,PAN}$, is based on $k_\infty$ for the $CH_3C(O)O_2$ + $NO_2$ reaction; [h] Based on Zabel et al. (1989), as recommended by Lightfoot et al. (1992), for isomeric mixtures formed from reactions of Cl with butane, hexane or octane. Assumed to be at high pressure limit at 800 Torr; [i] $k_{b\,PN}$ is rounded average of the reported rate coefficients for $C_2$-$C_8$ alkyl peroxy radicals; [j] Based on Roberts and Bertman (1992). Assumed to be at high pressure limit at 760 Torr; [k] $k_{b\,PAN}$, is based on a value of $3.4 \times 10^{-4}$ s$^{-1}$ at 298 K, which is the average of those reported for $n$-$C_3H_7C(O)OONO_2$, $n$-$C_4H_9C(O)OONO_2$ and $n$-$C_5H_{11}C(O)OONO_2$ (Kabir et al, 2014) and $CH_2=C(CH_3)C(O)O_2$ (Roberts and Bertman, 1992) (see Fig. 1). $E/R$ is based on the average of the high pressure limit values for $CH_3C(O)O_2NO_2$, $C_2H_5C(O)O_2NO_2$ and $CH_2=C(CH_3)C(O)O_2$, and also consistent with the approximate value for $n$-$C_5H_{11}C(O)OONO_2$ (Kabir et al., 2014); [l] Pressure-independent generic rate coefficient for thermal decomposition of $ROC(O)O_2$ is a factor of two greater, based on data for $CH_3OC(O)O_2$ and $C_6H_5OC(O)O_2$ (Kirchner et al., 1999), with reduced thermal stability also consistent with data for $C_2H_5OC(O)O_2$ (Bossolasco et al., 2011).





**Table 5. Kinetic data for the reactions of alkyl and oxygenated peroxy radicals with $NO_3$. Where available, the temperature dependence is given by $k = A.\exp(-E/RT)$.**

| Peroxy radical | $A$ | $E/R$ | $k_{298 K}$ | Comment |
|---|---|---|---|---|
| | $(10^{-12} \text{ cm}^3 \text{ molecule}^{-1} \text{ s}^{-1})$ | (K) | $(10^{-12} \text{ cm}^3 \text{ molecule}^{-1} \text{ s}^{-1})$ | |
| *Alkyl and cycloalkyl* | | | | |
| $CH_3O_2$ | | | 1.2 | (a) |
| $C_2H_5O_2$ | 8.9 | 390 | 2.4 | (b) |
| $c\text{-}C_5H_9O_2$ | | | ~1.2 | (c) |
| $c\text{-}C_6H_{11}O_2$ | | | 1.9 | (c) |
| *β-Hydroxyalkyl* | | | | |
| $(CH_3)_2C(OH)CH_2O_2$ | 16 | 480 | 3.2 | (d) |
| *Alkoxyalkyl* | | | | |
| $CH_3OCH_2O_2$ | 13.6 | 435 | 3.2 | (d) |
| *Oxoalkyl* | | | | |
| $CH_3C(O)CH_2O_2$ | 5.47 | 282 | 2.1 | (d) |
| *Acyl* | | | | |
| $CH_3C(O)O_2$ | | | 3.2 | (e) |

**Comments**
[a] IUPAC Task Group recommendation (http://iupac.pole-ether.fr/); [b] $k_{298 K}$ based on an average of the values reported by Biggs et al. (1995), Ray et al. (1996), Vaughan et al. (2006) and Laversin et al. (2016). $E/R$ taken from Laversin et al. (2016); [c] Taken from Vaughan et al. (2006); [d] Taken from Kalalian et al. (2018) ; [e] Taken from Doussin et al. (2003). Canosa-Mas et al. (1996) reported a comparable value of $k = (4.0 \pm 1.0) \times 10^{-12} \text{ cm}^3 \text{ molecule}^{-1} \text{ s}^{-1}$ over the range 403-443 K.

5    **Table 6. Kinetic data for the reactions of peroxy radicals with OH. Where available, the temperature dependence is given by $k = A.\exp(-E/RT)$.**

| Peroxy radical | $A$ | $E/R$ | $k_{298 K}$ | Comment |
|---|---|---|---|---|
| | $(10^{-11} \text{ cm}^3 \text{ molecule}^{-1} \text{ s}^{-1})$ | (K) | $(10^{-10} \text{ cm}^3 \text{ molecule}^{-1} \text{ s}^{-1})$ | |
| $CH_3O_2$ | 3.7 | -350 | 1.2 | (a) |
| $C_2H_5O_2$ | | | 1.2 | (b) |
| (*n*- and *i*-) $C_3H_7O_2$ | | | 1.4 | (c) |
| (*n*- and *sec*-) $C_4H_9O_2$ | | | 1.5 | (c) |

**Comments**
[a] IUPAC Task Group recommendation (http://iupac.pole-ether.fr/) based on Assaf et al. (2016) and Yan et al. (2016), with an uncertainty of a factor of two assigned to $k_{298K}$; [b] IUPAC Task Group recommendation (http://iupac.pole-ether.fr/) based on Faragó et al. (2015), with an uncertainty of a factor of 1.6 assigned to $k_{298K}$. A consistent value of $k_{298 K} = (1.3 \pm 0.3) \times 10^{-10} \text{ cm}^3 \text{ molecule}^{-1} \text{ s}^{-1}$ has more recently been reported by Assaf et al. (2017b); [c] Global rate coefficients, $k_{298K} = (1.4 \pm 0.3) \times 10^{-10}$ and $(1.5 \pm 0.3) \times 10^{-10} \text{ cm}^3 \text{ molecule}^{-1} \text{ s}^{-1}$, reported by Assaf et al. (2017b) for isomeric mixtures of peroxy radicals formed from reactions of Cl atoms with propane and butane, respectively.





**Table 7.** Kinetic data for the reactions of hydrocarbon and oxygenated peroxy radicals with HO$_2$. Where available, the temperature dependence is given by $k = A.\exp(-E/RT)$.

| Peroxy radical | A | E/R | $k_{298\,K}$ | Comment |
|---|---|---|---|---|
| | ($10^{-13}$ cm$^3$ molecule$^{-1}$ s$^{-1}$) | (K) | ($10^{-12}$ cm$^3$ molecule$^{-1}$ s$^{-1}$) | |
| ***Alkyl and cycloalkyl*** | | | | |
| CH$_3$O$_2$ | 3.8 | -780 | 5.2 | (a) |
| C$_2$H$_5$O$_2$ | 6.4 | -710 | 6.9 | (a) |
| *neo*-C$_5$H$_{11}$O$_2$ | 1.4 | -1380 | 14.4 | (b) |
| *c*-C$_5$H$_9$O$_2$ | 2.1 | -1323 | 17.8 | (b) |
| *c*-C$_6$H$_{11}$O$_2$ | 2.6 | -1245 | 17.0 | (b) |
| decane-derived | | | 19.5 | (c),(d) |
| tetradecane-derived | | | 21.1 | (c),(e) |
| ***Allyl (alk-2-enyl)*** | | | | |
| CH$_2$=CHCH$_2$O$_2$ | ~10 | -700 | ~10 | (f) |
| ***Benzyl*** | | | | |
| C$_6$H$_5$CH$_2$O$_2$ | 1.5 | -1310 | 12.0 | (a) |
| ***α-Hydroxyalkyl*** | | | | |
| HOCH$_2$O$_2$ | 0.056 | -2300 | 12.0 | (a) |
| ***β-Hydroxyalkyl*** | | | | |
| HOCH$_2$CH$_2$O$_2$ | | | 13.0 | (a) |
| CH$_3$CH(OH)CH(O$_2$)CH$_3$ | | | 15.0 | (a),(g) |
| (CH$_3$)$_2$C(OH)CH$_2$O$_2$ | 0.56 | -1650 | 14.0 | (a) |
| (CH$_3$)$_2$C(OH)C(O$_2$)(CH$_3$)$_2$ | | | 15.0 | (c) |
| HO-*c*-C$_6$H$_{10}$O$_2$ | | | 22.4 | (c),(h) |
| α-pinene-derived | | | 20.9 | (c),(i) |
| γ-terpinene-derived | | | 19.7 | (c),(i) |
| *d*-limonene-derived | | | 21.2 | (c),(i) |
| ***Hydroxyalkenyl*** | | | | |
| isoprene-derived | | | 17.4 | (c),(j) |
| ***Alkoxyalkyl*** | | | | |
| CH$_3$OCH$_2$O$_2$ | | | ~10 | (k) |
| ***Oxoalkyl*** | | | | |
| CH$_3$C(O)CH$_2$O$_2$ | | | 9.0 | (a),(l) |
| ***Acyl*** | | | | |
| CH$_3$C(O)O$_2$ | 31.4 | -580 | 22.0 | (m) |
| C$_6$H$_5$C(O)O$_2$ | 110 | -364 | 37.0 | (a),(n) |

**Comments**

[a] IUPAC Task Group recommendation (http://iupac.pole-ether.fr/); [b] Based on Rowley et al. (1992a; 1992b) and Boyd et al. (2003a); [c] Taken from Boyd et al. (2003a); [d] Mixture of C$_{10}$H$_{21}$O$_2$ radicals derived from the reaction of OH with decane; [e] Mixture of C$_{14}$H$_{29}$O$_2$ radicals derived from the reaction of OH with tetradecane; [f] Approximate value from Boyd et al. (1996), based on extrapolation of higher temperature data (393-426 K) using assumed value of E/R = -700 K; [g] Taken from Jenkin and Hayman (1995); [h] Derived from the reaction of OH with cyclohexene. RO$_2$ population dominated by β-hydroxy peroxy radical, HO-*c*-C$_6$H$_{10}$-O$_2$, formed from OH addition; [i] RO$_2$ population dominated by hydroxy peroxy radicals formed from OH addition to the given monoterpene; [j] Mixture of HOC$_5$H$_8$O$_2$ radicals derived from the reaction of OH with isoprene; [k] Approximate value from Jenkin et al. (1993a), based on steady state concentration of HO$_2$ formed from the self-reaction of CH$_3$OCH$_2$O$_2$ during modulated photolysis; [l] Based on Bridier et al. (1993); [m] $k_{298\,K}$ based on Groß et al. (2014) and Winiberg et al. (2016). E/R determined by correcting previously recommended value (-980 K [a]) for inferred effects of radical formation channel over the range 250-300 K (see Sect. S4); [n] Based on Roth et al. (2010).





**Table 8. Branching ratios assigned to reaction channels (R6a)-(R6e) for reactions of hydrocarbon and oxygenated peroxy radical classes with HO$_2$ at 298 K.**

| Peroxy radical class | Channel branching ratio | | | | | Comment |
|---|---|---|---|---|---|---|
| | $k_{6a}/k_6$ | $k_{6b}/k_6$ | $k_{6c}/k_6$ | $k_{6d}/k_6$ | $k_{6e}/k_6$ | |
| Alkyl (and default) | 1.00 | - | - | - | - | (a) |
| Acyl (R ≠ phenyl) | 0.37 | 0.13 | - | 0.50 | - | (b) |
| Acyl (R = phenyl) | 0.65 | 0.15 | - | 0.20 | - | (c) |
| β-Oxoalkyl (prim.) | 0.82 | - | - | 0.18 | - | (d) |
| β-Oxoalkyl (sec., tert.) | 0.52 | - | - | 0.48 | - | (e) |
| α-Alkoxyalkyl (prim., sec.) | 0.54 | - | 0.26 | - | 0.20 | (f) |
| α-Alkoxyalkyl (tert.) | 1.00 | - | - | - | - | (g) |
| β-Hydroxyallyl | 0.92 | - | - | 0.08 | - | (h) |
| β-Nitrooxyallyl | 0.50 | - | - | 0.50 | - | (i) |

**Comments**

[a] Based on studies of CH$_3$O$_2$ and C$_2$H$_5$O$_2$ (as summarised by Orlando and Tyndall, 2012), and also used as a default in all cases other than those covered by comments (b)-(i); [b] Based on studies of CH$_3$C(O)O$_2$ (Niki et al., 1985; Horie and Moortgat, 1992; Hasson et al., 2004; Jenkin et al., 2007; Dillon and Crowley, 2008; Groß et al., 2014; Winiberg et al., 2016); see Sect. S4. Hasson et al. (2012) also reported broadly comparable branching ratios for C$_2$H$_5$C(O)O$_2$ and C$_2$H$_5$C(O)O$_2$; [c] Based on studies of C$_6$H$_5$C(O)O$_2$. $k_{6d}/k_6$ based on Dillon and Crowley (2008) and Roth et al. (2010), with other branching ratios based on those reported by Roth et al. (2010); [d] Based on studies of CH$_3$C(O)CH$_2$O$_2$ (Jenkin et al., 2008; Dillon and Crowley, 2008; Hasson et al., 2012); [e] Based on studies of CH$_3$C(O)CH(O$_2$)CH$_3$ (Dillon and Crowley, 2008; Hasson et al., 2012) and of CH$_3$C(O)CH(O$_2$)CH$_2$OH (Praske et al. (2015). Praske et al. (2015) also reported possible minor contribution of channel (R6e) and/or (R6c) for CH$_3$C(O)CH(O$_2$)CH$_2$OH; [f] Based on studies of HOCH$_2$O$_2$ (Jenkin et al., 2007) and CH$_3$OCH$_2$O$_2$ (Jenkin et al., 2010). Contribution of OH formation in those studies was originally attributed to channel (R6d), but is allocated here to channel (R6e) on the basis of the theoretical study of Nguyen et al. (2010); [g] Full reaction is assigned to channel (R6a), because channels (R6c) and (R6e) are unavailable for tertiary radicals owing to the absence of an α- H atom; [h] Based on study of hydroxyallyl peroxy radicals formed in isoprene system by Liu et al. (2013), with support from the studies of Paulot et al. (2009) and Navarro et al. (2013); [i] Based on study of nitrooxyallyl radicals formed in isoprene system by Schwantes et al. (2015).



**Table 9. Kinetic data for the self-reactions of hydrocarbon peroxy radicals. Where available, the temperature dependence is given by $k = A.\exp(-E/RT)$.**

| Peroxy radical | $A$ | $E/R$ | $k_{298\,K}$ | Comment |
|---|---|---|---|---|
| | ($10^{-13}$ cm$^3$ molecule$^{-1}$ s$^{-1}$) | (K) | ($10^{-14}$ cm$^3$ molecule$^{-1}$ s$^{-1}$) | |
| $CH_3O_2$ | 1.03 | -365 | 35 | (a) |
| *Alkyl* | | | | |
| *Primary* | | | | |
| $C_2H_5O_2$ | 0.76 | 0 | 7.6 | (a) |
| $n$-$C_3H_7O_2$ | | | 30 | (a) |
| $n$-$C_4H_9O_2$ | | | 40 | (b)* |
| $i$-$C_4H_9O_2$ | | | 100 | (b)* |
| $n$-$C_5H_{11}O_2$ | | | 39 | (c) |
| $neo$-$C_5H_{11}O_2$ | 0.017 | -1960 | 120 | (d) |
| *Secondary* | | | | |
| $i$-$C_3H_7O_2$ | 16 | 2200 | 0.1 | (a) |
| $sec$-$C_4H_9O_2$ | | | 7 | (b)* |
| $sec$-$C_5H_{11}O_2$ | | | 3.3 | (c),(e) |
| $sec$-$C_{10}H_{21}O_2$ | | | 9.4 | (c),(f) |
| $sec$-$C_{12}H_{25}O_2$ | | | ~14 | (c),(f) |
| $c$-$C_5H_9O_2$ | 2.9 | 555 | 4.5 | (g)* |
| $c$-$C_6H_{11}O_2$ | 0.77 | 184 | 4.2 | (g) |
| *Tertiary* | | | | |
| $t$-$C_4H_9O_2$ | 100 | 3900 | 0.0021 | (a) |
| *Allyl (alk-2-enyl)* | | | | |
| $CH_2=CHCH_2O_2$ (primary) | 0.54 | -760 | 69 | (h) |
| *Benzyl* | | | | |
| $C_6H_5CH_2O_2$ (primary) | 0.24 | -1620 | 550 | (a) |

**Comments**

*Reported rate coefficient not corrected for the effects of secondary chemistry, which can lead to either an overestimate or underestimate of the rate coefficient; [a] IUPAC Task Group recommendation (http://iupac.pole-ether.fr/); [b] Taken from Glover et al. (2005); [c] Taken from Boyd et al. (1999); [d] Based on Lightfoot et al. (1990); [e] Mixture of 2-pentyl and 3-pentyl peroxy radicals ; [f] Mixture of secondary peroxy radicals of given formula; [g] Based Rowley et al. (1991;1992c); [h] Based on Jenkin et al. (1993b) and Boyd et al. (1996).





**Table 10. Kinetic data for the self-reactions of oxygenated peroxy radicals. Where available, the temperature dependence is given by $k = A.\exp(-E/RT)$.**

| Peroxy radical | $A$ | $E/R$ | $k_{298\,K}$ | Comment |
|---|---|---|---|---|
| | ($10^{-13}$ cm$^3$ molecule$^{-1}$ s$^{-1}$) | (K) | ($10^{-14}$ cm$^3$ molecule$^{-1}$ s$^{-1}$) | |
| *α-Hydroxyalkyl* | | | | |
| HOCH$_2$O$_2$ (primary) | | | 620 | (a) |
| *β-Hydroxyalkyl* | | | | |
| *Primary* | | | | |
| HOCH$_2$CH$_2$O$_2$ | 0.78 | -1000 | 220 | (a) |
| (CH$_3$)$_2$C(OH)CH$_2$O$_2$ | 0.14 | -1740 | 480 | (a) |
| *Secondary* | | | | |
| CH$_3$CH(OH)CH(O$_2$)CH$_3$ | 0.077 | -1330 | 67 | (a) |
| HO-$c$-C$_6$H$_{10}$O$_2$ | | | 120 | (b) |
| *Tertiary* | | | | |
| (CH$_3$)$_2$C(O$_2$)CH$_2$OH | | | 1.5 | (b) |
| (CH$_3$)$_2$C(OH)C(O$_2$)(CH$_3$)$_2$ | 5.9 | 1420 | 0.5 | (c) |
| HO-$c$-C$_6$H$_8$(CH$_3$)$_2$O$_2$ | | | 2.0 | (b) |
| *Hydroxyallyl (hydroxyalk-2-enyl)* | | | | |
| *Primary* | | | | |
| HOCH$_2$CH=CHCH$_2$O$_2$ (δ-hydroxy) | | | 280 | (d) |
| HOCH$_2$C(CH$_3$)=C(CH$_3$)CH$_2$O$_2$ (δ-hydroxy) | | | 390 | (d) |
| *Secondary* | | | | |
| HOCH$_2$CH(O$_2$)CH=CH$_2$ (β-hydroxy) | | | 570 | (d) |
| *Tertiary* | | | | |
| HOCH$_2$C(CH$_3$)(O$_2$)C(CH$_3$)=CH$_2$ (β-hydroxy) | | | 6.9 | (d) |
| *Alkoxyalkyl* | | | | |
| CH$_3$OCH$_2$O$_2$ (primary) | | | 210 | (a) |
| *β-Oxoalkyl* | | | | |
| CH$_3$C(O)CH$_2$O$_2$ (primary) | | | 800 | (a) |
| *γ-Oxoalkyl* | | | | |
| CH$_3$C(O)C(CH$_3$)$_2$CH$_2$O$_2$ (primary) | | | 480 | (e) |
| $t$-C$_4$H$_9$C(O)C(CH$_3$)$_2$CH$_2$O$_2$ (primary) | | | 460 | (e) |
| *Acyl* | | | | |
| CH$_3$C(O)O$_2$ | 29 | -500 | 1600 | (a) |
| C$_2$H$_5$C(O)O$_2$ | | | 1700 | (a) |
| (CH$_3$)$_2$CHC(O)O$_2$ | | | 1440 | (f) |
| (CH$_3$)$_3$CC(O)O$_2$ | | | 1440 | (f) |
| C$_6$H$_5$C(O)O$_2$ | 3.4 | -1110 | 1400 | (a) |
| *Alkoxyacyl* | | | | |
| CH$_3$OC(O)O$_2$ (and HC(O)OCH$_2$O$_2$) | | | 2300 | (g) |

**Comments**
[a] IUPAC Task Group recommendation (http://iupac.pole-ether.fr/); [b] Taken from on Boyd et al. (2003b); [c] Based on Jenkin and Hayman (1995) and Boyd et al. (1997); [d] Taken from Jenkin et al. (1998); [e] Based on Le Crâne et al. (2006); [f] Based on Tomas and Lesclaux (2000) and Le Crâne et al. (2004); [g] Taken from Hansen et al. (2003). The kinetics of the two peroxy radicals formed from the reaction of Cl or F with methyl formate reported to possess indistinguishable kinetics.




**Table 11. Kinetic data for the cross-reactions of hydrocarbon or oxygenated peroxy radicals at 298 K. Where available, the temperature dependence expression is given in the comments.**

| Peroxy radical 1 | Peroxy radical 2 | $k_{298\,K}$ | Comment |
|---|---|---|---|
| | | $(10^{-13}\ cm^3\ molecule^{-1}\ s^{-1})$ | |
| $CH_3O_2$ | $C_2H_5O_2$ | 2 | (a) |
| | $neo\text{-}C_5H_{11}O_2$ | 15 | (b) |
| | $c\text{-}C_6H_{11}O_2$ | 0.9 | (b) |
| | $t\text{-}C_4H_9O_2$ | 0.031 | (a),(c) |
| | $CH_2=CHCH_2O_2$ | 17 | (b),(d) |
| | $C_6H_5CH_2O_2$ | < 20 | (b) |
| | $CH_3C(O)CH_2O_2$ | 38 | (a) |
| | $CH_3C(O)O_2$ | 110 | (a),(e) |
| $C_2H_5O_2$ | $neo\text{-}C_5H_{11}O_2$ | 5.6 | (b) |
| | $c\text{-}C_6H_{11}O_2$ | 0.4 | (b) |
| | $CH_3C(O)O_2$ | 100 | (b) |
| | $C_2H_5C(O)O_2$ | 120 | (a) |
| $CH_3C(O)O_2$ | $CH_3C(O)CH_2O_2$ | 50 | (a) |
| | $c\text{-}C_6H_{11}O_2$ | 104 | (f) |
| | $t\text{-}C_4H_9O_2$ | 111 | (f) |
| | $sec\text{-}C_{10}H_{21}O_2$ | 109 | (f) |
| | $sec\text{-}C_{12}H_{25}O_2$ | 105 | (f) |
| $HO\text{-}c\text{-}C_6H_9(CH_3)O_2$ (secondary) | $HO\text{-}c\text{-}C_6H_9(CH_3)O_2$ (tertiary) | 6.2 | (g) |
| $HOCH_2CH=CHCH_2O_2$ | $HOCH_2CH(O_2)CH=CH_2$ | 39 | (h) |
| $HOCH_2C(CH_3)=C(CH_3)CH_2O_2$ | $HOCH_2C(CH_3)(O_2)C(CH_3)=CH_2$ | 25 | (h) |

**Comments**
[a] IUPAC Task Group recommendation (http://iupac.pole-ether.fr/); [b] Taken from Villenave et al. (1996); [c] Temperature dependence expression is $3.8 \times 10^{-13}$ exp($-1430/T$); [d] Temperature dependence expression is $2.8 \times 10^{-13}$ exp($515/T$); [e] Temperature dependence expression is $2.0 \times 10^{-12}$ exp($500/T$); [f] Taken from Villenave et al. (1998); [g] Taken from Boyd et al. (2003b). The structures refer to the isomeric secondary and tertiary peroxy radicals formed from the addition of OH to 1-methylcyclohexene; [h] Taken from Jenkin et al. (1998). Presented values are limited to those reported for the cross reactions of the major radicals formed from the terminal addition of OH to buta-1,3-diene and the terminal addition of OH to 2,3-dimethyl-buta-1,3-diene.





**Table 12. Substituent activation factors applied to self-reaction rate coefficients, based on Eq. (16).**

| Substituent | $\alpha$ | $\beta$ | Comment |
|---|---|---|---|
| alkyl | 1.00 | 0 | (a) |
| $\beta$-hydroxy | $8.0 \times 10^{-5}$ | 0.4 | (b) |
| allyl (alk-2-enyl) | $4.0 \times 10^{-2}$ | 0.15 | (c) |
| benzyl | $5.8 \times 10^{-2}$ | 0.15 | (d),(e) |
| $\alpha$-alkoxy | $7.0 \times 10^{-5}$ | 0.4 | (f),(g) |
| $\beta$-oxo | $1.6 \times 10^{-4}$ | 0.4 | (h),(g) |
| $\gamma$-oxo | $5.3 \times 10^{-5}$ | 0.4 | (i),(g) |

Comments
[a] $\alpha = 1.00$ and $\beta = 0$ by definition for alkyl peroxy radicals. These are also used as a default for peroxy radical classes not covered by comments (b)-(i), with the exception of acyl peroxy radicals (discussed in Sect. 2.6); [b] Based on data for $\beta$-hydroxyalkyl peroxy radicals in Table 10; [c] Based on data for allyl and $\delta$-hydroxyallyl peroxy radicals in Tables 9 and 10; [d] Based on data for $C_6H_5CH_2O_2$ (Table 9); [e] $\beta$ assumed equivalent to that for allyl substituent; [f] Based on data for $CH_3OCH_2O_2$ (Table 10); [g] $\beta$ assumed equivalent to that for $\beta$-hydroxy substituent; [h] Based on data for $CH_3C(O)CH_2O_2$ (Table 10); [i] Based on data for $\gamma$-oxoalkyl peroxy radicals in Table 10.

**Table 13. Branching ratios assigned to parameterized permutation reactions of RO$_2$ (see text).**

| Peroxy radical class | Channel branching ratio | | | Comment |
|---|---|---|---|---|
| | $k_{9a}/k_9$ | $k_{9b}/k_9$ | $k_{9c}/k_9$ | |
| $CH_3O_2$ | $7.2 \times \exp(-885/T)$ | $(1-(k_{9a}/k_9))/2$ | $(1-(k_{9a}/k_9))/2$ | (a) |
| Primary and secondary | 0.6 | 0.2 | 0.2 | (b) |
| Tertiary and acyl | 0.8 | - | 0.2 | (c) |

Comments
[a] Based on IUPAC Task Group recommendation for the $CH_3O_2$ self-reaction (http://iupac.pole-ether.fr/). An alternative representation using temperature-dependent channel rate coefficients is provided in Sect. S5; [b] Based on a rounded mean of the reported 298 K branching ratios for the self-reactions of $C_2H_5O_2$, i-$C_3H_7O_2$, $HOCH_2CH_2O_2$, $(CH_3)_2C(OH)CH_2O_2$, $CH_3C(O)CH_2O_2$, $CH_3OCH_2O_2$, and $C_6H_5CH_2O_2$ based on IUPAC Task Group recommendations (http://iupac.pole-ether.fr/); neo-$C_5H_{11}O_2$, c-$C_6H_{11}O_2$ and $CH_2=CHCH_2O_2$, based on Lightfoot et al. (1990), Rowley et al. (1991), Jenkin et al. (1993a ; 1993b) and Boyd et al. (1996); and for the self- and cross- reactions of primary and secondary RO$_2$ formed from reactions of OH with conjugated dienes (Jenkin et al., 1998); [c] Based on a rounded mean of the reported 298 K branching ratios for the following cross-reactions: $CH_3C(O)O_2 + CH_3O_2$, $C_2H_5C(O)O_2 + C_2H_5O_2$ and $CH_3C(O)O_2 + CH_3C(O)CH_2O_2$, based on IUPAC Task Group recommendations (http://iupac.pole-ether.fr/); and $HOCH_2C(CH_3)(O_2)C(CH_3)=CH_2 + HOCH_2C(CH_3)=C(CH_3)CH_2O_2$ formed from reaction of OH with 2,3-dimethyl-buta-1,3-diene (Jenkin et al., 1998).



**Table 14. Rate coefficients assigned to template ring-closure reactions of peroxy radicals [a].**

| Radical | Product | | A | E/R | $k_{298\,K}$ | Comment |
|---|---|---|---|---|---|---|
| | | | $(s^{-1})$ | (K) | $(s^{-1})$ | |
| | | | - | - | $(3.6\text{-}2500) \times 10^2$ | (b) |
| | | sec.[c] | $1.0 \times 10^{10}$ | 8140 | 0.014 | (d) |
| | | tert. | $1.0 \times 10^{10}$ | 7740 | 0.053 | (e) |
| | | sec. | $1.0 \times 10^{10}$ | 7740 | 0.053 | (e) |
| | | tert. | $1.0 \times 10^{10}$ | 7340 | 0.20 | (f) |
| | | sec. | $4.8 \times 10^{10}$ | 7850 | 0.17 | (e) |
| | | tert. | $4.8 \times 10^{10}$ | 7450 | 0.67 | (g) |
| | | sec. | $4.8 \times 10^{9}$ | 6750 | 0.70 | (e) |
| | | tert. | $4.8 \times 10^{9}$ | 6350 | 2.7 | (h) |
| | | tert. | $1.4 \times 10^{10}$ | 7100 | 0.63 | (i) |

**Comments**

[a] Temperature dependence of rate coefficient given by $k = A \exp(-(E/R)/T)$. Rapid reaction of the product radical with $O_2$ dominates over the reverse ring-opening reaction under atmospheric conditions. Entries in bold font are based on reported data for the specific or closely-related structures, with other entries inferred using assumptions given in the following comments; [b] Range of 298 K values based on the calculations of Raoult et al. (2004), Glowacki et al. (2009) and Olivella et al. (2009) for the dominant conformer of the example peroxy radical, formed during the oxidation of benzene. Based on these data, and data for other aromatic systems, analogous ring-closure reactions are assumed to be the exclusive fates of corresponding peroxy radicals formed during the oxidation of aromatic hydrocarbons (Jenkin et al., 2018b); [c] Denotes substitution of product radical; [d] Based on information reported by Vereecken and Peeters (2004) for calculations for the given peroxy radical; [e] E/R for formation of a tertiary radical assumed to be 400 K lower than for formation of a secondary radical, corresponding to a difference in E of ≈ 3.3 kJ mol⁻¹. This is consistent with differences in energy barriers reported for formation of secondary and tertiary radicals (Vereecken and Peeters, 2004); [f] Based on the calculations of Vereecken and Peeters (2004) for a relevant tertiary peroxy radical formed during the oxidation of isoprene; [g] Based on the calculations of Vereecken and Peeters (2004) for a relevant tertiary peroxy radical formed during the oxidation of α-pinene. Applies specifically to *anti-* conformers, when the OH and peroxy radical groups on the opposite sides of the ring (as shown), which were calculated to account for 60 % of the *anti-* + *syn-* population (Vereecken and Peeters, 2004); [h] Based on the calculations of Vereecken and Peeters (2004) for a relevant tertiary peroxy radical formed during the oxidation of α-pinene. Applies specifically to *syn-* conformers, when the OH and peroxy radical groups on the same side of the ring (as shown), which were calculated to account for 40 % of the *anti-* + *syn-* population (Vereecken and Peeters, 2004); [i] Based on the calculations of Vereecken and Peeters (2004) for the a relevant tertiary peroxy radical, formed during the oxidation of β-pinene; [h]



**Table 15. Rate coefficients assigned to selected H-shift isomerization reactions of peroxy radicals.**

| Radical | Product(s) | $k(T)$ (s$^{-1}$) | $k_{298\,K}$ (s$^{-1}$) | Comment |
|---|---|---|---|---|
| **1,4 formyl H-shift** | | | | |
| | + OH + CO | $3.0 \times 10^{7}\ \exp(-5300/T)$ | 5.7 | (a),(b) |
| **1,4 hydroxyl H-shift** | | | | |
| | + HO$_2$ | $3.6 \times 10^{12}\ \exp(-6310/T)$ | $2.3 \times 10^{3}$ | (b),(c) |
| | + HO$_2$ | $6.7 \times 10^{12}\ \exp(-5780/T)$ | $2.5 \times 10^{4}$ | (b),(d) |
| | + HO$_2$ | $5.6 \times 10^{12}\ \exp(-6010/T)$ | $9.8 \times 10^{3}$ | (b),(e) |
| **1,5 hydroxyl H-shift** | | | | |
| | + + OH | $1.9 \times 10^{11}\ \exp(-9750/T)$ | $1.2 \times 10^{-3}$ | (b),(f) |
| | + + OH | $1.0 \times 10^{11}\ \exp(-9750/T)$ | $6.2 \times 10^{-4}$ | (b),(f) |
| **1,6 hydroxyalkyl H-shift** | | | | |
| | OOH ... OH | $1.3 \times 10^{10}\ \exp(-8380/T) \times \exp(10^{8}/T^{3})$ | $3.5 \times 10^{-1}$ | (g) |
| **1,6 enol H-shift** | | | | |
| | OOH | $2.4 \times 10^{-1}\ T^{4.1}\ \exp(-2700/T)$ | $3.9 \times 10^{5}$ | (h) |

**Comments**

[a] Based on rate coefficient reported for the methacrolein-derived peroxy radical, HOCH$_2$C(CH$_3$)(O$_2$)C(=O)H, by Crounse et al. (2012). Applied to primary, secondary and tertiary α-formyl peroxy radicals; [b] The initially-formed hydroperoxy-substituted product radical decomposes spontaneously to produce the displayed products; [c] Based on the rate coefficient estimated for CH$_3$CH(OH)O$_2$ by Hermans et al. (2005); Applied to secondary α-hydroxyl peroxy radicals; [d] Based on the rate coefficient estimated for (CH$_3$)$_2$C(OH)O$_2$ by Hermans et al. (2005); Applied to tertiary α-hydroxy peroxy radicals; [e] Based on the rate coefficient estimated for *cyclo*-C$_6$H$_{10}$(OH)O$_2$ by Hermans et al. (2005); Applied generally to cyclic α-hydroxy peroxy radicals (i.e. where the OH and OO groups are substituents to a ring); [f] Based on rate coefficients reported by Peeters et al. (2014) for corresponding unsaturated secondary and tertiary β-hydroxy peroxy radicals formed in isoprene oxidation. Applied generally to unsaturated β-hydroxy peroxy radicals containing the sub-structures shown; [g] Based on the geometric mean of rate coefficients applied to (Z)-CH(OH)C(CH$_3$)=CHCH$_2$O$_2$ (CISOPAO2) and (Z)-CH(OH)CH=C(CH$_3$)CH$_2$O$_2$ (CISOPCO2) in MCM v3.3.1 (Jenkin et al., 2015), based on the calculations of Peeters et al. (2014) and observations of Crounse et al. (2011). Applied generally to unsaturated δ-hydroxy peroxy radicals containing the sub-structure shown, except for CISOPAO2 and CISOPCO2 themselves for which the species-specific rate coefficients are applied (see Table S5). Rapid reaction of the product radical with O$_2$ dominates over the reverse isomerization reaction under atmospheric conditions; [h] Based on the geometric mean of rate coefficients reported for (Z)-HOCH=C(CH$_3$)CH(O$_2$)CH$_2$OH and (Z)-HOCH=CHC(CH$_3$)(O$_2$)CH$_2$OH in the calculations of Peeters and Nguyen (2012). Applied to peroxy radicals containing the sub-structure shown. Rapid reaction of the product radical with O$_2$ dominates over the reverse isomerization reaction under atmospheric conditions.