# Peer review of "Estimation of rate coefficients and branching ratios for reactions of organic peroxy radicals for use in automated mechanism construction"

_Atmospheric Chemistry and Physics, 2019_

## Short Comment (SC1) · 27 Feb 2019

Dear Authors,

Just a quick comment: most of the kinetic data reported for the RO2 cross-reactions are old and have been measured by rather indirect methods. Why have the data reported recently from the direct and speciated measurements of the RO2s not been included (Noziere et al; J. Phys. Chem. 2017, attached) ?

Thank you,

[Figure]

Sincerely,

Barbara

Please also note the supplement to this comment:
https://www.atmos-chem-phys-discuss.net/acp-2019-44/acp-2019-44-SC1-
supplement.pdf

[Figure]

**Supplement:**

THE JOURNAL OF
**PHYSICAL CHEMISTRY A**

Cite This: *J. Phys. Chem. A* 2017, 121, 8453-8464

Article

pubs.acs.org/JPCA

**Speciated Monitoring of Gas-Phase Organic Peroxy Radicals by Chemical Ionization Mass Spectrometry: Cross-Reactions between $CH_3O_2$, $CH_3(CO)O_2$, $(CH_3)_3CO_2$, and c-$C_6H_{11}O_2$**

*Published as part of The Journal of Physical Chemistry virtual special issue "Veronica Vaida Festschrift".*

Barbara Nozière[*,†] and David R. Hanson[‡]

†CNRS/IRCELYON, Villeurbanne, France
‡Augsburg College, Minneapolis, United States

🄢 *Supporting Information*

**ABSTRACT:** Organic peroxy radicals ("$RO_2$", with R organic) are key intermediates in most oxygen-rich systems, where organic compounds are oxidized (natural environment, flames, combustion engines, living organisms, etc). But, until recently, techniques able to monitor simultaneously and distinguish between $RO_2$ species ("speciated" detection) have been scarce, which has limited the understanding of complex systems containing these radicals. Mass spectrometry using proton transfer ionization has been shown previously to detect individual gas-phase $RO_2$ separately. In this work, we illustrate its ability to speciate and monitor several $RO_2$ simultaneously by investigating reactions involving $CH_3O_2$, $CH_3C(O)O_2$, c-$C_6H_{11}O_2$, and $(CH_3)_3CO_2$. The detection sensitivity of each of these radicals was estimated by titration with NO to between 50 and 1000 Hz/ppb, with a factor from 3 to 5 of uncertainties, mostly due to the uncertainties in

knowing the amounts of added NO. With this, the $RO_2$ concentration in the reactor was estimated between $1 \times 10^{10}$ and $1 \times 10^{12}$ molecules $cm^{-3}$. When adding a second radical species to the reactor, the kinetics of the cross-reaction could be studied directly from the decay of the first radical. The time-evolution of two and sometimes three different $RO_2$ was followed simultaneously, as the $CH_3O_2$ produced in further reaction steps was also detected in some systems. The rate coefficients obtained are (in molecule$^{-1}$ cm$^3$ s$^{-1}$): $k_{CH3O2+CH3C(O)O2} = 1.2 \times 10^{-11}$, $k_{CH3O2+t-butylO2} = 3.0 \times 10^{-15}$, $k_{c-hexylO2+CH3O2} = 1.2 \times 10^{-13}$, $k_{t-butylO2+CH3C(O)O2} = 3.7 \times 10^{-14}$, and $k_{c-hexylO2+t-butylO2} = 1.5 \times 10^{-15}$. In spite of their good comparison with the literature and good reproducibility, large uncertainties ($\times 5/5$) are recommended on these results because of those in the detection sensitivities. This work is a first illustration of the potential applications of this technique for the investigation of organic radicals in laboratory and in more complex systems.

**INTRODUCTION**

Organic peroxy radicals ("$RO_2$", with R organic) are key species produced during the oxygen-based (or "aerobic") combustion of organic compounds and thus ubiquitous in the natural environment (atmosphere, surface waters, natural fires, etc), technological processes (combustion engines, power production) and even in living organisms. As their organic structure strongly affects their reactivity, information on their individual reactions is key to the understanding of such oxidation systems and the prediction of their outcome (ozone formation in the atmosphere, preignition in engines, etc). Unfortunately, classical monitoring techniques for $RO_2$, such as UV absorption, electron spin resonance (ESR),[1,2] and electron paramagnetic resonance (EPR; with or without spin trapping), cannot differentiate between different radicals. These techniques thus require even the simplest radical system to be analyzed with complex kinetic models, involving a number of

assumptions, resulting, at best, in large uncertainties in the results and, at worse, in overlooking important reaction channels such as those identified over the past decade.[3-7] As emphasized in reviews of the topic,[3] such technical limits have been the main obstacle to the investigation of $RO_2$ reaction kinetics even in laboratory. Techniques monitoring simultaneously and distinguishing between different $RO_2$ ("speciated" detection) in more complex systems have also been lacking, which has limited the progress of all the fields of research mentioned above. In atmospheric chemistry, indirect techniques have been developed to monitor the $RO_2$ in the atmosphere, which consist in converting all the $RO_2$ into one single species, which is monitored: reacting the $RO_2$ with NO

**Received:** July 2, 2017
**Revised:** September 27, 2017
**Published:** October 16, 2017

and monitoring either the $NO_2$ produced by luminescence ("PERCA")[8] or the $HO_2$ radical produced by laser-induced fluorescence (ROxLIF[9] and FAGE[10]); or reacting the $RO_2$ with labeled $^{34}SO_2$ to produce $H_2^{34}SO_4$, which is measured by mass spectrometry (ROxMAx[11] or PerCIMS[12]). While these techniques are valuable, and the only tools providing information on atmospheric $RO_2$ so far, they lose all information on the individual radicals and provide an overall $RO_2$ signal (or "$\Sigma RO_2$"). This information is not detailed enough to describe accurately the atmospheric radical cycles, as shown by the discrepancies between measured and modeled atmospheric radical levels.[13,14] Even the semispeciation between saturated and unsaturated (alkenes and aromatic) $RO_2$s proposed with FAGE[10] does not account for the orders of magnitude of difference in reactivity within each class of radicals (for instance, between $(CH_3)_3CO_2$ and $CH_3C(O)O_2$ among the saturated ones). Very recent works propose the specific detection of $CH_3O_2$ by conversion into $CH_3O$, which is then measured by LIF,[15] but not of other $RO_2$s.

Mass spectrometry (MS), when combined with a mild ionization technique avoiding fragmentation, is intrinsically speciated, because the ions produced are directly linked to the molecules or radicals analyzed through their mass (or mass/charge ratio, $m/z$). Numerous chemical ionization techniques for $RO_2$ have thus been explored since the 1980s, but few were found suitable to all types of $RO_2$s or systems: electron transfer with $SF_6^-$, $O_2^-$,[16,17] or excited rare gas[18] are only applicable to low-pressure systems; reactions with $I^-$ or $O_3^-$ only work with acylperoxyl radicals $(R-C(O)-O_2)$;[17] reaction with $O_2^+$ leads to some fragmentations of the radicals (75% for $CH_3O_2$).[19] More recently, chemical ionization with $NO_3^-$ was shown to detect highly oxidized C10−C12 $RO_2$ radicals ("HOMs") and used to monitor them in smog chamber,[20−24] and also possibly in the atmosphere,[20] although this was not confirmed. But this technique does not detect smaller, more volatile $RO_2$s that control the atmospheric radical cycles ($CH_3O_2$, $CH_3C(O)O_2$, etc). By contrast, proton transfer with $H_3O^+$ and water clusters, $H_3O^+(H_2O)_n$, which is based on the following ionization reaction:

$$RO_2 + H_3O^+(H_2O)_n$$
$$\rightarrow RO_2H^+(H_2O)_m + (n - m + 1)H_2O \tag{1}$$

seems to be efficient with all the $RO_2$ explored so far ($CH_3O_2$, $C_2H_5O_2$, $CH_3C(O)O_2$, iso-$C_3H_7O_2$, c-$C_6H_{11}O_2$)[25−30] and does not result in fragmentation. It is thus promising for application to complex systems. While previous work has demonstrated the detection of individual gas-phase $RO_2$ by this technique,[25−30] the present work explores its ability to monitor simultaneously different radicals and illustrates the advantages to be gained in their investigation. The radical production system and flow reactor conditions, which were similar to those in ref 30, probably hold an advantage over the turbulent flow reactor technique of Elrod and co-workers,[25−29] by allowing for longer reaction times and possibly larger radical production with the UV lights.

In this work, $CH_3O_2$, $CH_3C(O)O_2$, $(CH_3)_3CO_2$, and c-$C_6H_{11}O_2$ were first produced individually in a flow reactor, and their spectra were characterized. $CH_3O_2$ and $CH_3C(O)O_2$ were chosen for their relevance in the atmosphere, and $(CH_3)_3CO_2$ and c-$C_6H_{11}O_2$ were selected to explore more complex mechanisms and very slow kinetics. Various amounts of NO were then added to the reactor to titrate the radicals and

estimate their detection sensitivities. Finally, a second radical was added periodically to each radical system, and the kinetics of their cross-reactions was investigated, for the first time, from the decays of the individual radicals. In particular, the rate coefficients for the cross-reactions between $CH_3O_2$ and $CH_3C(O)O_2$, and between $CH_3O_2$ and $(CH_3)_3CO_2$, which are known in the literature, were remeasured to validate the method used in this work.

**■ MATERIALS AND METHODS**

**Flow System and Radical Generation.** The radicals were generated and reacted in a cylindrical glass reactor (inner diameter: 5 cm, length: 120 cm, Figure 1) disposed vertically, in

[Figure]

**Figure 1.** Schematic of the experimental setup.

an air flow (2−4 sLm, standard temperature = 273 K and pressure = 1 atm) near atmospheric pressure (0.6−0.9 atm). Typical residence times in the reactor were thus between 30 s and 1 min. Reynold's numbers for the flows were between 100 and 150, thus well in the laminar regime, with a mixing length of ∼25 cm. The organic precursors, $CH_4$, $CH_3CHO$, $CH(CH_3)_3$, or c-$C_6H_{12}$, each from $1 \times 10^{13}$ to $1 \times 10^{16}$ molecule $cm^{-3}$, and $Cl_2$ ($2 \times 10^{15}$ molecule $cm^{-3}$) were mixed into the air flow and introduced in the reactor through the top inlet. As this inlet was situated 10 cm above the reactor itself, this ensured that all the gases were mixed within the top 15 cm (∼10%) of the reactor. The radicals were produced by irradiating the reactor over 280−400 nm with four fluorescent lights (Philips TL12, 40 W) placed symmetrically at 2−3 cm around it. This led to the formation of Cl atoms:

$$Cl_2 + h\nu \rightarrow 2Cl \tag{2}$$

which, then, reacted with the organic precursors to produce each $RO_2$ radical. For $CH_3O_2$:

$$Cl + CH_4 \rightarrow CH_3 + HCl \tag{3}$$

$$CH_3 + O_2 + M \rightarrow CH_3O_2 + M \tag{4}$$

For the $CH_3C(O)O_2$ radical:

$$Cl + CH_3CHO \rightarrow CH_3C(O) \tag{5}$$

$$CH_3C(O) + O_2 + M \rightarrow CH_3C(O)O_2 + M \tag{6}$$

DOI: 10.1021/acs.jpca.7b06456
*J. Phys. Chem. A* 2017, 121, 8453−8464

Irradiation tests performed on $CH_3CHO$ and in the absence of $Cl_2$ in the reactor showed that this compound was not photolyzed by the UV lights.

For the $(H_3C)_3CO_2$ radical:

$$Cl + (H_3C)_3CH \rightarrow (H_3C)_3C + HCl \qquad (7)$$

$$(H_3C)_3C + O_2 + M \rightarrow (H_3C)_3CO_2 + M \qquad (8)$$

And for the $c\text{-}C_6H_{11}O_2$ radical:

$$Cl + c\text{-}C_6H_{12} \rightarrow c\text{-}C_6H_{11} + HCl \qquad (9)$$

$$c\text{-}C_6H_{11} + O_2 + M \rightarrow c\text{-}C_6H_{11}O_2 + M \qquad (10)$$

Different configurations were used in the experiments: either the reactor was irradiated over its entire length or only on the bottom 50 cm (the top part being covered with aluminum foil). Although the second configuration made the production and observation of the radicals slightly easier to control, both gave similar spectra and kinetic results and were thus not differentiated in the results presented here.

**Sampling and Detection.** A schematic of the experimental setup is presented in Figure 1. The gas mixtures and $RO_2$ radicals present in the reactor were sampled for analysis into a quadrupole mass spectrometer built for this project and similar to the one described in refs 30 and 31. But, while in the previous setups with this type of instrument, the output of the reactor was directly integrated to the ionization region of the mass spectrometer,[25−30] in this work the sampling was performed through a line (∼10 cm, diameter 1/4 in.) connecting the bottom outlet of the reactor to the ionization region of the mass spectrometer and equipped with a valve to keep the pressure in the ionization region independent from that in the flow reactor. This ionization region was a cylinder (Delrin, 5 cm of diameter × 5 cm in length) operated at a total pressure from 10 to 15 mbar. Flowing small concentrations of water (<10%) in a flow of $N_2$ (20−40 sccm) through a source maintained at high voltage (typically +800 V) produced a distribution of reagent ions, $H_3O^+(H_2O)_n$ with $n = 1$ to 5. The typical ion drift time in the ionization region was ∼0.3 ms.

Depending on the conditions in the ionization region (temperature, relative humidity), the most abundant ions were for $n = 2$ ($m/z = 55$) or $n = 3$ ($m/z = 73$). These ions were accelerated toward the entrance of the mass spectrometer chamber by the voltage difference between the source (+800 V) and the spectrometer entrance (+20 V). The sampling flow from the reactor (typically 30 sccm) was introduced into the ionization region radially, at approximately two-thirds of its length, so that the sampling flow and the beam of reagent ions mixed efficiently. The signal intensities for various compounds in the spectra varied roughly between 100 and $1 \times 10^5$ Hz, with background signals between 1 and 100 Hz. Mass spectra were recorded with 0.1 amu increments.

As the humidity in the ionization region affected the detection sensitivity for the $RO_2$ (see ref 30 and next section) the ratio of the signal intensities for the two main water clusters, $S_{73}/S_{55}$, was used as a proxy for these conditions and as a variable when describing the detection sensitivities. Note that all the analyses presented in this work (calibrations, kinetic studies), are based on the ratios of the radical signals to the total reagent ion signal (or "water signal") instead of absolute signals (in Hz) to account for potential changes in the detection sensitivities during the experiments. The detection sensitivities reported in this work (in Hz/ppb) were thus

calculated for a total reagent ion signal of $1 \times 10^6$ Hz. However, the water proton signals (especially $S_{55}$ and $S_{73}$) often exceeded the linear range (for count rate) of the electron multiplier; thus, the sensitivities reported in this work cannot be compared to theoretical sensitivities, nor can they be converted into equivalent ion−molecule rate coefficients.

**Determination of the Radical Concentrations and Rate Coefficients.** The method used in this work to determine the $RO_2$ concentrations and detection sensitivities was similar to the one used in our previous study,[30] and it is illustrated in Figure 2: various concentrations of NO (typically

[Figure]

**Figure 2.** Determination of the detection sensitivity by titration with NO for $CH_3O_2$ ($m/z = 84$, experiment of Sept 30, 2015). (A) Evolution of $S_{RO2}$ with time upon addition of different levels of [NO] and determination of $r_{RO2+NO}$ and $\Delta S_{RO2}$. (B) Determination of the detection sensitivity from the slope of $\Delta S_{RO2}$ vs $\Delta[RO_2]$.

0.05 to 2 ppb) were added to the reactor, alternating cycles with ON on and NO off. The decrease in $RO_2$ concentration, $\Delta[RO_2]$, upon adding NO, observed by the decrease in signal $\Delta S_{RO2}$, was then assumed to result from the consumption of each $RO_2$ by 1 equiv of NO. The detection sensitivity, $Sens(RO_2)$, was thus given by the ratio

$$Sens(RO_2) = \frac{\Delta S_{RO2}}{\Delta[RO2]} = \frac{\Delta S_{RO2}}{[NO]} \qquad (11)$$

However, in our work, the added (or initial) [NO] could not be determined precisely from the instrument settings, because it was introduced in very small flows and because of significant losses of NO (and conversion to HONO) to the walls of the system. Instead, added [NO] was determined from the initial decay rate of $RO_2$, $r_{RO2+NO}$ ($s^{-1}$; Figure 2A), divided by the corresponding rate coefficient, $k_{RO2+NO}$ (molecule $cm^3$ $s^{-1}$), taken from the literature:

DOI: 10.1021/acs.jpca.7b06456
*J. Phys. Chem. A* 2017, 121, 8453−8464

$$[\text{NO}] = \frac{r_{\text{RO2+NO}}}{k_{\text{RO2+NO}}} \qquad (12)$$

assuming that NO was well-mixed at the level of the reactor where the measurements were made. The accuracy in this method and in the resulting radical detection sensitivities depended on the accuracy in measuring the decay rate $r_{\text{RO2+NO}}$, thus in correcting for or ruling out potential contributions of the instrument dynamic response, wall losses, and mixing effects. In addition, both the accuracy in measuring the decay rate and the equivalence between $\Delta S_{\text{RO2}}$ and added $[\text{NO}]$ implied that both the decay rate and $\Delta S_{\text{RO2}}$ resulted only from the reaction between $\text{RO}_2$ and NO and were not impacted by other reactions, such as the self-reactions of the $\text{RO}_2$, and their reactions with $\text{HO}_2$ or other $\text{RO}_2$ in the system. The potential contributions of instrument response, wall losses, and mixing effects to the measured $r_{\text{RO2+NO}}$ were eliminated by adding different levels of NO in the system and confirming that $r_{\text{RO2+NO}}$ varied proportionally with $[\text{NO}]$. In particular, this was expected to eliminate potential biases due to mixing effects, as NO was added in very small flows in the reactor, and the flow and mixing patterns in the reactor were thus not expected to depend on $[\text{NO}]$. The component of $\Delta S_{\text{RO2}}$ not varying with $[\text{NO}]$ was thus assumed to be free from mixing effects, and the sensitivity, $\text{Sens}(\text{RO}_2)$, was determined from the slopes of $\Delta S_{\text{RO2}}$ versus $[\text{NO}]$ (Figure 2B), instead of the ratio in eq 11. The potential contributions of the self-reactions and reactions with $\text{HO}_2$ or other $\text{RO}_2$ to the measured $r_{\text{RO2+NO}}$ and $\Delta S_{\text{RO2}}$ will be discussed in the Results section.

The rate coefficients for the cross reactions were determined with a similar method: producing one radical, $\text{R}_1\text{O}_2$, continuously in the reactor and producing the second one, $\text{R}_2\text{O}_2$, periodically by switching the flow of the corresponding precursor on and off. Monitoring both radicals simultaneously allowed determination of the rate coefficients for their cross-reactions, $k_{\text{cross}}$, directly from the initial decay rate of $\text{R}_1\text{O}_2$, $r_{\text{R1O2+R2O2}}$ ($\text{s}^{-1}$):

$$k_{\text{cross}} = \frac{r_{\text{R1O2+R2O2}}}{[\text{R}_2\text{O}_2]} \qquad (13)$$

As in the titration with NO, this method requires the equivalence between $[\text{R}_2\text{O}_2]$ and $\Delta[\text{R}_1\text{O}_2]$, thus that each $\text{R}_1\text{O}_2$ is consumed by one equiv of $\text{R}_2\text{O}_2$, with no contribution of other reactions. $\Delta[\text{R}_1\text{O}_2]$ was, in turn, obtained from the corresponding signals, $\Delta S_{\text{R1O2}}$, and the detection sensitivity obtained from the titration experiments, $\text{Sens}(\text{R}_1\text{O}_2)$. Potential contributions of instrument response, wall losses, or mixing to the measured $r_{\text{R1O2+R2O2}}$ decays were also eliminated by varying $[\text{R}_2\text{O}_2]$ in the reactor. As with NO, flow patterns and mixing times in the reactor were not expected to vary with $[\text{R}_2\text{O}_2]$, as the organic precursors were introduced in small flows compared to the total flow. For all the radicals and over the range of concentrations studied, $r_{\text{R1O2+R2O2}}$ varied proportionally with $[\text{R}_2\text{O}_2]$; thus, $k_{\text{cross}}$ was determined from the component of $r_{\text{R1O2+R2O2}}$ varying with $\Delta[\text{R}_1\text{O}_2]$, rather than from the ratio in eq 13. To validate this method two rate coefficients already known in the literature, namely, $k_{\text{CH3O2+CH3C(O)O2}}$ and $k_{\text{CH3O2+t-butylO2}}$, were remeasured in this work.

**Chemicals.** High-pressure gas mixtures of acetaldehyde and cyclohexane, each ~1% in $\text{N}_2$, were prepared by injecting known amounts of the pure liquids (typically, ~6 mL of acetaldehyde, >99.5%, Fluka, and 10 mL of cyclohexane >99%,

Merck) in evacuated 6 L cylinders, measuring the resulting pressure change (typically, 300−500 mbar) and pressurizing the cylinders with $\text{N}_2$ to a total pressure between 50 and 100 bar.

All the other gas mixtures were purchased from manufacturers: $\text{N}_2$ quality 4.5, synthetic air 80/20, $\text{Cl}_2$, 1% in $\text{N}_2$, $\text{CH}_4$, 1% in $\text{N}_2$, isobutane, 1% in $\text{N}_2$, all Linde. NO, 1% in $\text{N}_2$, Air Liquide.

**■ RESULTS AND DISCUSSION**

Different series of experiments were performed in this work to characterize different aspects of the $\text{RO}_2$ reactivity, which are presented in sections (a) through (d) below. The complete list of experiments is given in Table S1 of the Supporting Information.

**a. Radical Identification and Monitoring of All $\text{RO}_2$s in the System.** The first step of this study was to make sure that the radicals produced in the reactor had the expected mass spectra (main peaks) and to determine potential interferences from other compounds on their masses. The ionization reaction 1 implies that a peroxy radical of molecular weight M produces ions of mass M+1, M+19, M+37, etc. Peaks corresponding to these masses were thus sought for in the mass spectra obtained when irradiating the reaction mixtures. However, the radicals reacted rapidly in the reactor, producing many stable products, following the generic reactions:[3,32]

$$\text{RO}_2 + \text{RO}_2 \rightarrow 2\text{RO} \qquad (16)$$

$$\rightarrow \underline{\text{ROH}} + \underline{\text{R'CHO}} \qquad (17)$$

$$\rightarrow \underline{\text{ROOR}} + \text{O}_2 \qquad (18)$$

$$\text{RO} + \text{O}_2 \rightarrow \underline{\text{R'CHO}} + \text{HO}_2 \qquad (19)$$

$$\text{RO}_2 + \text{HO}_2 \rightarrow \underline{\text{ROOH}} + \text{O}_2 \qquad (20)$$

As proton transfer is sensitive to most oxygenated molecules, all the products underlined above were detected and resulted in much more intense signals than the radicals in the spectra. Thus, to isolate the $\text{RO}_2$ signals, NO was introduced periodically into the reactor, to remove the $\text{RO}_2$ rapidly, according to the reaction:

$$\text{RO}_2 + \text{NO} \rightarrow \text{RO} + \text{NO}_2 \qquad (21)$$

The $\text{RO}_2$ spectra were thus obtained by subtracting the spectra of the reaction mixtures obtained in the absence and in the presence of NO. The results are presented in Figure 3. $\text{CH}_3\text{O}_2$ displayed almost exclusively an ion peak at $m/z = 84$ ($n = 2$ in eq 1, Figure 3A), although occasionally, under very dry conditions, the peak at $m/z = 66$ ($n = 1$) was observed. $\text{CH}_3\text{C(O)O}_2$ was mostly observed at $m/z = 94$ ($n = 1$ in eq 1, Figure 3B) but occasionally also at $m/z = 112$ ($n = 2$). $(\text{CH}_3)_3\text{CO}_2$ displayed two strong peaks at $m/z = 108$ ($n = 1$) and $m/z = 126$ ($n = 2$; Figure 3C), the relative intensity of which depended on the conditions in the reactor and ionization region; $c\text{-C}_6\text{H}_{11}\text{O}_2$ was the easiest radical to observe and displayed usually the most intense peak at $m/z = 134$ ($n = 1$ in eq 1, Figure 3D) but also significant ones at 152 ($n = 2$) and 170 ($n = 3$). In all cases, the peaks at $n = 0$ ($m/z = 48$ for $\text{CH}_3\text{O}_2$, $m/z = 76$ for $\text{CH}_3\text{C(O)O}_2$, $m/z = 90$ for $(\text{CH}_3)_3\text{CO}_2$, and $m/z = 116$ for $c\text{-C}_6\text{H}_{11}\text{O}_2$) were either not observed or strongly impacted by other species.

However, because NO was added at the top of the reactor, together with the organic precursors, it not only reacted with

DOI: 10.1021/acs.jpca.7b06456
J. Phys. Chem. A 2017, 121, 8453−8464

[Figure]

**Figure 3.** Mass spectra of the peroxy radicals: (A) $CH_3O_2$ (experiment of Sept 05, 2016); (B) $CH_3C(O)O_2$ (experiment of Dec 06, 2016); (C) $(CH_3)_3CO_2$ (experiment of May 10, 2017); (D) $c$-$C_6H_{11}O_2$ (experiment of June 14, 2016).

the $RO_2$ but also modified the product distribution. Thus, the major products formed in the absence of NO, such as ROOH, ROH, and ROOR, were also often visible in the difference spectra shown in Figure 3. Because ROOH is only 1 amu away from $RO_2$, variations of the corresponding signals $S_{ROOH}$ and $S_{RO2}$ were compared to check that $S_{RO2}$ was not impacted by ROOH. This was especially important for $CH_3C(O)O_2$ (Figure 3B), because the two peaks were not always entirely resolved. Typical $S_{ROOH}$ versus $S_{RO2}$ plots are presented in Figure S2 and show that, while these signals followed similar trends (as might be expected), they varied distinctively from each other and thus corresponded to distinct species (different signals corresponding to the same species would have appeared perfectly aligned).

In addition to the $RO_2$ produced directly by the precursors, the technique allowed for the detection of all other peroxy radicals present in the reactor. $CH_3O_2$ was thus observed in all

the mixtures where $CH_3C(O)O_2$ was present, which was expected, as it is produced rapidly by its self-reaction:[32]

$$2CH_3C(O)O_2 \rightarrow 2CH_3C(O)O + O_2 \tag{22}$$

$$2CH_3C(O)O + M \rightarrow CH_3 + CO_2 + M \tag{23}$$

$$CH_3 + O_2 + M \rightarrow CH_3O_2 + M \tag{24}$$

$CH_3O_2$ was also observed in the mixtures where $(CH_3)_3CO_2$ was present, as it is also produced by its self-reaction:

$$2(CH_3)_3CO_2 \rightarrow 2(CH_3)_3CO + O_2 \tag{25}$$

$$\rightarrow (CH_3)_3COOC(CH_3)_3 + O_2 \tag{26}$$

$$(CH_3)_3CO + M \rightarrow CH_3COCH_3 + CH_3 + M \tag{27}$$

followed by reaction 24 producing $CH_3O_2$.

Thus, in the investigation of the cross-reactions, where a second radical was added periodically to each radical system (see details in section d, below), up to three different $RO_2$ were observed simultaneously (Figure 4).

Occasionally, small amounts of other radicals due to contamination were observed in the reactor. For instance,

[Figure]

[Figure]

**Figure 4.** Simultaneous monitoring of three different $RO_2$ with the CIMS technique: (A) Real-time evolution of the signals for $CH_3O_2$ (black, $m/z = 84$), $CH_3C(O)O_2$ (red, $m/z = 94$), and $(CH_3)_3CO_2$ (blue, $m/z = 108$) upon periodic addition of $CH_3CHO$ to a system in which isobutane is present continuously (experiment of June 7, 2017). Note that the apparent decay of $CH_3C(O)O_2$ upon addition of $CH_3CHO$ is due to a change in the total signal; (B) Real-time evolution of the signals for $CH_3O_2$ (black, $m/z = 84$), $(CH_3)_3CO_2$ (blue, $m/z = 108$) and $c$-$C_6H_{11}O_2$ (red, $m/z = 134$) upon addition of various levels of NO in a system where both isobutane and c-hexane are present continuously (experiments of June 8, 2017).

DOI: 10.1021/acs.jpca.7b06456
J. Phys. Chem. A 2017, 121, 8453−8464

small amounts of c-$C_6H_{11}O_2$ were observed up to 2 d after performing experiments with cyclohexane. This had small impacts on the concentrations of the main radical studied, but the ability to detect unexpected radicals in the system confirmed the advantage of the chemical ionization mass spectrometry (CIMS) technique and its ability to investigate $RO_2$ chemistry in complex systems.

**b. Time Evolution of the Radicals: Decays with HO₂ and "Intensification Effects" at Low [NO].** Observing the evolution of the signals for the individual $RO_2$s as a function of time (or "Single Ion Mode") with a resolution of a few seconds revealed several kinetic effects, which gave some indications on the reactions controlling the $RO_2$ steady-state concentrations in the reactor and decay rates. One of these effects was that, for all the radicals studied except $CH_3C(O)O_2$, after stopping NO or switching the lamps on, $[RO_2]$ displayed a sharp increase followed by a fast decay, relaxing into a steady-state level after a few minutes (Figure 5A for c-$C_6H_{11}O_2$ and Figure S3 for $CH_3O_2$ and $(CH_3)_3O_2$). None of the stable reaction products monitored at the same time (ROOH, ROH, etc.; Figure 5A) displayed such decays, which ruled out flow or mixing effects, or changes in the detection performance as the explanation for these decays. The contrast between these fast decays and the steady profiles for the stable products (Figures 5A and S3) further confirmed that the signals had been correctly attributed between radicals and stable compounds. The sharp initial increase of $[RO_2]$ was attributed to the nearly instantaneous production of the radicals by irradiation and reactions of Cl with the organic precursors, and the following fast decay to their self-reaction and buildup of other species, in particular, $HO_2$, in turn reacting with the $RO_2$. Thus, for radicals such as $CH_3O_2$ and c-$C_6H_{11}O_2$, producing $HO_2$ in their self-reaction (eq 19), the decays and steady-state concentration was expected to be controlled by their self-reaction or/and by their reaction with $HO_2$. For $(CH_3)_3CO_2$ and $CH_3C(O)O_2$ (eqs 22−24 and 25−27), which do not produce $HO_2$ in their self-reaction but $CH_3O_2$, the decays and steady-state concentrations were expected to result from the self-reactions (mostly for $CH_3C(O)O_2$), cross-reactions with $CH_3O_2$, and reactions with the $HO_2$ produced by $CH_3O_2$. With $CH_3O_2$, $(CH_3)_3CO_2$, and c-$C_6H_{11}O_2$, the formation of stable products, in particular, of ROOH was observed over the same time scale as these decays (Figures 5A and S3), confirming that the $RO_2 + HO_2$ reactions took place. With $CH_3C(O)O_2$ no decays were observed, suggesting that they were too fast to be monitored ($>0.2\ s^{-1}$). The rate coefficient for the reaction of this radical with $HO_2$ being identical to that of other radicals,[33] this reaction could not account for such very fast decays. The decays and steady-state concentration of $CH_3C(O)O_2$ in the reactor were thus expected to be controlled by its self-reaction (which is the fastest known self-reaction of $RO_2$) and/or by its cross-reaction with $CH_3O_2$.

Another interesting kinetic effect was that, for all the radicals studied except $CH_3C(O)O_2$, adding very small amounts of NO to the reaction mixture resulted in larger $RO_2$ concentration than with $[NO] = 0$ ("intensification effect", Figure 5B for $(CH_3)_3CO_2$ and S4 for the other radicals). Only for larger amounts of NO the $RO_2$ started to be consumed. The intensification effects were attributed to NO consuming first (i.e., reacting faster with) the species acting as main sink for the $RO_2$s in the reactor, thus presumably $HO_2$. The net reduction of $[RO_2]$ observed at larger $[NO]$ was attributed to the consumption of all $HO_2$, leaving only $RO_2$ to react with NO.

[Figure]

[Figure]

[Figure]

**Figure 5.** Real-time, steady-state, and relative variations of the $RO_2$ signal in the reactor upon addition of $[NO]$. (A) Real-time decay for c-$C_6H_{11}O_2$ (black line, $m/z = 134$) due to $RO_2 + HO_2$, compared with the profiles for c-$C_6H_{11}OOH$ (blue line, $m/z = 135$), experiment of June 14, 2016. (B) Variations of $S_{RO2}$ with $[NO]$ and intensification effect for $(CH_3)_3CO_2$ ($m/z = 108$, experiment of May 11, 2017). (C) Comparison of the observed $\Delta S_{RO2}/S_{RO2}$ (red symbols) with calculated ones (blue symbols for complete equation, white symbols without the self-reaction) for the experiment shown in (B).

The absence of intensification effect with $CH_3C(O)O_2$ at low $[NO]$ was consistent with the fact that $HO_2$ was not a significant sink for this radical and that $CH_3O_2$, its expected main sink, did not react faster with NO than $CH_3C(O)O_2$ itself.

To verify these hypotheses and quantify the contribution of the different reactions to the radical concentrations in the reactor, $[RO_2]$ was calculated and compared to the experiments. Details on these zero-dimensional (0D) calculations are given in Section S5, and the rate constants used are given in Table S6A. Briefly, they included a source term, $F$ (molecule

DOI: 10.1021/acs.jpca.7b06456
*J. Phys. Chem. A* 2017, 121, 8453−8464

cm$^{-3}$ s$^{-1}$), accounting for the production of the RO$_2$ by irradiation and reaction of Cl with the precursors, first-order losses to the walls and in the exit flow (mathematically indistinguishable), the RO$_2$ self-reactions and reactions with HO$_2$, CH$_3$O$_2$, and NO, and an additional source term specific to the presence of NO, $F'_{NO}$ (molecule cm$^{-3}$ s$^{-1}$), accounting for the potential recycling of HO$_2$ into HO, in turn producing rapidly RO$_2$. Because, at that point of the analysis, the absolute concentrations of RO$_2$ were not known, the comparisons with the experiments were based on the relative signal change $\Delta S_{RO2}/S_{RO2} = ([RO_2]_o − [RO_2]_s)/[RO_2]_o$, where $[RO_2]_o$ and $[RO_2]_s$ are the radical steady-state concentrations in the absence and in the presence of NO, respectively. Note that, here, "steady-state" defines concentrations that are constant in time, due to the equilibration of the different contributions listed above (thus including source and exit flows, in addition to chemical terms). The values for $k_{wall}$, $F$, and $F'_{NO}$ were adjusted so that the calculated $\Delta S_{RO2}/S_{RO2}$ matched the observed ones (Figure 5C for (CH$_3$)$_3$CO$_2$ and Figure S7 for the other radicals), which provided values for these constants (given in Table S6B). The observed $\Delta S_{RO2}/S_{RO2}$ were well-accounted for with $F = 1 \times 10^8$ molecule cm$^{-3}$ s$^{-1}$ for CH$_3$O$_2$ and $1 \times 10^9$ for the other radicals, consistent with the much smaller reaction rate of Cl with CH$_4$ than with the other precursors. $k_{wall}$ was found to be identical for all RO$_2$s, with a value of ∼0.01 s$^{-1}$. The corresponding residence time, 100 s, was of the order of the residence time resulting from the flow rate in the reactor (30 s$^{-1}$ min), thus suggesting that the exit flow was a large component of $k_{wall}$. The calculations also showed that, for CH$_3$O$_2$, (CH$_3$)$_3$CO$_2$, and c-C$_6$H$_{11}$O$_2$, the self-reactions and reactions with HO$_2$ contributed relatively little (≤20%) to the steady-state concentrations and that, unlike what was expected, the suppression of HO$_2$ at low [NO] did not account for the observed intensification effects ($\Delta S_{RO2}/S_{RO2} < 0$ in Figures 5C and S7). To account for these effects it was necessary to introduce an additional source $F'_{NO}$, corresponding to the recycling of HO$_2$ into OH, in turn producing more RO$_2$. As a confirmation, the observed $\Delta S_{RO2}/S_{RO2}$ were best accounted for by scaling $F'_{NO}$ with [HO$_2$]. For CH$_3$O$_2$ and c-C$_6$H$_{11}$O$_2$, [HO$_2$] and $F'_{NO}$ varied inversely with [NO], and with (CH$_3$)$_3$CO$_2$, even more strongly (see Section S5). These additional sources thus resulted in an increase of the RO$_2$ sources by a factor of 2−5 over a very small range of very low [NO]. The threshold [NO] at which these effects disappeared and [RO$_2$]$_s$ started to decrease compared to [RO$_2$]$_o$ were estimated for each radical: 0.04 ppb for CH$_3$O$_2$, 0.3 ppb for (CH$_3$)$_3$CO$_2$, and 0.1 ppb for c-C$_6$H$_{11}$O$_2$ (Table S6B).

For CH$_3$C(O)O$_2$, the calculations showed that the self-reaction and cross-reaction with CH$_3$O$_2$ were the main contributions to the steady-state concentration, resulting in a weak variability $\Delta S_{RO2}/S_{RO2}$ with [NO] (Figure S7B). As the concentration of HO$_2$ was significant in this system, some recycling of HO$_2$ into OH and intensification effects could not be excluded, but the calculations showed that they would occur over a range of [NO] too low to be explored experimentally (≤0.01 ppb, Figure S7).

Above the threshold [NO] at which the intensification effects disappeared, most HO$_2$ was consumed, and the recycling sources became negligible. Thus, these recycling effects had no impact on the radical concentrations over the range of [NO] used in the determination of their detection sensitivities (next section).

The calculations allowed to determine the correction factors to apply to the observed $\Delta S_{RO2}$ to ensure equivalence with added [NO] in the titration experiments. Details are given in Section S5, and the results are presented in Table S6C. For the range of [NO] used in the titration experiments (next section), these corrections were mostly significant for CH$_3$O$_2$ (×0.4) and CH$_3$C(O)O$_2$ (×0.15) and much less (×1) for (CH$_3$)$_3$CO$_2$ or c-C$_6$H$_{11}$O$_2$. For all the radicals, they resulted mostly from the wall losses and reactions with CH$_3$O$_2$ (for CH$_3$C(O)O$_2$).

In addition to the experiments where NO was added to systems containing one or two RO$_2$, an experiment was performed where NO was added periodically to a system containing three different RO$_2$s: CH$_3$O$_2$, (CH$_3$)$_3$CO$_2$, and c-C$_6$H$_{11}$O$_2$ (Figure 4B). The mixture was similar as in the study of the cross-reaction between (CH$_3$)$_3$CO$_2$ and c-C$_6$H$_{11}$O$_2$ (see section d), except that both isobutane and c-C$_6$H$_{12}$ were introduced continuously in the reactor. The objective was to examine the evolution of each individual RO$_2$ upon addition of NO. Figure 4B clearly shows that, over a certain range, [NO] had opposite effects on the different RO$_2$s: small amounts of NO (∼0.3 ppb) resulted in intensification effects for all three radicals (last three cycles on the right-hand side), intermediate amounts (∼0.7 ppb, first four cycles) consumed (CH$_3$)$_3$CO$_2$ but increased the concentrations of CH$_3$O$_2$ and c-C$_6$H$_{11}$O$_2$. Finally, larger amounts of NO (∼1.4 ppb, cycles 5 and 6) consumed all three radicals. Although expected, such opposite effects of NO on RO$_2$s have, to our knowledge, not been observed directly before. This is mostly because indirect and conversion techniques monitor the sum of the RO$_2$ concentrations, in which opposite effects compensate each other. Such effects are yet likely to occur in complex mixtures such as in smog chambers or the atmosphere, which underlines the interest of the present technique for the monitoring of RO$_2$s.

**c. RO$_2$ Detection Sensitivities.** As described in the Materials and Methods section, the detection sensitivities for the RO$_2$s, Sens(RO$_2$), and RO$_2$ concentrations were determined by titrating the radicals with NO and measuring the decrease of [RO$_2$]$_s$ compared to [RO$_2$]$_o$. This required to use NO concentrations larger than the threshold reported above for the intensification effects: ∼0.1−0.5 ppb for CH$_3$O$_2$, 0.01−0.05 ppb for CH$_3$C(O)O$_2$, and 0.5−2 ppb both for (CH$_3$)$_3$CO$_2$ and c-C$_6$H$_5$O$_2$. In addition, the contributions to intensification effects and HO$_2$ reaction were kept minimal by reducing the RO$_2$ concentrations in the reactor, by limiting either the amount of Cl$_2$ in the reactor or irradiation (only one set of lamps on). As explained above, Sens(RO$_2$) was obtained from the observed change in signal, $\Delta S_{RO2}$, and the corresponding change in steady-state concentration, $\Delta[RO_2]$, assuming equivalence between $\Delta[RO_2]$ and added [NO]. [NO] was, in turn, determined from the rate of decay of the radical, $r_{RO2+NO}$ (eq 12). To eliminate potential contributions of wall losses and mixing effects, different [NO] were added to the reactor to confirm that both $r_{RO2+NO}$ and $\Delta S_{RO2}$ varied proportionally with [NO] (Figures 2B and S8 for all the titration experiments performed) and Sens(RO$_2$) was obtained from the slopes of $\Delta S_{RO2}$ versus [NO] rather than eq 11. In addition, the $\Delta S_{RO2}$ measured for CH$_3$O$_2$ and CH$_3$C(O)O$_2$ were corrected by the factors calculated in the previous section to ensure equivalence between $\Delta[RO_2]$ and added [NO]. The resulting sensitivities, calculated for a total reagent ion signal, $S_o = 1 \times 10^6$ Hz, are presented in Figure 6 and Table S9. They varied between 50 Hz/ppb for CH$_3$O$_2$ under humid conditions

DOI: 10.1021/acs.jpca.7b06456
J. Phys. Chem. A 2017, 121, 8453−8464

[Figure]

**Figure 6.** Detection sensitivities for the radicals as a function of the relative humidity in the ionization region, represented by the proxy, $S_{73}/S_{55}$. Black = $CH_3O_2$, $m/z = 84$; red = c-$C_6H_{11}O_2$, $m/z = 134$; blue = $(CH_3)_3CO_2$, $m/z = 126$; green = $CH_3C(O)O_2$, $m/z = 94$.

to ~5000 Hz/ppb for $CH_3C(O)O_2$ under dry conditions. These results can be compared with the radical theoretical sensitivity, calculated as in our previous work:[30]

$$TS = k_n t_d S_o \qquad (29)$$

where $k_n$ is the rate coefficient for the reaction between the radical and the reagent ion $H_3O^+(H_2O)_n$, $t_d$ is the ion drift time, and $S_o$ is the total ions signal, assumed equal to 1 MHz here. In our setup, for a drift length of 5 cm, a mobility of 130 cm/V and a voltage difference of 600 V, $t_d \approx 3 \times 10^{-4}$ s. In our previous work,[30] $k_n$ was estimated to $2 \times 10^{-9}$ molecules cm$^{-3}$ s$^{-1}$ for c-$C_6H_{11}O_2$ and decreasing with the radical molecular weight. This gives a maximum TS of 180 Hz/ppb for the $RO_2$s in the present work. Comparing with the sensitivities obtained experimentally thus suggested uncertainties of a factor 3 for c-$C_6H_{11}O_2$ and $(CH_3)_3CO_2$, and of approximately a factor 5 for $CH_3O_2$ and for most values obtained for $CH_3C(O)O_2$ (if ignoring the measurement at 5000 Hz/ppb). Of these uncertainties, up to approximately a factor 2, can possibly be attributed to the water proton signals exceeding the linear range of the electron multiplier, leaving the remaining uncertainties on the determination of [NO], thus the measurements of the decay rates. Both types of uncertainties can be improved in the future by reducing the total proton signal and by using a sensitive technique to monitor directly the small amounts of [NO] added to the reactor.

In spite of these large uncertainties, the results indicate a decrease of sensitivity with water vapor for all radicals (Figure 6), consistent with the one reported previously for $CH_3O_2$.[30]

Using these calibrations gave estimates for the $RO_2$ concentrations in the flow reactor between $1 \times 10^{10}$ and $1 \times 10^{12}$ molecules cm$^{-3}$. A typical background signal of 100 Hz in the instrument implied a detection limit between $1 \times 10^8$ and $1 \times 10^9$ molecules cm$^{-3}$ for the radicals, similar to previous studies.[30]

**d. Kinetics of Cross Reactions.** As explained in "Material and Methods" the cross-reactions between different radicals were investigated by producing one radical, $R_1O_2$, continuously in the reactor and producing the second one, $R_2O_2$, periodically by switching the flow of the corresponding precursor on and off. For the first time, these cross-reactions could be observed directly by monitoring in real-time the decrease of $R_1O_2$ upon addition of $R_2O_2$ (Figure 7), and the rate coefficient $k_{cross}$ could be determined from the corresponding decay rate $r_{R1O2+R2O2}$ (s$^{-1}$) instead of having to extract the individual information

from complex signals, as with nonspeciated detection techniques. The concentration of added $[R_2O_2]$ was obtained from the signal change $\Delta S_{R1O2}$ and the detection sensitivity for $R_1O_2$ obtained in the previous section. As in the titration experiments, various $[R_2O_2]$ were added to the reactor to verify that $r_{R1O2+R2O2}$ varied proportionally (Figure S10), and $k_{cross}$ was determined from the slope of $r_{R1O2}$ versus $\Delta[R_1O_2]$ (Figure S10) rather than eq 13 to eliminate contributions of wall losses, mixing, and any other processes not varying with $[R_2O_2]$.

As in the reactions with NO, the radical concentrations had been lowered in the reactor to minimize the secondary reactions. In addition, to ensure equivalence between $\Delta[R_1O_2]$ and added $[R_2O_2]$ in the analyses, the correction factors to apply to the measured $\Delta S_{R1O2}$ were determined by performing similar calculations as for the titration experiments (Section S11): $\Delta S_{R1O2}/S_{R1O2}$ was calculated for various values of added $[R_2O_2]$ and compared with the experimental data. The results are presented in Figure S12. Very good agreements were obtained in all cases when using the constants determined in the previous calculations ($k_{wall}$ and source term $F$). The correction factors $x$ to apply to the observed $\Delta S_{R1O2}$, $(\Delta S_{R1O2})_{obs}$, were determined as in the titration experiments, by calculating the $\Delta S_{R1O2}$ corresponding to equivalence, $(\Delta S_{R1O2})_{eq}$, by excluding the contributions of all reactions other than the wall losses (mostly, exit flow) and cross-reactions of interest. The correction factors obtained are given in Table S6C. For all the radicals and range of concentrations used in these experiments, they were similar to the correction factors used in the titration experiments (0.4 for $CH_3O_2$, 0.15 for $CH_3C(O)O_2$, and 1 for $(CH_3)_3CO_2$ and c-$C_5H_{11}O_2$). This was expected, as they resulted from the same reactions: wall losses (exit flow) and reactions with $CH_3O_2$. In each series of experiments, the average value for the rate coefficient was thus first determined by applying a linear regression to all the individual measurements of $r_{R1O2}$ versus corrected $\Delta[R_1O_2]$. Then, compensating each measurement point for the slope and intercept of these regressions gave series of individual determinations for the same rate coefficients, presented in Table 1, which were used to estimate the statistical uncertainties in the results. The average values for the two coefficients that were already known, namely, $k_{CH3O2+CH3C(O)O2}$ and $k_{CH3O2+(CH3)3CO2}$, were in very good agreement with those recommended in the literature (see Table 1). In addition, the 17 individual measurements of $k_{CH3O2+CH3C(O)O2}$ and 11 individual measurements of $k_{CH3O2+(CH3)3CO2}$ were within 20 and 35% of the literature values. Although the other rate coefficients, $k_{c\text{-hexylO2+CH3O2}}$, $k_{t\text{-butylO2+CH3C(O)O2}}$, and $k_{c\text{-hexylO2+t\text{-butylO2}}}$, have not been reported before, to our knowledge, their average values were close to their expected values of $k_{cross} \approx \sqrt{(k_{self1} \times k_{self2})}$,[32] and the statistical dispersion on their 5 to 7 individual measurements were of 50% or less. These uncertainties are surprisingly small compared to the large uncertainties in the detection sensitivities ($\times 5/5$) and can be possibly attributed to some compensation between systematic errors, as shown by replacing eqs 11 and 12 in eq 13:

$$k_{cross} = \frac{\Delta S'_{R1O2}}{\Delta S_{R1O2}} \times \frac{r_{R1O2+R2O2}}{r_{R1O2+NO}} \times k_{R1O2+NO} \qquad (30)$$

For instance, it was shown above that the correction factors on the $\Delta S_{RO2}$ were identical in the titration and cross-reaction experiments because they resulted from the same reactions. It is

DOI: 10.1021/acs.jpca.7b06456
J. Phys. Chem. A 2017, 121, 8453−8464

[Figure]

**Figure 7.** Real-time evolution of the signals for the different RO$_2$s in the cross-reactions experiments. Blue lines = R$_1$O$_2$ continuously present; red lines = R$_2$O$_2$, periodically added. (A) CH$_3$O$_2$ (red, m/z = 84) + CH$_3$C(O)O$_2$ (blue, m/z = 94), experiment of Oct 01, 2015; (B) CH$_3$O$_2$ (red, m/z = 84) + (CH$_3$)$_3$CO$_2$ (blue, m/z = 108), experiment of May 12, 2017; (C) CH$_3$O$_2$ (blue, m/z = 66) + c-C$_6$H$_{11}$O$_2$ (red, m/z = 134), experiment of June 02, 2017; (D) CH$_3$C(O)O$_2$ (red, m/z = 94) + (CH$_3$)$_3$CO$_2$ (blue, m/z = 108), experiment of June 07, 2017; (E) c-C$_6$H$_{11}$O$_2$ (red, m/z = 134) + (CH$_3$)$_3$CO$_2$ (blue, m/z = 108), experiment of June 08, 2017.

possible that, for instance, the decay rates are also systematically underestimated by not properly taking into account some mixing effects, which would result in similar errors in the titration and cross-reaction experiments, canceling each other out in eq 30. However, direct measurements of [NO] in the experiments and further model calculations, such as two-dimensional fluid dynamics reactor analysis, would be needed to identify the source of error in the current determinations of the RO$_2$ sensitivities and quantify those in the decay rate measurements. Until these uncertainties are identified and quantified, we recommend the same uncertainties in the reported rate coefficients as in the detection sensitivities, thus ×5/5.

**■ CONCLUSIONS**

This work illustrates the performance of the CIMS with proton transfer for the speciated detection of RO$_2$ in complex systems. Reported for the first time, two to three different RO$_2$ were monitored simultaneously, and their cross-reactions could be observed directly from the decay of one radical when a second one was produced. The rate coefficients for these cross-reactions were determined directly from the individual signals, instead of being extracted from overall signals with kinetic models, as with nonspeciated detection techniques. This opens the possibility for important improvements in the investigation of the kinetics of these species in the laboratory and ultimately in the understanding of the radical cycles in the atmosphere and other systems, by reducing the number of unknowns in the observations. The main uncertainties in this work were those

DOI: 10.1021/acs.jpca.7b06456
J. Phys. Chem. A 2017, 121, 8453−8464

**Table 1. Rates Constants for the Cross-Reactions Measured in This Work**

| | $R_1O_2$ $m/z$ | $S_{73}/S_{55}$ | $Sens_{R1O2}$ (Hz/ppb) | $k_{cross}$ (molecule$^{-1}$ cm$^3$ s$^{-1}$) | lit values |
|---|---|---|---|---|---|
| $CH_3O_2 + CH_3C(O)O_2$ | $CH_3C(O)O_2$ | 0.6 | 750 | $1.42 \times 10^{-11}$ | |
| | | | | $1.33 \times 10^{-11}$ | |
| | | | | $1.14 \times 10^{-11}$ | |
| | | | | $1.08 \times 10^{-11}$ | |
| | | | | $1.01 \times 10^{-11}$ | |
| Oct 01, 2015 | | | | ave = $1.20 \times 10^{-11}$ | |
| June 06, 2017 | | 0.07 | 750 | $8.70 \times 10^{-12}$ | |
| | | | | $1.53 \times 10^{-11}$ | |
| | | | | $1.18 \times 10^{-11}$ | |
| | | | | $1.66 \times 10^{-11}$ | |
| | | | | $1.32 \times 10^{-11}$ | |
| | | | | ave $1.31 \times 10^{-11}$ | |
| | 94 | | | | |
| June 07, 2017 | | 0.13 | 5100 | $1.02 \times 10^{-11}$ | |
| | | | | $1.55 \times 10^{-11}$ | |
| | | | | $5.53 \times 10^{-12}$ | |
| | | | | $1.34 \times 10^{-11}$ | |
| | | | | $8.56 \times 10^{-12}$ | |
| | | | | $1.10 \times 10^{-11}$ | |
| | | | | $1.03 \times 10^{-11}$ | |
| | | | | ave = $1.06 \times 10^{-11}$ | $(1.1 \pm 0.3) \times 10^{-11}$ ref [33] |
| $CH_3O_2 + (CH_3)_3CO_2$ | $(CH_3)_3CO_2$ | | | | |
| May 12, 2017 | 108 | 0.25 | 20 | $2.71 \times 10^{-15}$ | |
| | | | | $3.28 \times 10^{-15}$ | |
| | | | | $3.79 \times 10^{-15}$ | |
| | | | | $2.74 \times 10^{-15}$ | |
| | | | | $3.00 \times 10^{-15}$ | |
| | | | | $3.60 \times 10^{-15}$ | |
| | | | | ave $3.2 \times 10^{-15}$ | |
| May 16, 2017 | 126 | 15 | 2 | $2.45 \times 10^{-15}$ | |
| | | | | $1.93 \times 10^{-15}$ | |
| | | | | $2.45 \times 10^{-15}$ | |
| | | | | $4.38 \times 10^{-15}$ | |
| | | | | $3.14 \times 10^{-15}$ | |
| | | | | ave = $2.9 \times 10^{-15}$ | $(3.0 \pm 0.3) \times 10^{-15}$ ref [32] |
| $CH_3O_2 + c\text{-}C_6H_{11}O_2$ | $CH_3O_2$ 66 | 0.08 | 160 | $8.94 \times 10^{-14}$ | |
| June 02, 2017 | | | | $2.50 \times 10^{-13}$ | |
| | | | | $8.96 \times 10^{-14}$ | |
| | | | | $1.22 \times 10^{-13}$ | |
| | | | | $6.17 \times 10^{-14}$ | |
| | | | | ave = $1.2 \times 10^{-13}$ | |
| $CH_3C(O)O_2 + (CH_3)_3CO_2$ | $(CH_3)_3CO_2$ 126 | 1 | 250 | $3.08 \times 10^{-14}$ | |
| June 07, 2017 | | | | $2.76 \times 10^{-14}$ | |
| | | | | $3.74 \times 10^{-14}$ | |
| | | | | $3.75 \times 10^{-14}$ | |
| | | | | $5.09 \times 10^{-14}$ | |
| | | | | ave = $3.7 \times 10^{-14}$ | |
| $(CH_3)_3CO_2 + c\text{-}C_6H_{11}O_2$ | $(CH_3)_3CO_2$ 108 | 0.17 | 10 | $3.10 \times 10^{-16}$ | |
| June 08, 2017 | | | | $8.78 \times 10^{-16}$ | |
| | | | | $1.80 \times 10^{-15}$ | |
| | | | | $3.53 \times 10^{-15}$ | |
| | | | | $1.05 \times 10^{-15}$ | |
| | | | | ave = $1.5 \times 10^{-15}$ | |

on the detection sensitivities estimated for the radicals (×5/5), which were mostly due to the difficulties in knowing the amounts of NO added to the reactor and in the determination of the radical decay rates. These uncertainties can be readily solved by experimental changes, such as measuring directly the small NO concentrations with a sensitive technique and by replacing time-dependent measurements by static methods, such as using a movable injector to introduce one radical precursor and measuring the changes in the steady-state concentrations. Until the source for these uncertainties has been identified, similarly large uncertainties (×5/5) are recommended on the reported rate coefficients, in spite of

DOI: 10.1021/acs.jpca.7b06456
J. Phys. Chem. A 2017, 121, 8453−8464

their apparent good comparison with the literature and good reproducibility.

One advantage of the technique used in this work was to monitor in real time individual RO$_2$ radicals and their nonsteady-state kinetic effects (fast increase and decays in RO$_2$ concentrations). Another advantage was to monitor simultaneously different RO$_2$s and observe directly their opposite behaviors (increase or decrease in steady-state [RO$_2$]) at low [NO]. Many such effects are likely to take place in complex systems, such as the atmosphere, but are currently undetected by the nonspeciated (conversion) techniques. The CIMS technique would thus be a valuable tool to study RO$_2$ in more complex mixtures than in flow reactors, such as smog chambers, where it would greatly improve the understanding of the radical mechanisms and validation of models.

**■ ASSOCIATED CONTENT**

**ⓈSupporting Information**

The Supporting Information is available free of charge on the ACS Publications website at DOI: 10.1021/acs.jpca.7b06456.

> List of the experiments presented in this manuscript. Comparisons of the signals at the RO$_2$ masses, $S_{RO2}$, with those at the ROOH masses, $S_{ROOH}$. Fast decays of the individual RO$_2$ observed after stopping the NO flow in the reactor. Steady-state [RO$_2$] increase and decrease upon addition of NO in the reactor. Calculation of $\Delta S_{RO2}/S_{RO2}$ as a function of NO in the reactor. List of the kinetic constants used in the calculations of $\Delta S_{RO2}/S_{RO2}$ as a function of [NO]. Comparison between observed and calculated $\Delta S_{RO2}/S_{RO2}$ as a function of [NO] added in the reactor. $\Delta S_{RO2}$ versus [NO] plots used for the calibration of the detection sensitivities. Results of the detection sensitivity calibrations. $r_{R1O2}$ versus $\Delta S_{RO2}$ plots used for the determination of the rate coefficients for the cross-reactions. Calculations of $\Delta S_{R1O2}/S_{R1O2}$ as a function of [R$_2$O$_2$] in the cross-reactions. Calculated $\Delta S_{R1O2}/S_{R1O2}$ as a function of added [R$_2$O$_2$] in the cross reactions (PDF)

**■ AUTHOR INFORMATION**

**Corresponding Author**

*E-mail: barbara.noziere@ircelyon.univ-lyon1.fr.
Barbara Nozière: 0000-0001-5841-1310
David R. Hanson: 0000-0001-5719-377X

**Author Contributions**

The manuscript was written through contributions of all authors. All authors have given approval to the final version of the manuscript. B.N. helped to build the CIMS instrument used in the experiments, performed the experiments and analyses, and wrote the manuscript. D.H. designed and built the CIMS instrument and helped with writing some experimental parts of the manuscript (e.g., CIMS detection and highlighting experimental detail) and contributed to the discussion of the analysis.

**Notes**

The authors declare no competing financial interest.

**■ ACKNOWLEDGMENTS**

This work was funded by the National Science Foundation, Grant No. NSF-ATM 0232057, which is gratefully acknowledged by the authors. B.N. also warmly thanks the many people whose valuable help have made this work possible: L. Zink (then at NCAR, Boulder, CO) for building the CIMS instrument, M. Grasmueck (Univ. of Miami, FL) for transporting it to Univ. of Miami, and W. Esteve (then at Univ. of Miami) for setting it up, and M. Dupanloup (CNRS, Ircelyon, France) is gratefully thanked for repairing the CIMS detection system. Special thanks are due to H. Wegmann (Wegmann Marin) for transporting the instrument to Stockholm Univ., Sweden, and CNRS, France, helping to set it up, and for his constant support throughout this long project. Conversations with J. Orlando and G. Tyndall, NCAR, Boulder, CO, are also gratefully acknowledged.

**■ REFERENCES**

(1) Mihelcic, D.; Musgen, P.; Ehhalt, D. H. An improved method of measuring tropospheric NO$_2$ and RO$_2$ by matrix-isolation and electron-spin-resonance. *J. Atmos. Chem.* **1985**, 3, 341−361.

(2) Fuchs, H.; Brauers, T.; Haseler, R.; Holland, F.; Mihelcic, D.; Musgen, P.; Rohrer, F.; Wegener, R.; Hofzumahaus, A. Intercomparison of peroxy radical measurements obtained at atmospheric conditions by laser-induced fluorescence and electron spin resonance spectroscopy. *Atmos. Meas. Tech.* **2009**, 2, 55−64.

(3) Orlando, J. J.; Tyndall, G. S. Laboratory studies of organic peroxy radical chemistry: an overview with emphasis on recent issues of atmospheric significance. *Chem. Soc. Rev.* **2012**, 41, 6294−6317.

(4) Dillon, T. J.; Crowley, J. N. Direct detection of OH formation in the reactions of HO$_2$ with CH$_3$C(O)O$_2$ and other substituted peroxy radicals. *Atmos. Chem. Phys.* **2008**, 8, 4877−4889.

(5) Peeters, J.; Nguyen, T. L.; Vereecken, L. HOx radical regeneration in the oxidation of isoprene. *Phys. Chem. Chem. Phys.* **2009**, 11, 5935−5939.

(6) Jenkin, M. E.; Hurley, M. D.; Wallington, T. J. Investigation of the radical product channel of the CH$_3$OCH$_2$O$_2$ + HO$_2$ reaction in the gas phase. *J. Phys. Chem. A* **2010**, 114, 408−416.

(7) Peeters, J.; Muller, J.-F. HOx radical regeneration in isoprene oxidation via peroxy radical isomerisations. II: experimental evidence and global impact. *Phys. Chem. Chem. Phys.* **2010**, 12, 14227−14235.

(8) Cantrell, C. A.; Stedman, D. H.; Wendel, G. J. Measurement of atmospheric peroxy-radicals by chemical amplification. *Anal. Chem.* **1984**, 56, 1496−1502.

(9) Fuchs, H.; Holland, F.; Hofzumahaus, A. Measurement of tropospheric RO$_2$ and HO$_2$ radicals by a laser-induced fluorescence instrument. *Rev. Sci. Instrum.* **2008**, 79, 12.

(10) Whalley, L. K.; Blitz, M. A.; Desservettaz, M.; Seakins, P. W.; Heard, D. E. Reporting the sensitivity of laser-induced fluorescence instruments used for HO$_2$ detection to an interference from RO$_2$ radicals and introducing a novel approach that enables HO$_2$ and certain RO$_2$ types to be selectively measured. *Atmos. Meas. Tech.* **2013**, 6, 3425−3440.

(11) Hanke, M.; Uecker, J.; Reiner, T.; Arnold, F. Atmospheric peroxy radicals: ROXMAS, a new mass-spectrometric methodology for speciated measurements of HO$_2$ and sigma RO$_2$ and first results. *Int. J. Mass Spectrom.* **2002**, 213, 91−99.

(12) Reiner, T.; Hanke, M.; Arnold, F. Atmospheric peroxy radical measurements by ion molecule reaction mass spectrometry: A novel analytical method using amplifying chemical conversion to sulfuric acid. *J. Geophys. Res.* **1997**, 102, 1311−1326.

(13) Lelieveld, J.; Butler, T. M.; Crowley, J. N.; Dillon, T. J.; Fischer, H.; Ganzeveld, L.; Harder, H.; Lawrence, M. G.; Martinez, M.; Taraborrelli, D.; et al. Atmospheric oxidation capacity sustained by a tropical forest. *Nature* **2008**, 452, 737−740.

(14) Hofzumahaus, A.; Rohrer, F.; Lu, K. D.; Bohn, B.; Brauers, T.; Chang, C. C.; Fuchs, H.; Holland, F.; Kita, K.; Kondo, Y.; et al.

Amplified trace gas removal in the troposphere. *Science* **2009**, *324*, 1702−1704.

(15) Onel, L.; Brennan, A.; Seakins, P. W.; Whalley, L.; Heard, D. E. A new method for atmospheric detection of the $CH_3O_2$ radical. *Atmos. Meas. Technol. Discuss.* **2017**, *2017*, 1−20.

(16) Eberhard, J.; Villalta, P. W.; Howard, C. J. Reaction of isopropyl peroxy radicals with NO over the temperature range 201−401 K. *J. Phys. Chem.* **1996**, *100*, 993−997.

(17) Villalta, P. W.; Howard, C. J. Direct kinetics study of the $CH_3C(O)O_2$ + NO reaction using chemical ionization mass spectrometry. *J. Phys. Chem.* **1996**, *100*, 13624−13628.

(18) Kondow, T. Ionization of clusters in collision with high-Rydberg rare gas atoms. *J. Phys. Chem.* **1987**, *91*, 1307−1316.

(19) Villalta, P. W.; Huey, L. G.; Howard, C. J. A temperature-dependent kinetics study of the $CH_3O_2$ + NO reaction using chemical ionization mass spectrometry. *J. Phys. Chem.* **1995**, *99*, 12829−12834.

(20) Jokinen, T.; Sipilä, M.; Richters, S.; Kerminen, V.-M.; Paasonen, P.; Stratmann, F.; Worsnop, D.; Kulmala, M.; Ehn, M.; Herrmann, H.; et al. Rapid autoxidation forms highly oxidized $RO_2$ radicals in the atmosphere. *Angew. Chem., Int. Ed.* **2014**, *53*, 14596−14600.

(21) Berndt, T.; Richters, S.; Kaethner, R.; Voigtländer, J.; Stratmann, F.; Sipilä, M.; Kulmala, M.; Herrmann, H. Gas-phase ozonolysis of cycloalkenes: formation of highly oxidized $RO_2$ radicals and their reactions with NO, $NO_2$, $SO_2$, and other $RO_2$ radicals. *J. Phys. Chem. A* **2015**, *119*, 10336−10348.

(22) Mentel, T. F.; Springer, M.; Ehn, M.; Kleist, E.; Pullinen, I.; Kurtén, T.; Rissanen, M.; Wahner, A.; Wildt, J. Formation of highly oxidized multifunctional compounds: autoxidation of peroxy radicals formed in the ozonolysis of alkenes − deduced from structure−product relationships. *Atmos. Chem. Phys.* **2015**, *15*, 6745−6765.

(23) Richters, S.; Herrmann, H.; Berndt, T. Highly oxidized $RO_2$ radicals and consecutive products from the ozonolysis of three sesquiterpenes. *Environ. Sci. Technol.* **2016**, *50*, 2354−2362.

(24) Richters, S.; Pfeifle, M.; Olzmann, M.; Berndt, T. Endo-cyclization of unsaturated $RO_2$ radicals from the gas-phase ozonolysis of cyclohexadienes. *Chem. Commun.* **2017**, *53*, 4132−4135.

(25) Scholtens, K. W.; Messer, B. M.; Cappa, C. D.; Elrod, M. J. Kinetics of the $CH_3O_2$+NO reaction: Temperature dependence of the overall rate constant and an improved upper limit for the $CH_3ONO_2$ branching channel. *J. Phys. Chem. A* **1999**, *103*, 4378−4384.

(26) Ranschaert, D. L.; Schneider, N. J.; Elrod, M. J. Kinetics of the $C_2H_5O_2$+$NO_x$ reactions: Temperature dependence of the overall rate constant and the $C_2H_5ONO_2$ branching channel of $C_2H_5O_2$+NO. *J. Phys. Chem. A* **2000**, *104*, 5758−5765.

(27) Elrod, M. J.; Ranschaert, D. L.; Schneider, N. J. Direct kinetics study of the temperature dependence of the $CH_2O$ branching channel for the $CH_3O_2$+$HO_2$ reaction. *Int. J. Chem. Kinet.* **2001**, *33*, 363−376.

(28) Chow, J. M.; Miller, A. M.; Elrod, M. J. Kinetics of the $C_3H_7O_2$+NO reaction: Temperature dependence of the overall rate constant and the i-$C_3H_7ONO_2$ branching channel. *J. Phys. Chem. A* **2003**, *107*, 3040−3047.

(29) Miller, A. M.; Yeung, L. Y.; Kiep, A. C.; Elrod, M. J. Overall rate constant measurements of the reactions of alkene-derived hydroxyalkylperoxy radicals with nitric oxide. *Phys. Chem. Chem. Phys.* **2004**, *6*, 3402−3407.

(30) Hanson, D.; Orlando, J.; Noziere, B.; Kosciuch, E. Proton transfer mass spectrometry studies of peroxy radicals. *Int. J. Mass Spectrom.* **2004**, *239*, 147−159.

(31) Hanson, D. R.; Greenberg, J.; Henry, B. E.; Kosciuch, E. Proton transfer reaction mass spectrometry at high drift tube pressure. *Int. J. Mass Spectrom.* **2003**, *223*, 507−518.

(32) Lightfoot, P. D.; Cox, R. A.; Crowley, J. N.; Destriau, M.; Hayman, G. D.; Jenkin, M. E.; Moortgat, G. K.; Zabel, F. Organic peroxy-radicals - Kinetics, spectroscopy and tropospheric chemistry. *Atmos. Environ., Part A* **1992**, *26*, 1805−1961.

(33) Atkinson, R.; Baulch, D. L.; Cox, R. A.; Crowley, J. N.; Hampson, R. F.; Hynes, R. G.; Jenkin, M. E.; Rossi, M. J.; Troe, J.; et al. Evaluated kinetic and photochemical data for atmospheric chemistry: Volume II - gas phase reactions of organic species. *Atmos. Chem. Phys.* **2006**, *6*, 3625−4055.

DOI: 10.1021/acs.jpca.7b06456
*J. Phys. Chem. A* 2017, 121, 8453−8464

---

## Author Comment (AC1) · 4 Mar 2019

**Authors' response to discussion comment by Barbara Nozière on:** Jenkin et al., Atmos. Chem. Phys. Discuss., https://doi.org/10.5194/acp-2019-44.

We are very grateful to Barbara Nozière for contributing a comment to the discussion of our paper, and for reminding us of the paper by Nozière and Hanson (2017) reporting speciated monitoring of organic peroxy radicals using CIMS, and consideration of a series of cross-reactions of different peroxy radicals.

As indicated in Barbara Nozière's comment, there were already kinetics studies of a number of peroxy radical cross-reactions (summarized in Table 11 of our paper), based on UV absorption detection. Although those studies were complicated by overlap of the peroxy radical absorption spectra, they were nonetheless direct measurements based on observation of the time-dependence of the peroxy radical absorptions at different wavelengths. In practice, the majority of the kinetics studies of peroxy radical self-reactions, cross-reactions and reactions with $HO_2$ are based on this type of measurement, which collectively form a substantial and invaluable data base.

When we became aware of the Nozière and Hanson (2017) study during the course of our work, we were encouraged that the kinetics data using their alternative technique were reported to endorse the existing data, although we noted that the rate coefficients were also reported to have very large uncertainties of a factor of $\times 5/5$. The data were therefore not factored into our preferred values at the time, and we were hoping that there would be subsequent studies reported with a more optimistic assessment of uncertainties (e.g. with the improvements outlined by Nozière and Hanson in their Conclusions section). As a result, we failed to discuss or cite this study in our paper, which was an oversight. This would be corrected in a revised manuscript.

Prompted by Barbara Nozière's comment, we have reviewed the Nozière and Hanson paper again. We are reminded that kinetic data are actually reported for one previously unstudied cross reaction ($c\text{-}C_6H_{11}O_2 + t\text{-}C_4H_9O_2$) which should have been considered in our work. However, we have also noted that one measurement ($CH_3C(O)O_2 + t\text{-}C_4H_9O_2$) is in substantial disagreement with existing data, and we would very much value Barbara Nozière's opinion on this. Summarising the results:

(i)    The Nozière and Hanson rate coefficients for the $CH_3C(O)O_2 + CH_3O_2$ and $CH_3O_2 + t\text{-}C_4H_9O_2$ reactions are in very good agreement with our tabulated data for these reactions, which are based on the IUPAC task group recommendations. That for the $CH_3O_2 + c\text{-}C_6H_{11}O_2$ reaction is also in agreement with our tabulated value, based on Villenave and Lesclaux (1996). We would make these points in a revised manuscript. We note that Nozière and Hanson (2017) were apparently unaware of the Villenave and Lesclaux (1996) determination.

(ii)    As indicated above, we overlooked the new data for the $c\text{-}C_6H_{11}O_2 + t\text{-}C_4H_9O_2$ reaction, and this would be considered in a revised manuscript.

(iii)    The Nozière and Hanson rate coefficient for the $CH_3C(O)O_2 + t\text{-}C_4H_9O_2$ reaction ($3.7 \times 10^{-14}$ cm$^3$ molecule$^{-1}$ s$^{-1}$) is a factor of 300 lower than that reported by Villenave et al. (1998) ($1.1 \times 10^{-11}$ cm$^3$ molecule$^{-1}$ s$^{-1}$). It also challenges the main conclusion of the Villenave et al. work, that the rate coefficients for all $CH_3C(O)O_2 + RO_2$ reactions are approximately $10^{-11}$ cm$^3$ molecule$^{-1}$ s$^{-1}$, which has been widely adopted in mechanism development and modelling studies. Villenave et al. (1998) reached this conclusion by studying the reactions of $CH_3C(O)O_2$ with a series of primary, secondary and tertiary $RO_2$ with widely different self-reaction reactivities. Nozière and Hanson (2017) were apparently unaware of the Villenave et al. (1998) determination, and therefore did not discuss this large disagreement and its wider implications. As indicated above, we would like to take advantage of this discussion to seek Barbara Nozière's opinion regarding possible reasons for this large discrepancy.

**References**

Nozière, B. and Hanson, D. R.: Speciated monitoring of gas-phase organic peroxy radicals by chemical ionization mass spectrometry: cross-reactions between $CH_3O_2$, $CH_3(CO)O_2$, $(CH_3)_3CO_2$, and $c$-$C_6H_{11}O_2$. J. Phys. Chem. A, 121, 8453-8464, 2017.

Villenave , E., and Lesclaux, R.: Kinetics of the cross reactions of $CH_3O_2$ and $C_2H_5O_2$ radicals with selected peroxy radicals, J. Phys. Chem., 100, 14372-14382, 1996.

Villenave , E., Lesclaux, R., Seefeld, S., and Stockwell, W. R.: Kinetics and atmospheric implications of peroxy radical cross reactions involving the $CH_3C(O)O_2$ radical, J. Geophys. Res., 103 (D19), 25273-25285, 1998.

---

## Referee Comment (RC1) · Anonymous Referee #1 · 5 Mar 2019

This manuscript discusses structure-activity relationships for peroxy radicals with its most common co-reactants in atmospheric conditions. The SARs are developed based on a selection of the available literature (mostly experimental data), and aim to provide site-specific rate coefficients and product distributions as appropriate for the reactions studied. The derivation of the SARs is well developed and explained, and the SARs strike a good balance between covering the mechanistic aspects of the target reactions on the one hand, and a pragmatic approach fitting data to a suitable function on the other hand, with good recovery of the training set. The data used as the training set

is not an exhaustive literature tabulation. Some experimental data is missing (see also the comment by B. Nozière), and while some theoretical data is used, the potential of combing theoretical and experimental data has not been fully exploited. Overall, however, I feel that reasonable choices were made, giving a good summary of the reactivity trends discernible from the literature data, even if one could have a different view on what data to include in the training set, what weight to assign to each datum (which is not all that obvious especially for theoretical data at lower levels of methodology), or how to parameterize the SAR. What was missing a bit in places is reference to existing SARs and their approaches, but I recognize this paper is focused on presenting a new SAR, and need not be made longer by rigorous review or historic overview.

To put the usability of the SARs to the test, I have applied them in the development of a small mechanism (∼100 reactions). The SARs prove to be quite usable even with a simple calculator, though during these efforts I found that adding a few additional subheadings would have made it easier to locate the desired information in the text: e.g. rate coefficients vrs. product distribution; self-reactions versus cross-reactions versus product distributions, etc.

Overall, this paper presents a good overview of the status quaestionis, and presents a set of very valuable SARs. Publication of the paper after minor revisions is recommended.

Specific comments:

p. 3, line 22: The generic rate coefficient for RO2 + NO is appropriate for many peroxy radicals, but RO2 derived from aromatics have been reported to have slightly higher rate coefficients. The difference may not warrant a different class, but a short mention might be useful.

p. 4, line 17: State explicitly (again) that nCON does not include the peroxy radical oxygen atoms, as an equally logical choice could have been a nCON based on the full molecular stoichiometry, i.e. including all functionalities. It might be useful to have a

short reminder in other places as well.

p. 4: The parameterization of the nitrate yield may need to be updated soon following recent work of John Orlando et al (NCAR). No publication is available to my knowledge, but interesting results were presented at conferences; I suggest contacting these authors to see if there is a need for alternative SAR parameters.

p. 7, line 13: formation of CI from $CH_3O_2$ + OH: Also state that the small to negligible yield of CI is consistent with theoretical data.

p. 8, R6c and R6e: R-HO is perhaps better written as R-H=O, unless the authors mean to imply that the H-atom transferred is not necessarily adjacent to the peroxy radical group.

p. 10: readability might be improved if using a notation for kRO2RO2 that indicates whether an expression pertains to self-reactions vrs. cross-reactions. Additional sub-headings might be useful to make finding specific topics easier when applying the SAR (reference self reactions, self reactions, cross reactions, branching ratios,...).

p. 11: line 29: "... if the peroxy radical contains more than one benzyl group". A benzyl group is $C_6H_5$-C.$H_2$, and there can be only one. The authors probably mean multiple beta-phenyl groups?

p. 11, line 29: the formula for calculating alpha and beta needs an equation number to allow unambiguous references in implementations.

p. 12: line 7: "This is regarded as a logical choice, because $CH_3O_2$ is the most abundant organic peroxy radical in the atmosphere". An explicit or semi-explicit mechanism as seems to be the target here is not used all that often for global modeling or even regional modeling as they tend to be too large. Without having access to any reliable statistical data, I would guess that e.g. the MCM is more often used to model specific experiments such as environmental chambers or lab studies, where $CH_3O_2$ is not necessarily the dominant proxy radical, if it is present at all in non-negligible concentration.

In many studies, only one or a few primary VOCs are present, and the RO2 population pool is heavily biased towards one or a few of the reactivity classes presented in the SI, especially in the early stages of the oxidation. Such consideration might be mentioned in the main paper. For me personally, given what I perceive as the main use of mechanisms of the envisioned detail, the most logical choice would be to separate the RO2 pool into reactivity classes.

Figures: While I recognize that adding uncertainty intervals on all the underlying data would make the figures visually cluttered, it could be useful to indicate somewhere in the caption what the typical uncertainty or scatter is on the data points underlying the fitting parameters.

SI, page 7, "The reaction of OH with ROOOH is expected to occur significantly by initial addition to the OOOH group". There are no free orbitals to accommodate an addition of OH, only abstraction, complexation, and substitution. I propose "... by initial attack on the OOOH group".

---

## Short Comment (SC2) · 7 Mar 2019

**About uncertainties on RO2 reaction rate constants**

I thank the authors of the manuscript Jenkin et al., Atmos. Chem. Phys. Discuss., https://doi.org/10.5194/acp-2019-44 for their response.

It appears from this response that some points need to be clarified concerning the uncertainties on the RO2 reaction rate constants reported by different groups. These uncertainties are not smaller than the factor x5/5 reported in our 2017 paper, they just have not been properly discussed. I hope that the explanation below will convince you and the atmospheric and kinetic community that our rate constants deserve to be taken into account at least as much as those from other groups, since they were dismissed on the ground of large uncertainties on the RO2 calibrations, while other works did not even perform such calibrations or involve larger mechanistic assumptions.

**1) UV absorbance methods**
Methods based on monitoring RO2 by UV absorbance, while more direct than those based on stable products, are still indirect because the rate constants are not obtained directly from the experimental data (= by methods that are free of assumptions, such as measuring decay rates) but by fitting kinetic mechanisms to these data. These mechanisms always involve assumptions since they are only as valid as the knowledge at the time. This is **potentially the largest source of uncertainties with these methods, yet these uncertainties are never even discussed** because there is no way to quantify them. As you pointed out, most of the current knowledge on RO2 reactions is based on rate constants obtained by these methods in the mid-1990's. These works pre-date by a decade the discovery of the isomerizations in the isoprene system and other autoxidation reactions. Thus, obviously, they do not take these channels into account in their kinetic analyses. This is why I qualified these works as "old". Autoxidation is just an example and, while it concerns only a small number of the RO2 studied in the 1990's, there can always be more mechanistic surprises in the future. **Thus the mechanistic uncertainties on these data is real, even if not quantified**.

It is precisely because I was aware of these limitations (being a former member of the Bordeaux group myself) that I spent many years developing a better way to monitor RO2. **The whole point of our new technique is that**, **not only it does not require any mechanistic assumption** (the rate constants being measured directly from experimental decay rates), and is thus free from the corresponding uncertainties, **but it can also detect unexpected new channels**.

Beside the large - but unquantified- mechanistic uncertainties in the 1990's kinetic studies, those officially reported are not far from ours: Villenave et al., J. Geophys. Res., 1998, for instance, reports uncertainties by factors 1.6 to 1.9 on the cross-reactions, thus nearly x2/2 (or did I misunderstand ?). This seems a minimum for a study where 12 different species contribute to the absorbance and the kinetic mechanism includes 23 reactions !
This specific system is further discussed in point 4) below.

**2) Mass spectrometric techniques ($NO_3^-$ and $NH_4^+$ ions)**
**The uncertainties in the mass spectrometric systems** (Berndt et al. 2015; 2018a; 2018b… and this is also probably true for similar works from the Helsinki group) **are not lower**, but only carefully eluded (as a matter of fact, only Berndt et al. 2015 are reporting uncertainties at all):
In chemical ionization techniques the detection sensitivity (= ratio signal over concentration) depends on the rate of the ion-analyte reaction, which, in principle, varies with different compounds (or RO2). Yet, in all the studies using $NO_3^-$ or $NH_4^+$ ionization, the authors elude the problem by **assuming** that all these reactions are at collision limit, thus that the detection sensitivity is identical

for all the analytes, which conveniently avoids the need for time-consuming calibrations of the concentrations (and correspondingly increases the publication rate). Yet, this essential point is not clearly presented as an assumption in the papers. And **I am not aware of any validation for it, nor of any attempt to calibrate the RO2 concentrations**. At least for the $NH_4^+$ systems used in the 2018 papers I recently had a verbal confirmation from the authors that "they have no idea of the concentration of RO2 in their system" because "there is no way to calibrate it". In the absence of any calibration of the RO2 concentrations, an uncertainties factor of x10/10 on these concentrations can be easily assumed, as this is the range of variability observed between different compounds in proton transfer reactions (and there is no reason to assume this is very different with $NO_3^-$ or $NH_4^+$ reactions). Therefore the resulting rate constants must carry the same x10/10 uncertainties. Of course, you are welcome to check all this with the authors themselves.

**3) Uncertainties in the Noziere & Hanson 2017 paper**
**To my opinion, the factor x5/5 reported in our 2017 paper is overly conservative.** Reporting such a large factor was essentially a request from my co-author because the calibrations of the RO2 concentrations had not been performed with the same method as previously (Hanson et al., Int. J. Mass. Spectrom., 2004). In particular, the quantities of NO added to titrate the RO2 were not measured directly but from the RO2 decay rate and my co-author was worried that mixing effects might have impacted the results. To lower that risk, I had performed the calibrations with various amounts of NO, and obtained the calibration factor from the slope $\Delta RO2/\Delta NO$, while potential mixing effects were expected to contribute to the intercept (mixing effects in 3 sLm bath gas not expected to change when adding various amounts of a trace compound).
The repeatability obtained in the results also support the fact that the x5/5 factor might be too conservative (one of the reviewers on our paper even pointed it out). In conclusion, **all the rate constants reported in our 2017 paper deserve to be taken into account by the community, at least as much as those reported by other groups**.

Concerning future calibrations, I am not planning more work on cross-reactions in the immediate future as I am currently studying other reactions. But I did perform recently a calibration of the RO2 in a similar flow system while monitoring NO with a photoluminescent NOx detector. For a series of primary RO2 (1-butylO2, 1-pentylO2 and 1-hexylO2) I obtained concentrations between 9 x 10(11) and 2.5 x 10(12) molec. cm-3. This seems to confirm the estimates in the 2017 papers, although these RO2 were produced by the direct photolysis of iodinated precursors rather than from Cl2. However, calibrating RO2 with NO is not straightforward, as it requires a "titration factor" taking into account the fact that each RO2 does not consume exactly one equivalent of NO. This factor can only be estimated from a kinetic model, with all the drawbacks expressed in point 1) above.
Therefore, it might be more accurate to calibrate the RO2 system with a PERCA in the future.

**4) Discrepancies in the peroxyacetyl radical ("PA") + t-butylO2 rate constant**
Concerning the PA+ t-butylO2 reaction, it is difficult to be sure of which rate constant is the most accurate, as this is a complex system. PA produces large concentrations of CH3O2, which, in turn, reacts fast with PA.

In our work, this reaction was studied by maintaining t-butylO2 constant in the reactor and adding PA periodically (Fig 4A and 7D). Fig. 4A shows that the addition of PA (red curve) consumes some of the CH3O2 present (produced by the self-reaction of t-butylO2, black curve) so that the overall concentration of t-butylO2 (blue curve) increases from the net effect of being consumed by PA and having its reaction with CH3O2 suppressed. As there is initially much less CH3O2 present than tbutylO2 (slow self-reaction) **this qualitatively confirms that the rate constant of reaction of PA with t-butylO2 must be much smaller than that of PA with CH3O2, otherwise t-butylO2 should be consumed instead of CH3O2**. But this also indicates that the amounts of PA added were very small, otherwise the CH3O2 level should have increased upon PA addition. Thus, the t-butylO2 profile might not have been much impacted by the reaction with PA and the rate constant reported might actually contain large uncertainties. I need more time to look into this, but will make sure to communicate the conclusion once I am sure.

The UV-based study of Villenave et al. 1998 (which, indeed, I had missed) might also contain large uncertainties, as their Fig. 1 shows that neither PA nor t-butylO2 are the largest contributions to the absorbance, but rather the build-up of CH3O2. Their analysis might not be very sensitive to the PA + t-butylO2 reaction either, but perhaps more to CH3O2 + PA.

---

## Referee Comment (RC2) · Paul O. Wennberg (Referee) · 8 Mar 2019

In this study, Jenkin and colleagues describe the formulation of 'rules' for the rate coefficients and product yields for reactions of organic peroxy radicals for use in mechanism construction. This manuscript documents how these rules are created and is not intended as a full review of the state-of-knowledge of such reactions. As a result of this scope (which is understandable and indeed necessary), at times this reviewer wishes for more detailed discussion of the choices made and critical review of the background literature. Clearly, however, this is not necessary within the context of the goals of this

paper. That said, below I highlight a few areas where I believe the authors might go further in justifying and improving their description of the RO2 chemistry. It would also be helpful if the authors address at the onset what is meant that these 'rules' are meant to "guide" the mechanism development. Please explain, for example, how, within the new MCM / GECKO framework, the authors intend to reconcile differences between specific reactions where experimental data exist and the rules/SAR based estimates (e.g. will the latter take precedent or the former in setting the rates / products?).

Specific comments (Page#.Line#):

2.22 In general where the competition is with NO, I'd suggest using 'NO' rather than 'NOx'.

2.29 HOMs: Include reference to Bianchi, 2019 - https://pubs.acs.org/doi/10.1021/acs.chemrev.8b00395

4.13 I believe that Teng was the first to point out that for multifunctional compounds, the nitrate branching ratios should (and do) scale more closely with heavy atoms than just carbon. Perhaps "updated by Arey et al. (2001) and Teng et al. (2015)"? In your definition of n(NCO), does the peroxy radical moiety count towards the 'O'? I'd suggest being explicit.

7.16 Should note that Caravan (2018) found a somewhat larger R5b/R5 (to methanol) at higher pressure.

7.17&7.22 Worth noting that Muller (2016) calculate that R5c/R5 is  $\sim$ .1 for CH3OO and Caravan (2018) suggest that they do see some CH3OOOH from this reaction.

8.8 Given that your fit to kHO2RO2 vs nCON is identical to that shown in Wennberg et al., 2018, figure 2, I guess that nCON does not include the peroxy moiety? We didn't weight our fit by the stated uncertainty - perhaps that should be done? Also, although we didn't consider this in our isoprene review, I expect that the T-dependence will depend on nCON at some level (presumably less strong for large nCON). For large

**ACPD**
n and low T, for example, the current parameterized rate will exceed that for kAPHO2 - this seems unreasonable.

8.24 "is taken to be the default where no information is available". This is the type of comment that I do not know how to interpret. In this context, does that mean for any RO2 + HO2 not described in Table 8?

9.17 (section 2.6). Thank you for engaging with Barbara Noziere's comment on this manuscript. I concur with her that the reported uncertainties in many RO2 + RO2 studies are underestimated given the (often) under-constrained observations of only bulk RO2 abundances. Thus, using reported uncertainty as a screen for which studies to include in formulating the SAR needs to be done critically. While the data shown in Figs. 4 and 5 gives some confidence in the resulting parameterizations, the log-log presentation hides the disagreement somewhat. Perhaps worth including a residual (fit-measure/measure) as a second panel.

13.20 Add Ng et al. to list of 'ROOR' studies - https://www.atmos-chem-phys.net/8/4117/2008/

13.16 Given all the recent results (e.g. those listed in 13.20), I don't see a reason not to recommend (generically) a few percent branching yield for R'OOR formation. I suspect that this is more correct than assuming 0% as is currently done.

14.1 Recognizing that this is a fast-moving area of research, Section 3 still seems a bit cursory and could be advanced using some recent literature as guidance. I believe that this is worth the time as there is now wide recognition that H-shift and endocylization reactions are important in many systems.

To more accurately capture this chemistry, the parameterization used could be improved using new observations and theoretical calculations (the section is currently based largely on older literature). Here are some of the recent literature I am aware of that could be used to broaden and deepen the recommendations:
Mohamed, 2018: https://pubs.acs.org/doi/pdfplus/10.1021/acs.jpca.7b11955 Otkjaer, 2018: https://pubs.acs.org/doi/abs/10.1021/acs.jpca.8b06223 Praske, 2017: https://www.pnas.org/content/115/1/64 Praske, 2018: https://pubs.acs.org/doi/10.1021/acs.jpca.8b09745 Bianchi, https://pubs.acs.org/doi/10.1021/acs.chemrev.8b00395 2019: Xu, https://pubs.acs.org/doi/10.1021/acs.jpca.8b11726 2019: 2019: Moller.

https://pubs.acs.org/doi/10.1021/acs.jpca.8b10432

14.22 Xu, 2019 offers new experimental and theoretical calculations for peroxy radical unimolecular chemistry following addition of OH and O2 to alpha and beta pinene that could be added to Table 14.

14.28 Otkjaer, 2018 offers high-level calculations of ring-size and constituent dependence of the H-shift chemistry for a number of organic substrates that should provide guidance for a first estimate for the rates of these reactions for consideration in the auto-generated mechanism.

14.28 Table 15. Should make clear what are calculated and experimental determinations. Also, k298K of alpha-formyl peroxy radicals the rate should be 0.57 s-1 (typo).

15.22 (and in SI) Assuming that the new mechanism will retain at least to the two radical pools produced following OH addition to isoprene, I do not understand why the 1,6 H-shift rates are not treated separately given there is significant evidence (Crounse, Teng) that a much larger fraction of the chemistry following addition at C4 will undergo this H-shift. Because the H-shift rates (not rate coefficients) for the C1 and C4 addition differ by an order of magnitude, use of the geometric mean will yield significant errors. Thus, I suggest it would be prudent to follow the recipe (if not the rates) described in Wennberg et al., 2018; Teng et al., 2018.

16.1-9 The literature cited above goes some way towards meeting the recommendations presented in this paragraph. I'd recommend considering them in the 'rules' developed in this work. **ACPD**

---

## Short Comment (SC3) · 12 Mar 2019

Thank you for providing additional information and discussion in your follow-up comment. Please see my full response in the attached file.

Please also note the supplement to this comment:
https://www.atmos-chem-phys-discuss.net/acp-2019-44/acp-2019-44-SC3-supplement.pdf

---

## Referee Comment (RC3) · Luc Vereecken (Referee) · 1 Apr 2019

Prof. Wennberg notes that "Teng was the first to point out that for multifunctional compounds, the nitrate branching ratios should (and do) scale more closely with heavy atoms than just carbon."

Historically, that is not quite accurate, as this has been discussed as far back as the turn of the century, and several models incorporated nitrate yields that are based on the number of heavy atoms, or even estimates that try to account for rigidity and other

factors affecting quantum state density and hence lifetime/pressure dependence. Much of this was based on theoretical state density and partition function calculations, and this data was exchanged e.g. during Eurotrac meeting around the years 2000.

Mechanistically, it is clear that the pressure dependence is due to collisional stabilisation which, given that the energetics are not all that different between different RO2+NO reactions, is thus directly linked to the state density of the peroxy nitrite intermediate. This is mostly governed by the low-frequency modes, i.e. the number of modes generated by the molecular skeleton containing the heavy atoms, whereas the H-atoms only contribute by providing a bit of mass, a high-frequency modes that are barely excited at room temperature and thus don't contribute significantly to the state density. These theoretical state density calculations were used by e.g. Jozef Peeters to construct more complex models that weighted for e.g. double bonds and rings that do not contribute to high-density internal rotations and are thus not as effective as single-bonded chains in increasing the lifetime and hence nitrate yields.

In our work, such models were used as far back as 2001 (a-pinene oxidation, Peeters et al.), and as recent as 2012 ( b-pinene oxidation, Vereecken and Peeters) where the nitrate yields used do not match the Arey et al. model exactly, but rather are based at least on the number of heavy atoms, and sometimes accounted for double bonds and other effects. An example would be one of the first nitrate formation steps in Peeters et al. 2001, figure 1, formation of RO3, C10 Arey et al. tert nitrate yield 10% 10.45%; C10+O2 tert nitrate yield 11.11%, used yield is rounded 11%.

At that time, it was felt to be sufficient to refer to Arey et al., as the theory-based model was due to be published in full, and it was in many respects a theory-based reparameterization of the Arey et al. model. An unfortunate choice, as ultimately Peeters never published his model, despite extensive hints in in our papers that this was due to happen; the main block was that no theoretical characterization of the nitrite to nitrate interconversion process was ever available, suggested now to be either a roaming reaction or a singlet-triplet-singlet double surface hop, both of which are very hard to

do computationally, and thus not characterized even today. Other authors did publish some work on this, e.g. Barker et al. 2003 probed the required energetic and rovibrational characteristics of the nitrite-nitrate interconversion process, but no computationally supported solution was ever found. Other scientists in those days likewise attempted to come up with models based on a quantification of the microscopic mechanism, but all faltered on the lack of a characterization of the nitrite-nitrate conversion step, as well as the odd differences in yields between primary, secondary and tertiary nitrates, which from a theoretical-mechanistic point of view remains unexplained. It may be that some models were presented as talks or posters at some conference, describing these efforts, but my memory does not stretches back that far, and I have only printed proceedings from this period, making searches too time-consuming for a merely historic reminiscence.

While it is possible that Teng et al. were the first to *explicitly* publish this finding in a peer-reviewed paper, the use of heavy atom number instead of carbon number in the prediction of nitrate yields thus dates back about 2 decades. Technically, for theoreticians, Barker et al. 2003 already indicates clearly that heavy atom count is more appropriate than carbon number, as that analysis is based on state density, and essentially only lacks a good description of the nitrite-nitrate conversion. The upcoming results on nitrate yields obtained at NCAR could likewise solve some of the conceptual problems related to prim/sec/tert yields that hampered development of theory-based models.

Feeling old, Luc Vereecken

---

## Referee Comment (RC4) · Geoffrey Tyndall (Referee) · 3 Apr 2019

This manuscript, the next in a series describing protocols for the automatic generation of chemical mechanisms, addresses the reactions of organic peroxy radicals. Methods are given for the calculation of both overall rate constants and product branching ratios.

The manuscript is detailed, and addresses all or most of the possible reaction partners for RO2 in the atmosphere. This is a lot to cover, and the manuscript is at times a little scant, but in general does a good job at giving enough information to follow what the

authors are trying to say.

I have one relatively minor technical comment, plus a general observation about alkyl nitrate yields, following on from Luc Vereecken's comment.

Page 3, line 18. The first carbon atom in this RO2 radical seems to be missing some bonds. I suspect it is meant to be the oxo dihydroperoxy radical, so C(O)(OOH)CH2. . . etc

Further thoughts on the temperature dependence of alkyl nitrate yields.

In their 1987 paper, Atkinson et al. [1] parameterized the nitrate yield as a function of temperature and pressure, leading to a pressure dependent term, Yo(298)*[M] multiplied by a temperature dependence (T/300)ˆmo with mo = -2.99, and Yo(300) = Aexp(n), where n is the number of carbon atoms. The high pressure yield in this formulation had a temperature coefficient of -4.69.

In 1989, Carter and Atkinson [2] instead parameterized the ratio ka/kb, and found the best fit with mo = 0, and m(inf) = -8.0. So all the temperature dependence was in the high pressure limit, which leverages the whole curve down to low pressure.

Arey et al. (2001) [3] adopted this latter formulation to extrapolate their room temperature values to other temperatures.

In our 2012 review paper (Orlando and Tyndall, 2012) [4] we attempted to combine the low pressure and temperature dependent terms, using Yo(298)[M](T/298). This is of course erroneous, since if mo=0 the temperature dependence vanishes (other than that implicit in [M]).

It appears that Jenkin et al. (main manuscript Page 4, line 16; SI Page 2) copied our incorrect version in their current manuscript. It is possible that Carter, Atkinson and Arey have updated their fit at some point to include a (T/298) term. However, we cannot remember having seen this anywhere (although we are even older than Dr. Vereecken, and we may have forgotten it).

We apologize for introducing this error into the literature. Note that the formula given in Calvert et al. (2009) "Mechanisms of Atmospheric Oxidation of the Alkanes" is correct, while that in Calvert et al. (2015) "The Mechanisms of Reactions Influencing Atmospheric Ozone" is not.

[1] R. Atkinson, S. M. Aschmann, and A. M. Winer, J. Atmos. Chem., 5 (1987), 91. [2] W. P. L. Carter and R. Atkinson, J. Atmos. Chem., 8 (1989), 165. [3] J. Arey, S. M. Aschmann, E. S. C. Kwok, and R. Atkinson, J. Phys. Chem., A 105 (2001), 1020. [4] J. J. Orlando and G. S. Tyndall, Chem. Soc. Rev., 41 (2012) 6294.
* * *
* * *

---

## Author Comment (AC2) · 27 Apr 2019

**Authors' responses to referee and discussion comments on:** Jenkin et al., Atmos. Chem. Phys. Discuss., https://doi.org/10.5194/acp-2019-44.

We are very grateful to the referees for their supportive comments on this work, and for their helpful suggestions for modifications and improvements. Responses to the comments are now provided (the original comments are shown in blue font). We received one additional comment shortly after the discussion closed. This is also reproduced here, along with our response.

**A. Comments by Referee 1**

**General comments**:

This manuscript discusses structure-activity relationships for peroxy radicals with its most common co-reactants in atmospheric conditions. The SARs are developed based on a selection of the available literature (mostly experimental data), and aim to provide site-specific rate coefficients and product distributions as appropriate for the reactions studied. The derivation of the SARs is well developed and explained, and the SARs strike a good balance between covering the mechanistic aspects of the target reactions on the one hand, and a pragmatic approach fitting data to a suitable function on the other hand, with good recovery of the training set. The data used as the training set is not an exhaustive literature tabulation. Some experimental data is missing (see also the comment by B. Nozière), and while some theoretical data is used, the potential of combing theoretical and experimental data has not been fully exploited. Overall, however, I feel that reasonable choices were made, giving a good summary of the reactivity trends discernible from the literature data, even if one could have a different view on what data to include in the training set, what weight to assign to each datum (which is not all that obvious especially for theoretical data at lower levels of methodology), or how to parameterize the SAR. What was missing a bit in places is reference to existing SARs and their approaches, but I recognize this paper is focused on presenting a new SAR, and need not be made longer by rigorous review or historic overview.

To put the usability of the SARs to the test, I have applied them in the development of a small mechanism (~100 reactions). The SARs prove to be quite usable even with a simple calculator, though during these efforts I found that adding a few additional subheadings would have made it easier to locate the desired information in the text: e.g. rate coefficients vrs. product distribution; self-reactions versus cross-reactions versus product distributions, etc.

Overall, this paper presents a good overview of the status quaestionis, and presents a set of very valuable SARs. Publication of the paper after minor revisions is recommended.

Response: We are grateful to the referee for these very positive and supportive comments on our work – and also for testing the methods in the development of a small mechanism. It is very gratifying to know that the methods have been found to be practical and usable. We acknowledge the referee's point about sub-headings. We have therefore added "kinetics" and "product branching ratios" subsections to section 2.1 on $RO_2$ + NO and section 2.5 on $RO_2$ + $HO_2$; and "kinetics of self-reactions", and "parameterized representation" subheadings to section 2.6 on $RO_2$ permutation reactions. Section 2.2 to 2.4 are relatively short, and these have therefore been left without subsections.

Although the tabulations we provide are probably not exhaustive, we feel that they are extensive, and provide good coverage of the hydrocarbon and oxygenated $RO_2$ *bimolecular* reactions for which there are laboratory experimental data. As stated in Sect. 3, we have not attempted exhaustive coverage of the fast moving topic of *unimolecular* $RO_2$ radical reactions, which will necessarily need to be revisited in future work (see also response to comments B11-B13).

We have not aimed to list all studies of all bimolecular reactions, but have given an evaluated or preferred rate coefficient for each reaction we tabulate – which we think cover most (if not all) for which there are reported reliable experimental data. In many cases, these are based on evaluations such as those of the *IUPAC Task Group on Atmospheric Chemical Kinetic Data Evaluation*, and therefore consider data from many studies; as our own evaluations have also done, where possible. We therefore believe the reaction listing is larger than in previous $RO_2$ reviews, partly because it can

include more recent data. For example, the table below illustrates that data for a larger number of bimolecular reactions of hydrocarbon and oxygenated $RO_2$ are presented than those appearing in the reviews of Orlando and Tyndall (2012) and Calvert et al. (2015) - noting that those reviews did not claim to be exhaustive, and also consider halogenated peroxy radicals, and reactions with halogenated species (e.g. ClO) that are outside the scope of our study.

| Reaction | This work | Orlando and Tyndall (2012) | Calvert et al. (2015) |
|---|---|---|---|
| $RO_2$ + NO | 23 | 13 | 14 |
| $RO_2$ + $NO_2$ ($k_f$, $k_b$) | 6, 9 | 7, 7 | 5, 6 |
| $RO_2$ + $NO_3$ | 8 | 5 | 5 |
| $RO_2$ + OH | 4 | - | - |
| $RO_2$ + $HO_2$ | 23 | 11 | 13 |
| $RO_2$ + $RO_2$ | 38 | 10 | 10 |
| $RO_2$ + R'$O_2$ | 20 | 3 | 5 |

Specific comments:

**Comment A1**: p. 3, line 22: The generic rate coefficient for RO2 + NO is appropriate for many peroxy radicals, but RO2 derived from aromatics have been reported to have slightly higher rate coefficients. The difference may not warrant a different class, but a short mention might be useful.

Response: We are only aware of one reported experimental rate coefficient for an aromatic-derived $RO_2$, but would be grateful to be pointed towards other data if available.

The experimental rate coefficient we are aware of is that listed for the 1,3,5-trimethylbenzene-derived $RO_2$ in Table 1, and is actually slightly lower than the generic value, $k_{RO2NO}$. This was reported by Elrod (2011) for a mixture of two complex radicals of molecular formula $HOC_9H_{12}[OO]O_2$, although with one isomer likely dominant (as stated in footnote (m) of Table 1). The reported rate coefficient was $7.7 \times 10^{-12}$ $cm^3$ molecule$^{-1}$ s$^{-1}$ at 296-298 K, with an estimated error of ± 30 %. This therefore agrees with $k_{RO2NO}$ ($9.0 \times 10^{-12}$ $cm^3$ molecule$^{-1}$ s$^{-1}$ at 298 K), and was reported by Elrod (2011) to confirm that the use of this same generic value for all aromatic-derived "peroxide bicyclic" $RO_2$ in the MCM was acceptable.

**Comment A2**: p. 4, line 17: State explicitly (again) that nCON does not include the peroxy radical oxygen atoms, as an equally logical choice could have been a nCON based on the full molecular stoichiometry, i.e. including all functionalities. It might be useful to have a short reminder in other places as well.

Response: We have further clarified this point as suggested (see also comment B3). The revised text reads as follows in the revised manuscript and SI (new text in red font):

"$n_{CON}$ is the number of carbon, oxygen and nitrogen atoms in the organic group (R) of the peroxy radical (i.e. excluding the peroxy radical oxygen atoms and equivalent to the carbon number in alkyl peroxy radicals), T is the temperature (in K) and [M] is the gas density (in molecule cm$^{-3}$)."

In conjunction with the existing indication that it is equivalent to the carbon number in alkyl peroxy radicals (and must therefore exclude the peroxy radical oxygens), and the $n_{CON}$ = 2 examples, $C_2H_5O_2$ and $HOCH_2O_2$, given in Sect. 2.4 (page 7, line 22 in original manuscript), we believe that readers will understand the definition.

**Comment A3**: p. 4: The parameterization of the nitrate yield may need to be updated soon following recent work of John Orlando et al (NCAR). No publication is available to my knowledge, but interesting results were presented at conferences; I suggest contacting these authors to see if there is a need for alternative SAR parameters.

Response: We thank the referee for this information. We provide more discussion of the nitrate yield parameterization below in the responses to reviewer comments B3, C1 and D2 and additional comment E1.

**Comment A4**: p. 7, line 13: formation of CI from CH3O2 + OH: Also state that the small to negligible yield of CI is consistent with theoretical data..

Response: We thank the referee for this information. The relevant sentence has now been amended to read:

"However, no evidence for formation of $CH_2O_2$ and $H_2O$ has been observed at room temperature, indicating that this product channel is at most minor (< 5%) (Yan et al., 2016; Assaf et al., 2017a; Caravan et al., 2018), this also being consistent with theoretical data (e.g. Müller et al., 2016)."

**Comment A5**: p. 8, R6c and R6e: R-HO is perhaps better written as R-H=O, unless the authors mean to imply that the H-atom transferred is not necessarily adjacent to the peroxy radical group.

Response: The referee is correct that the transferred H-atom is adjacent to the peroxy radical group. The product is therefore now represented as $R_{-H}=O$ (or $R'_{-H}=O$) at all relevant points in the manuscript.

**Comment A6**: p. 10: readability might be improved if using a notation for kRO2RO2 that indicates whether an expression pertains to self-reactions vrs. cross-reactions. Additional subheadings might be useful to make finding specific topics easier when applying the SAR (reference self reactions, self reactions, cross reactions, branching ratios,...).

Response: We agree with the referee that this (quite long) section was quite difficult to navigate through, and have added subheadings for "kinetics of self-reactions" and "parameterized representation". To clarify, the rate coefficient expressions either refer to the "self-reactions" (i.e. Eqs (14)-(17)), or to the "parameterized representation of the permutation reaction reactions" (i.e. Eqs (21)-(25)) and now appear in the "kinetics of self-reactions" and "parameterized representation" subsections, respectively. The $k_{RO2RO2}$ parameters always refer to self-reactions, and the shorter parameters, $k_{AP}$ and $k_{RO2}$, refer to the pseudo-unimolecular parameterized representation. We agree that this is clearer with the new section structure.

**Comment A7**: p. 11: line 29: "... if the peroxy radical contains more than one benzyl group". A benzyl group is C6H5-C.H2, and there can be only one. The authors probably mean multiple beta-phenyl groups?

Response: Within a strict definition, we acknowledge that the referee is correct. In fact, we are using benzyl even more generically in this discussion to mean a $\beta$-aryl group (i.e. including $\beta$-phenyl groups and substituted $\beta$-phenyl groups). We have now therefore changed "benzyl" to "$\beta$-aryl" at the relevant points in the manuscript. Accordingly, we have also changed generic uses of the term "phenyl" to "aryl".

Using the referee's reasoning, it is probably also strictly incorrect to use the term "allyl" generically, as this refers specifically to $CH_2CHC.H_2$ – although the term "allyl" seems to be used very widely as a generic term for all alk-2-enyl groups. We have now therefore also changed "allyl" to the more generic term "allylic" at the relevant points in the manuscript.

**Comment A8**: p. 11, line 29: the formula for calculating alpha and beta needs an equation number to allow unambiguous references in implementations.

Response: This has been rectified in the revised manuscript.

**Comment A9**: p. 12: line 7: "This is regarded as a logical choice, because CH3O2 is the most abundant organic peroxy radical in the atmosphere". An explicit or semi-explicit mechanism as seems to be the target here is not used all that often for global modeling or even regional modeling as they tend to be too large. Without having access to any reliable statistical data, I would guess that e.g. the MCM

is more often used to model specific experiments such as environmental chambers or lab studies, where CH3O2 is not necessarily the dominant proxy radical, if it is present at all in non-negligible concentration.

In many studies, only one or a few primary VOCs are present, and the RO2 population pool is heavily biased towards one or a few of the reactivity classes presented in the SI, especially in the early stages of the oxidation. Such consideration might be mentioned in the main paper. For me personally, given what I perceive as the main use of mechanisms of the envisioned detail, the most logical choice would be to separate the RO2 pool into reactivity classes.

Response: The referee raises some interesting points. Based on previous applications of the MCM and GECKO-A, the mechanisms to which the methods will be applied are likely to be very varied. In the paper we present (i) methods for estimating self- and cross-reaction rate coefficients (i.e. Eqs. (17) and (20)) that could be used in a fully explicit representation; (ii) a parameterized method involving 9 reactant peroxy radical classes that can be used in a highly explicit mechanism; and (iii) a parameterized method based on a single reactant peroxy radical class, which can be used to limit the number of permutation reactions further, as required. We therefore cover a wide variety of possible applications. The choice to present the simpler parameterization in the main paper was primarily to limit the length of an already quite long section, with this logically expanded to the related 9 class parameterization in the SI. This was not intended to imply that the single class parameterization is our recommended method. That we have presented the 9 class parameterization confirms that we have covered the referee's preferred approach (as stated in the final sentence of comment A9), along with information that hopefully serves the needs of others.

The single class parameterization has traditionally been used in the MCM as one simplification measure. The MCM has been applied in regional models (e.g. Li et al., 2015), and is frequently used as a reference benchmark in reduced mechanism development. We would therefore like to provide further explanation here of why $CH_3O_2$ is a logical choice for defining the parameterized rate coefficients for reactions of non-acyl peroxy radicals with the single-class $RO_2$ pool. $CH_3O_2$ is invariably simulated to be the most abundant peroxy radical in the atmosphere, present at sufficient concentration to make it a major reaction partner – and usually the major reaction partner. Even in the isoprene dominated tropical boundary layer simulations of Jenkin et al. (2015), it accounted for between 35 % and 40 % of the peroxy radical population across the wide $NO_x$ range considered (about 30 ppt to 8 ppb). As stated in the current paper, it is also in the middle of the peroxy radical self-reaction reactivity range. For example, its self-reaction rate coefficient ($3.5 \times 10^{-13}$) is intermediate between those reported for the two most abundant OH + isoprene-derived peroxy radicals (0.69 and $57 \times 10^{-13}$; geometric mean $6.3 \times 10^{-13}$) and between those calculated here for large secondary and tertiary $\beta$-hydroxy peroxy radicals (0.079 and $15 \times 10^{-13}$; geometric mean $1.1 \times 10^{-13}$), as formed, for example, from reaction of OH with a number of monoterpenes and sesquiterpenes (e.g. $\alpha$-pinene, limonene, $\beta$-caryophyllene).

However, we agree with the referee that the MCM has been widely used to simulate a variety of chamber systems, and that the alternative 9 class parameterization or an explicit representation might be more appropriate in some cases. When tractable, MCM authors have always verbally recommended using an explicit representation of peroxy radical self- and cross-reactions, although this recommendation has not been stated on the website. Of course, the current paper is not discussing current or past versions of the MCM, it is aimed at providing the basis for the automated generation of the next generation of mechanisms, with the potential for providing optional approaches. As indicated above, the methods presented therefore cover a wide range of possible applications where a representation of peroxy radical permutation reactions might be required.

**Comment A10**: Figures: While I recognize that adding uncertainty intervals on all the underlying data would make the figures visually cluttered, it could be useful to indicate somewhere in the caption what the typical uncertainty or scatter is on the data points underlying the fitting parameters.

Response: We investigated including error bars on the plots, and can confirm that this does generally make them very cluttered and unclear. However, we agree that some indication of scatter would be helpful, particularly on plots with a log scale. In view of the referee's comment (see also response to comment B8), we have included lines showing factor of three increase and decrease ranges in Fig. 4; and note that Fig. 5 already includes a line illustrating a factor of two change in the rate coefficient.

**Comment A11**: SI, page 7, "The reaction of OH with ROOOH is expected to occur significantly by initial addition to the OOOH group". There are no free orbitals to accommodate an addition of OH, only abstraction, complexation, and substitution. I propose "... by initial attack on the OOOH group".

Response: We thank the referee for pointing out this error, which has been corrected as suggested in the revised SI.

**B. Comments by Paul O. Wennberg (Referee)**

**Opening comment:**

In this study, Jenkin and colleagues describe the formulation of 'rules' for the rate coefficients and product yields for reactions of organic peroxy radicals for use in mechanism construction. This manuscript documents how these rules are created and is not intended as a full review of the state-of-knowledge of such reactions. As a result of this scope (which is understandable and indeed necessary), at times this reviewer wishes for more detailed discussion of the choices made and critical review of the background literature. Clearly, however, this is not necessary within the context of the goals of this paper. That said, below I highlight a few areas where I believe the authors might go further in justifying and improving their description of the RO2 chemistry. It would also be helpful if the authors address at the onset what is meant that these 'rules' are meant to "guide" the mechanism development. Please explain, for example, how, within the new MCM / GECKO framework, the authors intend to reconcile differences between specific reactions where experimental data exist and the rules/SAR based estimates (e.g. will the latter take precedent or the former in setting the rates / products?).

Response: We thank the referee for these positive comments on our work, and for the suggestions for additions and improvements.

The referee asks for additional information on how the methods are applied, and we are pleased to provide an overview here. The main aim of this work is to document a set of estimation methods (SARs and generic rate coefficients) which can be used define the chemistry of peroxy radicals in mechanism development. It therefore has broadly the same aim as previous published SAR studies, and follows on from our preceding papers covering OH + VOC reactions (e.g. Jenkin et al., 2018). It very much fits into the strategy outlined by Vereecken et al. (2018) (cited in our Introduction) to help promote the sustainable development of chemically detailed mechanisms that reflect current kinetic and mechanistic knowledge.

The methods, or rules, presented in our paper are intended to be formulated to allow practical use in automated mechanism generators. They therefore contribute to a *detailed chemical protocol* that allows a generator to produce fully explicit chemical mechanisms, containing all reactions of all intermediates. This is the first step in the process.

In practice, such mechanisms are of course too large to be usable (e.g. see Aumont et al, 2005), and a *reduction protocol* also needs to be defined. This is a further set of rules that allows the mechanisms to be trimmed or simplified (e.g. by omitting minor reaction channels beneath a threshold contribution). These methods are under revision, and may in any case vary depending on the intended application of the mechanism being generated. These methods will therefore be reported in future mechanism generation/application papers, and are generally not reported here. The only exceptions to this are the parameterization options for the peroxy radical permutation reactions, which will likely be required in most applications.

The vast majority of the reactions in a generated mechanism are unstudied (e.g. MCM v3.3.1 contains about 1200 $RO_2$ radicals which all need to react). However, for the small subset of reactions for which there are measured data, the preferred data set is used to overwrite the relevant estimated parameters (i.e. a reliable experimentally-determined parameter does indeed take precedence over an estimated parameter).

**Comment B1**: Specific comments (Page#.Line#):

2.22 In general where the competition is with NO, I'd suggest using 'NO' rather than 'NOx'..

Response: Although a very minor point indeed, we generally agree with the referee. At this point, however, the preceding paragraph has summarized the reactions of $RO_2$ radicals with NO and $NO_2$ (i.e. $NO_x$) and the related species $NO_3$. The discussion is moving on to reactions with other species, so we feel the term "$NO_x$" is appropriate here.

**Comment B2**: 2.29 HOMs: Include reference to Bianchi, 2019 - https://pubs.acs.org/doi/10.1021/acs.chemrev.8b00395

Response: Thank you for alerting us to this very recent review paper, which is cited in the revised manuscript as suggested.

**Comment B3:** 4.13 I believe that Teng was the first to point out that for multifunctional compounds, the nitrate branching ratios should (and do) scale more closely with heavy atoms than just carbon. Perhaps "updated by Arey et al. (2001) and Teng et al. (2015)"? In your definition of n(NCO), does the peroxy radical moiety count towards the 'O'? I'd suggest being explicit.

Response: Thank you for this suggestion. The Teng et al. (2015) work is now cited at this point in the revised manuscript (see also the responses to the related referee comments C1 and D2 and additional comment E1). As indicated above in the response to referee comment A2, we have further clarified the definition of $n_{CON}$ in the revised manuscript and SI.

**Comment B4:** 7.16 Should note that Caravan (2018) found a somewhat larger R5b/R5 (to methanol) at higher pressure.

Response: Caravan et al. (2018) report a methanol yield 6-9 %, based on MPIMS measurements at both 30 Torr and 740 Torr. The additional formation at the longer time scales in their chamber experiments was reported to have a contribution from heterogeneous conversion of the low yield of $CH_3OOOH$ formed. They applied a value of 7 % in their global model calculations (based on their MPIMS measurement of a 6-9 % yield), with this agreeing with theory. We therefore decided not to overcomplicate the text, as yields for several different channels are being discussed.

**Comment B5:** 7.17&7.22Worth noting that Muller (2016) calculate that R5c/R5 is ~ .1 for CH3OO and Caravan (2018) suggest that they do see some CH3OOOH from this reaction.

Response: We have now included this point, although we have instead cited the result of the Assaf et al. (2018) calculation for consistency with our approach to formation of larger ROOOH products in the subsequent paragraph. The added text reads:

"It is noted that Caravan et al. (2018) also reported evidence for minor $CH_3OOOH$ formation at atmospheric pressure, via channel (R5c), although this has been calculated to be formed in very low yield (1.7 %) by Assaf et al. (2018)."

**Comment B6:** 8.8 Given that your fit to kHO2RO2 vs nCON is identical to that shown in Wennberg et al., 2018, figure 2, I guess that nCON does not include the peroxy moiety? We didn't weight our fit by the stated uncertainty - perhaps that should be done? Also, although we didn't consider this in our isoprene review, I expect that the T-dependence will depend on nCON at some level (presumably less strong for large nCON). For large n and low T, for example, the current parameterized rate will exceed that for kAPHO2 – this seems unreasonable.

Response: As indicated above in the response to referee comments A2 and B3, we have further clarified the definition of $n_{CON}$ in the revised manuscript and SI.

Fig. 2 does graph the same quantities as the figure in Wennberg et al. (2018), although it also includes data for some additional peroxy radical classes. We had not realised the fitted parameters (based on alkyl peroxy and β-hydroxy peroxy radical data) were essentially identical to Wennberg et al. (2018), as this analysis was carried out in 2016 and is only now being presented in a publication. It is a logical extension to our previous use of this type of function for the $RO_2 + HO_2$ reaction (e.g. Jenkin et al., 1997); with the change from carbon number to $n_{CON}$ being consistent with our approach for representing the size dependence of the yield of thermalized α-hydroxy peroxy radicals from the reactions of α-hydroxyalkyl radicals with $O_2$. That was published in an earlier paper in this series (Jenkin et al., 2018). We also considered using the mass of the organic group, which works equally well.

Weighting the analysis, based on reported uncertainty, actually has little effect. This is because most of the points are quite close to the curve. A significant change would require one of those farthest from the line to be much more precisely determined than the rest – which is not the case.

As we indicated at the relevant point in the manuscript, the temperature dependence is typical of that reported for > $C_2$ alkyl and β-hydroxy $RO_2$ radicals (see Fig. R1, below) and remains unchanged from that applied previously by Saunders et al. (2003) – as also adopted by Wennberg et al. (2018). On the basis of the (albeit limited) data, it would seem difficult to justify making the temperature dependence weaker as $n_{CON}$ increases. Additional temperature-dependent data for large peroxy radicals are clearly required to confirm or modify this assumption.

[Figure]

Fig. R1 Temperature coefficients for reactions of alkyl and β-hydroxyalkyl $RO_2$ radicals with $HO_2$ as a function of $n_{CON}$.
* * *
The referee is correct that $k_{RO2HO2}$ will exceed $k_{APHO2}$ if the temperature is reduced enough. This is because of the weaker temperature dependence applied to $k_{APHO2}$, this being based on the value for $CH_3C(O)O_2 + HO_2$. Although data are scarce, the only other rate coefficient for an acyl peroxy radical ($C_6H_5C(O)O_2$) also has a reported weak temperature coefficient (see Table 7) - again, additional temperature-dependent data for large peroxy radicals are required. Based on the coefficients we originally reported for the high $n_{CON}$ limit, the cross-over occurred at about 230 K, with $k_{RO2HO2}$ and $k_{APHO2}$ still within a factor of 1.4 at 210 K.

Since we submitted the paper, Hui et al. (2019) have published a new temperature-dependent kinetics and branching ratio study for $CH_3C(O)O_2 + HO_2$ (extending down to below 230 K), the first to report the temperature-dependence of the OH-forming channel. Although their results support our

use of a reduced temperature dependence for the rate coefficient (compared with earlier $CH_3C(O)O_2$ + $HO_2$ data), their reported value, $E/R$ = -(720 ± 170) K, is slightly stronger than the value of -580 K that we used. We have therefore revised our parameterization to take account of this – and this slightly reduces the high $n_{CON}$ cross-over temperature to about 225 K, with $k_{RO2HO2}$ and $k_{APHO2}$ still within a factor of 1.2 at 210 K. Given that these temperatures are well outside the studied range of most $RO_2$ + $HO_2$ reactions, we feel this is acceptable.

The resultant updated information on the treatment of acyl peroxy + $HO_2$ reactions in the revised manuscript is now summarized:

(i) The revised kinetics entry for $CH_3C(O)O_2$ in Table 7 is as follows:

| Peroxy radical | $A$ | $E/R$ | $k_{298\,K}$ | Comment |
|---|---|---|---|---|
| | ($10^{-13}$ cm$^3$ molecule$^{-1}$ s$^{-1}$) | (K) | ($10^{-12}$ cm$^3$ molecule$^{-1}$ s$^{-1}$) | |
| *Acyl* $CH_3C(O)O_2$ | 17.9 | -720 | 20.0 | (m) |

$^m$ $k_{298\,K}$ based on Groß et al. (2014), Winiberg et al. (2016) and Hui et al. (2019). $E/R$ based on Hui et al. (2019) (see Sect. S4);

Regarding product branching ratios, footnote (b) in Table 8 has also been updated to read:

"Based on studies of $CH_3C(O)O_2$ (Niki et al., 1985; Horie and Moortgat, 1992; Hasson et al., 2004; Jenkin et al., 2007; Dillon and Crowley, 2008; Groß et al., 2014; Winiberg et al., 2016; Hui et al., 2019); see Sect. S4. Hasson et al. (2012) also reported broadly comparable branching ratios for $C_2H_5C(O)O_2$ and $C_2H_5C(O)O_2$;"

(ii) Eq. (10) and preceding text in (new) sub-section 2.5.1 now reads:

"Based on the limited data for acyl peroxy radicals (see Fig. 2 and Table 7), and specifically that for $CH_3C(O)O_2$, the 298 K rate coefficients are assigned values that are almost a factor of two greater than those defined by Eq. (9). The temperature dependences reported for acyl peroxy radicals appear to be weaker than those for similar sized radicals in other classes, and the temperature coefficient is again based on that recommended for $CH_3C(O)O_2$. The following expression is therefore assigned to acyl peroxy radicals:

$$k_{APHO2} = 3.6 \times 10^{-12} \exp(720/T) [1-\exp(-0.23n_{CON})] \text{ cm}^3 \text{ molecule}^{-1} \text{ s}^{-1} \tag{10}"$$

Fig. 2 has also been slightly modified as a result.

(iii) The description of the temperature dependence of the channel branching ratios/rate coefficients in (new) sub-section 2.5.2 now reads (new or adjusted text in red font):

"….This class of reaction (in particular the reaction of $HO_2$ with $CH_3C(O)O_2$) has received the most attention, and is also a class for which radical propagation is reported to be particularly important at temperatures near 298 K. As shown in Table 8, channels (R6a), (R6b) and (R6d) are reported to contribute. The temperature dependence of $k_{6d}/k$ is based on the recent study of the $CH_3C(O)O_2$ + $HO_2$ reaction reported by Hui et al. (2019). The contributions and temperature dependences of $k_{6a}/k$ and $k_{6b}/k$ also take account of the wider database for the same reaction, in particular the experimental characterization of $k_{6a}/k_{6b}$ reported by Horie and Moortgat (1992). This procedure (described in detail in Sect. S4) results in the following fitted Arrhenius expressions for the individual channel rate coefficients:

$$k_{6a\,APHO2} = 3.11 \times 10^{-12} \exp(473/T) [1-\exp(-0.23n_{CON})] \tag{11}$$

$$k_{6b\,APHO2} = 9.14 \times 10^{-15} \exp(1900/T) [1-\exp(-0.23n_{CON})] \tag{12}$$

$$k_{6d\,APHO2} = 9.68 \times 10^{-12} \exp(225/T) [1-\exp(-0.23n_{CON})] \tag{13}$$

The corresponding temperature dependences of the channel rate coefficients, derived from the $CH_3C(O)O_2$ data, are thus applied to all (non-aryl) acyl peroxy radicals. The variation of the branching ratios and channel rate coefficients are illustrated for the $CH_3C(O)O_2$ + $HO_2$ reaction in Figs. S2 and S3, for the 230-300 K temperature range. Summation of the channel rate coefficients given in Eqs. (11)-(13) reproduces the values of $k_{APHO2}$ calculated for the overall reaction using Eq. (10) to within 5 % over this temperature range (see Sect. S4 for further details)."

**Comment B7:** 8.24 "is taken to be the default where no information is available". This is the type of comment that I do not know how to interpret. In this context, does that mean for any RO2 + HO2 not described in Table 8?

Response: The referee has interpreted the statement correctly. The answer to this question is actually given in footnote (a) of Table 8, where it states that formation of ROOH and $O_2$ is "…also used as a default in all cases other than those covered by comments (b)-(i)." We suspect most readers wanting to apply the information would examine Table 8 where the guidance is provided, and have therefore now added an additional reference to that table in the sentence quoted by the referee. We believe that Table 8 covers those systems for which evidence for the other product channels has been established. Unlike reviews of atmospheric chemistry, mechanism development protocols necessarily need to provide guidance on how to proceed when information is lacking.

**Comment B8:** 9.17 (section 2.6). Thank you for engaging with Barbara Noziere's comment on this manuscript. I concur with her that the reported uncertainties in many RO2 + RO2 studies are underestimated given the (often) under-constrained observations of only bulk RO2 abundances. Thus, using reported uncertainty as a screen for which studies to include in formulating the SAR needs to be done critically. While the data shown in Figs. 4 and 5 gives some confidence in the resulting parameterizations, the log-log presentation hides the disagreement somewhat. Perhaps worth including a residual (fit-measure/measure) as a second panel.

Response: The present authors include members of data evaluation panels, and therefore fully concur with the referee's point about critical evaluation. The referee is correct that the many kinetics studies using UV absorption detection were complicated by overlap of the peroxy radical absorption spectra, and therefore required careful interpretation and assessment. However, they were nonetheless direct measurements based on observation of the time-dependence of (initially) relatively simple chemical systems. Reported uncertainties may indeed be too low in some studies (particularly for complex systems in which sequential formation of a number of peroxy radicals occurs), but many studies base their uncertainties on reasonable sensitivity analyses and are therefore more reliable estimates. In practice, the majority of the reported kinetics studies of peroxy radical self-reactions, cross-reactions and reactions with $HO_2$ are based on this type of measurement, which collectively form a substantial and invaluable data base.

It is, of course, important and desirable that new and complementary methods are applied to confirm or challenge rate coefficients reported in those previous studies. Ideally, these should have the advantage of speciated detection of the reacting peroxy radicals, but without losing the advantages of direct time resolved observations of (initially) relatively simple chemical systems. As a result of our discussions with Barabara Nozière, we have factored in some of the Nozière and Hanson (2017) data into our tabulations. Their work has the advantage of speciated detection of peroxy radicals of different mass, although the method of extraction of kinetic data is less direct than in the UV absorption studies (i.e. based on perturbations to "steady state" concentrations at the exit of a flow tube). However, following critical evaluation, we have not taken account of their data for *t*-butyl peroxy radical kinetics, which seem to be subject to a number of significant complications and interferences – most notably the more significant production of the isomeric *i*-butyl peroxy radical in their system (see discussion comment SC3: Atmos. Chem. Phys. Discuss., https://doi.org/10.5194/acp-2019-44-SC3, 2019). Despite these complications, their extracted cross-reaction rate coefficients (nominally) for *t*-butyl peroxy radicals are all apparently close to the geometric mean of the self-reaction rate coefficients (i.e. the expected target value without complications), which we do not fully understand.

We thank the referee for the suggestion of including panels presenting the (calc-obs)/obs deviation, which we have considered carefully. However, having prepared such panels, it was apparent that they only repeated similar information to that which is already clear from the existing figures. This is because the vertical deviation of the points from the line on a log scale is a direct measure of the factor by which the values differ. In Fig. 4, we have instead included lines showing the factor of three increase and decrease ranges, within which all but one of the points fall (with most being much

closer). We think this is an acceptable alternative. In Fig. 5, a factor of two increase line is already included. No additional lines are added to avoid making the figure too cluttered.

**Comment B9:** 13.20 Add Ng et al. to list of 'ROOR' studies - https://www.atmos-chemphys.net/8/4117/2008/

Response: We thank the referee for this suggestion. This reference has now been included at the relevant point.

**Comment B10:** 13.16 Given all the recent results (e.g. those listed in 13.20), I don't see a reason not to recommend (generically) a few percent branching yield for R'OOR formation. I suspect that this is more correct than assuming 0% as is currently done.

Response: We agree that there is increasing evidence for the formation of ROOR/R'OOR, and that it is possible to do what the referee suggests in an explicit representation of the chemistry. The issue we are discussing here is the practical difficulty in representing this channel in the pseudo-unimolecular parameterization of the permutation reactions involving reaction with a pool (or pools) of peroxy radicals. This is because only the RO- substructure deriving from the reacting $RO_2$ can be represented in the product (i.e. the -OR' substructure relates to the variable distribution of peroxy radicals in the pool(s) and cannot be incorporated into the product). We put forward the basis of a possible (compromise) approach, but feel that much more information is required before this can be defined more fully, and we are keeping this under review. In view of the referee's comment, we have made it clearer that we are discussing the parameterization, both through inclusion of subsection headings (suggested by Reviewer A, General comments); and though a number of minor changes to the subsequent paragraph, which now reads as follows (new or adjusted text in red font):

"Although not currently included in the parameterized representation, channel (R9d) is listed to acknowledge the potential formation of peroxide products (i.e. reactions (R7c) and (R8d)). Although these channels have generally been reported to be minor for small peroxy radicals (e.g. Lightfoot et al., 1992; Orlando and Tyndall, 2012), recent studies suggest that they may be more significant for larger peroxy radicals containing oxygenated substituents, and they have been reported to play a role in the formation of low volatility products in a number of studies (Ziemann, 2002; Ng et al., 2008; Ehn et al., 2014; Jokinen et al., 2014; Mentel et al., 2015; Rissanen et al., 2015; Berndt et al., 2015; 2018a; 2018b; Zhang et al., 2015; McFiggans et al., 2019). These reactions may therefore play a potentially important role in particle formation and growth in the atmosphere. The product denoted "$RO_{(peroxide)}$" in reaction (R9d) notionally represents the monomeric contribution the given peroxy radical makes to the total formation of (dimeric) peroxide products. However, it is not an independent species for which subsequent gas phase chemistry can be rigorously defined, such that reaction (R9d) cannot be universally represented within the parameterization. In principle, it could be included for the permutation reactions of a subset of larger peroxy radicals, with the $RO_{(peroxide)}$ product assumed to transfer completely to the condensed phase (i.e. not participating in gas phase reactions). However, there is currently insufficient information on the structural dependence of the contributions of channels (R7c) or (R8d) to the overall self- and cross-reactions to allow the branching ratio of channel (R9d) to be defined reliably. Further systematic studies of these channel contributions are therefore required as a function of peroxy radical size and functional group content."

**Comment B11:** 14.1 Recognizing that this is a fast-moving area of research, Section 3 still seems a bit cursory and could be advanced using some recent literature as guidance. I believe that this is worth the time as there is now wide recognition that H-shift and endocylization reactions are important in many systems.

To more accurately capture this chemistry, the parameterization used could be improved using new observations and theoretical calculations (the section is currently based largely on older literature). Here are some of the recent literature I am aware of that could be used to broaden and deepen the recommendations:

Mohamed, 2018: https://pubs.acs.org/doi/pdfplus/10.1021/acs.jpca.7b11955

Otkjaer, 2018: https://pubs.acs.org/doi/abs/10.1021/acs.jpca.8b06223

Praske, 2017: https://www.pnas.org/content/115/1/64

Praske, 2018: https://pubs.acs.org/doi/10.1021/acs.jpca.8b09745

Bianchi, 2019: https://pubs.acs.org/doi/10.1021/acs.chemrev.8b00395

Xu, 2019: https://pubs.acs.org/doi/10.1021/acs.jpca.8b11726

Moller, 2019: https://pubs.acs.org/doi/10.1021/acs.jpca.8b10432

**Comment B12:** 14.22 Xu, 2019 offers new experimental and theoretical calculations for peroxy radical unimolecular chemistry following addition of OH and O2 to alpha and beta pinene that could be added to Table 14.

**Comment B13:** 14.28 Otkjaer, 2018 offers high-level calculations of ring-size and constituent dependence of the H-shift chemistry for a number of organic substrates that should provide guidance for a first estimate for the rates of these reactions for consideration in the auto-generated mechanism.

Response to comments B11-B13: We are very grateful to the referee for listing these references. These illustrate very well that this is a very fast moving area of research, and would seem to vindicate our decision not to attempt an exhaustive treatment at this stage. Although we might have included the earlier studies in the above list, we note that four of the papers have been published since 14th December 2018, two of them since our paper was submitted (18th January 2019), with one published less than two weeks before the referee posted his review. Because studies will no doubt continue to emerge rapidly over the coming months, we are fully aware that we will need to revisit the topic of unimolecular $RO_2$ reactions before we can attempt to define a set of SARs for automated mechanism generation, as we stated.

We have been unable to assimilate all this information, and work it up into a set of SAR methods, on the time scale of this discussion response. We have therefore edited the section to include the above references. We have re-emphasized at a number of points that the topic continues to be considered in ongoing work, and that a more complete treatment will be developed. The relevant changes to the paper are as follows:

(i) The introductory text in Sect. 1 has been changed to read (new or moved text in red font):

"In this paper, published data on the kinetics and branching ratios for the above bimolecular reactions of hydrocarbon and oxygenated $RO_2$ radicals are reviewed and discussed. Preliminary information is also presented for selected unimolecular isomerization reactions, which continue to be considered in ongoing work. The information on bimolecular reactions is used to define and document a set of rules and structure-activity relationship (SAR) methods (a chemical protocol)…".

(ii) The references listed by the referee are now all cited in the introductory text in Sect. 3.

(iii) Because it does not only consider ring-closure reactions, the information from Xu et al. (2019) has not been included in Table 14. However, the following text has been added at the relevant point in Sect. 3.1:

"It is noted that Xu et al. (2019) have also very recently reported information for a series of isomerization reactions (including ring-closure reactions) for the $\alpha$- and $\beta$-pinene systems, which are being considered in ongoing work."

The captions to both Tables 14 and 15 have been adjusted to indicate that the rate coefficients are currently representative rather than assigned. Although some may become the assigned rate coefficients in the finalized method, this provides the flexibility to update methods.

(iv) In the final paragraph of Sect. 3.2, the text about the need for information on 1,$n$ H-shift reactions has been amended to read (new or adjusted text in red font):

"….requires systematic information on the rates of a series of 1,$n$ H-shift reactions from C-H and O-H bonds in different environments. In this respect, it is noted that the systematic influence of a series of neighbouring functional groups and transition state sizes have been considered in theoretical studies of a number of model systems (e.g. Crounse et al., 2013; Jørgensen et al., 2016; Praske et al., 2017; Otkjaer et al., 2018). Such studies provide the basis for defining systematic structure-activity methods for a wide range of $RO_2$ radicals and their potential isomerization reactions, and are being considered in ongoing work."

We hope the above changes are acceptable. We did consider removing completely (i.e. deferring) the detailed information on unimolecular reactions of RO$_2$ radicals (Sect. 3), and retitling the paper to specify "bimolecular reactions". Although we recognize that Sect. 3 is preliminary, we feel it is nonetheless important that it is included. This is partly because some of the information it contains (e.g. the rate coefficients for the 1,4 hydroxyl H-shift reactions for stabilized $\alpha$-hydroxy peroxy radicals in Table 15) dovetails with information presented in the preceding paper on OH + aliphatic VOCs. In addition, it is important to emphasize that this is an important and fast moving topic area, which would be less well achieved by omitting the section completely.

**Comment B14:** 14.28 Table 15. Should make clear what are calculated and experimental determinations. Also, k298K of alpha-formyl peroxy radicals the rate should be 0.57 s-1 (typo).

Response: Thank you very much for spotting this error, which has been corrected.

**Comment B15:** 15.22 (and in SI) Assuming that the new mechanism will retain at least to the two radical pools produced following OH addition to isoprene, I do not understand why the 1,6 H-shift rates are not treated separately given there is significant evidence (Crounse, Teng) that a much larger fraction of the chemistry following addition at C4 will undergo this H-shift. Because the H-shift rates (not rate coefficients) for the C1 and C4 addition differ by an order of magnitude, use of the geometric mean will yield significant errors. Thus, I suggest it would be prudent to follow the recipe (if not the rates) described in Wennberg et al., 2018; Teng et al., 2018.

Response: We confirm that the method is exactly as the referee suggests for the isoprene-specific species. We think this is clearly stated at a number of points. The text starting from page 15 line 22 (discussing the generic rate coefficients in Table 15) reads as shown below. The final sentence indicates that the species-specific rate coefficients (rather than generic rate coefficient) are applied to the isoprene-derived species themselves:

"The rate coefficient assigned to the 1,6 hydroxyalkyl H-shift reaction is the geometric mean of rate coefficients applied to (Z)-CH$_2$(OH)C(CH$_3$)=CHCH$_2$O$_2$ (CISOPAO2) and (Z)-CH$_2$(OH)CH=C(CH$_3$)CH$_2$O$_2$ (CISOPCO2) in MCM v3.3.1. As discussed by Jenkin et al. (2015), those rate coefficients are derived from the LIM1 calculations of Peeters et al. (2014), but with some scaling to recreate the observations of Crounse et al. (2011; 2014). The generic rate coefficient is applied generally to unsaturated $\delta$-hydroxy peroxy radicals containing the sub-structure shown, but with the exceptions of CISOPAO2 and CISOPCO2 themselves, for which the species-specific rate coefficients are applied (see Sect. S6 and Table S5)."

Similarly, the relevant footnote (g) in Table 15 reads:

".... Applied generally to unsaturated $\delta$-hydroxy peroxy radicals containing the sub-structure shown, except for CISOPAO2 and CISOPCO2 themselves for which the species-specific rate coefficients are applied (see Table S5)."

Finally, Table S5 gives the species specific rate coefficients for the isoprene-derived species from MCM v3.3.1, with those from Wennberg et al. (2018) also provided in the footnotes to Table S5 and discussed in Sect. S5.

**Comment B16:** 16.1-9 The literature cited above goes some way towards meeting the recommendations presented in this paragraph. I'd recommend considering them in the 'rules' developed in this work.

Response: We thank the referee again for alerting us to the recent work, which will indeed help in the formulation of methods in ongoing work. We hope that the way we have dealt with this issue in the current paper is appropriate and acceptable.

**C. Comments by Luc Vereecken (Referee)**

**Comment C1**: Prof. Wennberg notes that "Teng was the first to point out that for multifunctional compounds, the nitrate branching ratios should (and do) scale more closely with heavy atoms than just carbon."

Historically, that is not quite accurate, as this has been discussed as far back as the turn of the century, and several models incorporated nitrate yields that are based on the number of heavy atoms, or even estimates that try to account for rigidity and other factors affecting quantum state density and hence lifetime/pressure dependence. Much of this was based on theoretical state density and partition function calculations, and this data was exchanged e.g. during Eurotrac meeting around the years 2000.

Mechanistically, it is clear that the pressure dependence is due to collisional stabilisation which, given that the energetics are not all that different between different RO2+NO reactions, is thus directly linked to the state density of the peroxy nitrite intermediate. This is mostly governed by the low-frequency modes, i.e. the number of modes generated by the molecular skeleton containing the heavy atoms, whereas the H-atoms only contribute by providing a bit of mass, a high-frequency modes that are barely excited at room temperature and thus don't contribute significantly to the state density. These theoretical state density calculations were used by e.g. Jozef Peeters to construct more complex models that weighted for e.g. double bonds and rings that do not contribute to high-density internal rotations and are thus not as effective as single-bonded chains in increasing the lifetime and hence nitrate yields.

In our work, such models were used as far back as 2001 (a-pinene oxidation, Peeters et al.), and as recent as 2012 ( b-pinene oxidation, Vereecken and Peeters) where the nitrate yields used do not match the Arey et al. model exactly, but rather are based at least on the number of heavy atoms, and sometimes accounted for double bonds and other effects. An example would be one of the first nitrate formation steps in Peeters et al. 2001, figure 1, formation of RO3, C10 Arey et al. tert nitrate yield 10% 10.45%; C10+O2 tert nitrate yield 11.11%, used yield is rounded 11%.

At that time, it was felt to be sufficient to refer to Arey et al., as the theory-based model was due to be published in full, and it was in many respects a theory-based reparameterization of the Arey et al. model. An unfortunate choice, as ultimately Peeters never published his model, despite extensive hints in in our papers that this was due to happen; the main block was that no theoretical characterization of the nitrite to nitrate interconversion process was ever available, suggested now to be either a roaming reaction or a singlet-triplet-singlet double surface hop, both of which are very hard to do computationally, and thus not characterized even today. Other authors did publish some work on this, e.g. Barker et al. 2003 probed the required energetic and rovibrational characteristics of the nitrite-nitrate interconversion process, but no computationally supported solution was ever found. Other scientists in those days likewise attempted to come up with models based on a quantification of the microscopic mechanism, but all faltered on the lack of a characterization of the nitrite-nitrate conversion step, as well as the odd differences in yields between primary, secondary and tertiary nitrates, which from a theoretical-mechanistic point of view remains unexplained. It may be that some models were presented as talks or posters at some conference, describing these efforts, but my memory does not stretches back that far, and I have only printed proceedings from this period, making searches too time-consuming for a merely historic reminiscence.

While it is possible that Teng et al. were the first to *explicitly* publish this finding in a peer-reviewed paper, the use of heavy atom number instead of carbon number in the prediction of nitrate yields thus dates back about 2 decades. Technically, for theoreticians, Barker et al. 2003 already indicates clearly that heavy atom count is more appropriate than carbon number, as that analysis is based on state density, and essentially only lacks a good description of the nitrite-nitrate conversion. The upcoming results on nitrate yields obtained at NCAR could likewise solve some of the conceptual problems related to prim/sec/tert yields that hampered development of theory-based models.

Feeling old, Luc Vereecken

Response: We thank Luc Vereecken for providing this informative comment. Some of the authors also recall discussions of this type within the EUROTRAC programme, and certainly the idea of

alternatives to carbon number (such as heavy atom number) in the parameterization of nitrate yields and other reactions has also been discussed in MCM meetings from about 10 years ago. We also look forward to further systematic information on the structural dependence of nitrate yields being reported, so that we can do a better job in representing the yields for the variety of structures formed in the future (see also response to Comment E1).

Similarly to yourself, we acknowledge that Teng et al. (2015) were the first to demonstrate the relationship to heavy atom number clearly and explicitly in relation to a systematic set of laboratory experimental data for oxygenated peroxy radicals and therefore feel that it is appropriate to cite that study at the relevant point.

**D. Comments by Geoffrey Tyndall (Referee)**

**Opening comment**: This manuscript, the next in a series describing protocols for the automatic generation of chemical mechanisms, addresses the reactions of organic peroxy radicals. Methods are given for the calculation of both overall rate constants and product branching ratios.

The manuscript is detailed, and addresses all or most of the possible reaction partners for RO2 in the atmosphere. This is a lot to cover, and the manuscript is at times a little scant, but in general does a good job at giving enough information to follow what the authors are trying to say.

Response: We are grateful to the referee for these supportive comments on our work. We acknowledge that the primary aim of the manuscript is give the necessary information to allow the estimation methods to be applied, rather than to provide a full review of the topic area. As indicated above (response to Reviewer A, General comments), however, we feel that we have presented an extensive set of information in support of our methods.

I have one relatively minor technical comment, plus a general observation about alkyl nitrate yields, following on from Luc Vereecken's comment.

**Comment D1:** Page 3, line 18. The first carbon atom in this RO2 radical seems to be missing some bonds. I suspect it is meant to be the oxo dihydroperoxy radical, so C(O)(OOH)CH2. . . etc

Response: Thank you very much for spotting this error (which also occured in Table 1). Quite a few people have read through this manuscript, and you are the first and only person to notice this. We correctly describe the species as a "complex oxo-di-hydroperoxy acyl peroxy radical" in footnote (o) of Table 1, but managed to omit the "oxo" group in the $RO_2$ structure, which should indeed read "$C(O)(OOH)CH_2CH_2CH_2CH(OOH)C(O)O_2$". This has been corrected in the text and Table 1.

**Comment D2** Further thoughts on the temperature dependence of alkyl nitrate yields.

In their 1987 paper, Atkinson et al. [1] parameterized the nitrate yield as a function of temperature and pressure, leading to a pressure dependent term, Yo(298)*[M] multiplied by a temperature dependence (T/300)^mo with mo = -2.99, and Yo(300) = Aexp(n), where n is the number of carbon atoms. The high pressure yield in this formulation had a temperature coefficient of -4.69.

In 1989, Carter and Atkinson [2] instead parameterized the ratio ka/kb, and found the best fit with mo = 0, and m(inf) = -8.0. So all the temperature dependence was in the high pressure limit, which leverages the whole curve down to low pressure.

Arey et al. (2001) [3] adopted this latter formulation to extrapolate their room temperature values to other temperatures.

In our 2012 review paper (Orlando and Tyndall, 2012) [4] we attempted to combine the low pressure and temperature dependent terms, using Yo(298)[M](T/298). This is of course erroneous, since if mo=0 the temperature dependence vanishes (other than that implicit in [M]).

It appears that Jenkin et al. (main manuscript Page 4, line 16; SI Page 2) copied our incorrect version in their current manuscript. It is possible that Carter, Atkinson and Arey have updated their fit at some point to include a (T/298) term. However, we cannot remember having seen this anywhere (although we are even older than Dr. Vereecken, and we may have forgotten it).

We apologize for introducing this error into the literature. Note that the formula given in Calvert et al. (2009) "Mechanisms of Atmospheric Oxidation of the Alkanes" is correct, while that in Calvert et al. (2015) "The Mechanisms of Reactions Influencing Atmospheric Ozone" is not.

[1] R. Atkinson, S. M. Aschmann, and A. M. Winer, J. Atmos. Chem., 5 (1987), 91. [2] W. P. L. Carter and R. Atkinson, J. Atmos. Chem., 8 (1989), 165. [3] J. Arey, S. M. Aschmann, E. S. C. Kwok, and R. Atkinson, J. Phys. Chem., A 105 (2001), 1020. [4] J. J. Orlando and G. S. Tyndall, Chem. Soc. Rev., 41 (2012) 6294.

Response: Thank you for communicating this error and for the additional information on where it appears. We have now corrected this, which we understand only requires the removal of the first (T/300) term. Because our example calculations (given in Sect. S1) are all for T = 298 K, this has no effect on the results (to three significant figures).

**E. Additional comment on nitrate yields from $RO_2$ + NO from John Crounse and Paul Wennberg (received shortly after the discussion closed)**

Comment E1: Do we interpret Table 3 correctly that the recommended beta-OO-OH + NO nitrate yields are based on equally weighted results from OBrien/Shepson, Matsunga/Ziemann, and Teng? The reason we raise this is that we understand there were potential analytical losses of these nitrates in the Shepson and Ziemann studies. In addition, O'Brien apparently did not account for $O(^3P)$ chemistry of the alkenes in their 1998 work, which seems to have been important in a number of their experiments. We discussed this in detail here:

https://www.atmos-chem-phys.net/15/4297/2015/acp-15-4297-2015-supplement.pdf

Response: Thank you for this helpful enquiry, and for reminding us of the potential interferences of $O(^3P)$ chemistry in the pioneering work of O'Brien, Shepson et al. (1998), as documented by Teng et al. (2015). Our basis for defining the effect of $\beta$-hydroxy groups was previously summarised in footnote (c) of Table 3, as follows:

"Based on a compromise of information from O'Brien et al. (1998), Matsunaga and Ziemann (2009; 2010), Yeh and Ziemann (2014b) and Teng et al. (2015) for $\beta$-hydroxy substituents, but also taking account of information reported for a number of other oxygenated systems (e.g. Tuazon et al., 1998a; Crounse et al., 2012; Lee et al., 2014) and previous consideration of the OH + isoprene system (Jenkin et al., 2015)."

Having reviewed our procedure, we can confirm that the O'Brien et al. (1998) data were not taken into account, and that reference to it should not have been included in the statement (and has been removed in the revised version of the paper). The yields calculated by our method are actually greater than those reported by O'Brien et al. (1998) by factors of 2 to 3. However, the approach is a compromise between the data reported by Ziemann and co-workers and by Teng et al.. Our calculated yields at the "high n plateau" are therefore about a factor of 1.4 greater than those reported by Matsunaga and Ziemann (2009) for linear alkenes, but under-estimate those reported by Teng. et al. (2015) for (lower n) terminal alkenes by a similar factor. They do, however, agree well with those reported by Tuazon et al. (1998). We also note that Teng et al. (2015) report lower yields for nitrates formed from internal alkenes (2-methylbut-2-ene, and 2,3-dimethylbut-2-ene). Our method recreates the reported value for 2-methylbut-2-ene very well (10.3 % vs. 9 ± 4 %), and presumably is also consistent with the (unspecified) preliminary lower yield for 2,3-dimethylbut-2-ene compared with hex-1-ene. At present, there is insufficient systematic information to provide different factors for $\beta$-hydroxy groups in different environments, such that a single factor is currently

applied to those formed from terminal acyclic alkenes, internal acyclic alkenes and cycloalkenes. We regard this as a reasonable compromise based on currently reported data, which can hopefully be improved upon when systematic data from a larger number of precursor alkenes/cycloalkenes is available. Ideally, such data would also allow the underlying function (based on Arey et al., 2001) to be optimised for different peroxy radical classes.

The Teng et al. (2015) data for β-hydroxy nitrates from terminal alkenes suggest no reduction in yield compared with those for alkyl nitrates containing the same number of heavy atoms. We considered using this as the basis for the effect of the beta-hydroxy group, but found that the calculated yields would overestimate those reported in almost all other studies. For example, the total calculated nitrate yield from OH + α-pinene would be about 29 %, compared with the reported value of (18 ± 9) % (Noziere et al, 1999) - and a gross overestimate of the 3.3 % hydroxynitrate yield reported very recently by Xu et al. (2019) (although we did not know that at the time). Similarly, the calculated nitrate yield from OH + isoprene at atmospheric pressure and the high NO limit (16 %) would be slightly outside the range of reported yields (4.4 - 14 %), although we recognise that the true value is likely towards the high end of the reported range.

As a result of this discussion, we have now included the following point in our recommendations list in Sect. 4:

[revised manuscript text omitted]